# IGF2BP3 recognizes m⁶A to regulate histone-to-protamine replacement during mouse sperm development

Dazhuang Wang[1,5], Zhenyi Huang[1,5], Yichun Zhou[1,5], Peiyan Chen[1,5], Gang Chang[2,5], Liwei Ke [1], Congying Jing[1], Haojie Yang[1], Jiexiang Zhao [1], Shaofang Ren[1], Yi Zheng[1], Yuhan Chen[1], Yunfan Xiang[3], Jun Liu [3] & Mei Wang [1,4✉]

## Abstract

Post-meiotic development of spermatids is under the control of a sophisticated RNA metabolic network, wherein the N6-methyladenosine (m⁶A) modification of mRNA, and proteins that bind to it, exert crucial functions in regulating sperm development from spermatogonia to spermatocytes. However, which m⁶A recognition proteins are involved in male post-meiotic spermiogenesis, and via which regulatory mechanisms, remains largely unknown. Here, we uncover the involvement of the m⁶A reader protein IGF2BP3 in the regulation of post-meiotic spermatid development. Genetic ablation of *Igf2bp3* results in spermatogenesis defects, leading to male sub-fertility or even infertility. Mechanistically, IGF2BP3 loss-of-function leads to the excessive translation of its target RNAs associated with histone-to-protamine replacement, particularly *Dot1l* and *Hdac11*. IGF2BP3 translationally represses these targets through its m⁶A-binding property and through its interaction with its binding partner YBX2. Sperm developmental defects of IGF2BP3 knockout mouse can be rescued by siRNAs targeting *Dot1l* and *Hdac11*. Collectively, our findings define the essential role of IGF2BP3-dependent regulation of protein biosynthesis in histone-to-protamine replacement during spermiogenesis, helping to understand the functions of m⁶A RNA modification in sperm development and male fertility.

**Keywords** IGF2BP3; m⁶A Reader; Translation Inhibition; Protamine Replacement; Spermiogenesis
**Subject Categories** Chromatin, Transcription & Genomics; Development; RNA Biology

## Introduction

Male germ cell development is precisely regulated by a sophisticated transcriptional network (Wang et al, 2018; Zhao et al, 2021). During these sequential processes, like post-meiotic spermiogenesis, a cluster of mRNAs needs to be synthesized and stored in messenger ribonucleoproteins (mRNPs), ensuring the translational fine-tuning for dosage and temporal control of specific proteins in spermatids. A group of RNA-binding proteins (RBPs) has been identified as critical regulators during spermiogenesis, controlling mRNA storage, elimination, and translation. For instance, Piwi-like RNA-mediated gene silencing 1 (Piwil1) has a dual function in mRNA decay and translation activation during spermiogenesis (Dai et al, 2019; Gou et al, 2014); YBX2 and YBX3 enable the mRNA storage in round spermatids by repressing its translation until they are needed for translation in elongating spermatids (Snyder et al, 2015; Yang et al, 2005a; Yang et al, 2007). However, the regulatory mechanism linking RNA stability and translation during post-meiotic spermatids has been largely unknown.

The post-meiotic phase of spermatogenesis involves spermatozoa maturation and several fundamental processes, such as chromatin compaction and flagellar development. Dysregulation at each step is highly associated with male infertility or subfertility (Bao and Bedford, 2016; Kleene, 2001, 2013; Steger, 2001; Zhang et al, 2020). During these processes, the histone-to-protamine transition is tightly coordinated with the chromatin remodeling and involves a group of epigenetic modification regulators, including some histone acetyltransferases, deacetylases, methyltransferases, and demethylases, which are dynamically expressed (Hundertmark et al, 2018; Li et al, 2022; Okada et al, 2007; Ou et al, 2025). Yet, whether these epigenetic regulators are precisely controlled at the transcriptional or translational levels by RNA-binding proteins remains to be fully elucidated.

N⁶-methyladenosine (m⁶A), the most prevalent mammalian internal mRNA modification, is present in all the spermatogenic cells, with particularly high enrichment in spermatids (Lin et al, 2017; Xu

[1]Guangdong Provincial Key Laboratory of Bone and Joint Degeneration Diseases, Department of Developmental Biology, School of Basic Medical Sciences, Southern Medical University, 510515 Guangzhou, P. R. China. [2]Department of Biochemistry and Molecular Biology, Shenzhen University Medical School, 518060 Shenzhen, P. R. China. [3]State Key Laboratory of Gene Function and Modulation Research, School of Life Sciences, Peking-Tsinghua Center for Life Sciences, Peking University, 100871 Beijing, P. R. China. [4]Department of Neonatology, Zhujiang Hospital, Southern Medical University, 510280 Guangzhou, P. R. China. [5]These authors contributed equally: Dazhuang Wang, Zhenyi Huang, Yichun Zhou, Peiyan Chen, Gang Chang. ✉E-mail: wangmei94@smu.edu.cn

et al, 2017). The dynamics of m⁶A are regulated by the interplay between RNA methyltransferases ('writers') and demethylases ('erasers'). Additionally, 'readers' or recognition proteins recognize m⁶A-modified RNAs and regulate their metabolism to perform their functions (Li et al, 2017; Liu et al, 2014; Wang et al, 2014; Wang et al, 2015; Zheng et al, 2013). During spermatogenesis, the functions and regulatory mechanisms of a cohort of m⁶A binding proteins have been extensively illuminated (Hsu et al, 2017; Kasowitz et al, 2018; Lasman et al, 2020; Qi et al, 2022; Tan et al, 2023b). Yet, it is still uncertain which binding protein is responsible for regulating post-meiotic m⁶A-modified RNAs during spermiogenesis. Insulin-like growth factor 2 mRNA-binding protein 3 (IGF2BP3) belongs to the family member of IGF2BPs, which includes IGF2BP1, IGF2BP2, and IGF2BP3. IGF2BP3, a well-known RBP and newly-identified m⁶A recognition protein, is highly conserved between zebrafish, *Xenopus*, chicken, mouse, and human (Deshler et al, 1998; Hammer et al, 2005; Huang et al, 2018; Huttelmaier et al, 2005; Mori et al, 2001; Ren et al, 2020; Sun et al, 2019; Vong et al, 2021; Zaccara et al, 2019). Recently, the pathological implications of IGF2BP3 in different types of cancers have been well-documented (Huang et al, 2018; Tang et al, 2023; Tran et al, 2022; Yang et al, 2023). Besides, it actively participates in oocyte, embryo, and placenta development (Li et al, 2014; Mori et al, 2001; Ren et al, 2020; Vong et al, 2021). However, the predominantly expression of IGF2BP3 in adult human and mouse testis led us to consider its possible involvement in spermatogenesis (Hammer et al, 2005).

In this study, we systematically identified the expression dynamics of IGF2BP3 from pachytene spermatocytes to elongating spermatids in mouse testes. Genetic ablation of mouse *Igf2bp3* resulted in male subfertility or infertility, characterized by sperm developmental defects and histone retention. The underlying mechanism involved in aberrant translational activation of m⁶A-modified RNAs related to histone-to-protamine transition when IGF2BP3 was deleted during spermiogenesis. These findings highlight the physiological significance of IGF2BP3 in sperm development and provide insights into its translational repression in male fertility.

# Results

## IGF2BP3 locates in late spermatocytes to elongating spermatids in mouse testes

In previous studies, mRNA m⁶A modifications had shown enrichment in late spermatocytes and spermatids compared to that in spermatogonia and early spermatocytes (Lin et al, 2017). To determine the m⁶A binding proteins in post-meiotic spermatids, we examined the coding genes of reported m⁶A binding proteins. It showed the highly specific expression of *Igf2bp3* in spermatids with our previously described mouse testicular single-cell RNA sequencing (scRNA-seq) data (Zhao et al, 2021) (Appendix Fig. S1A), implying the potential involvement of IGF2BP3 in the regulation of m⁶A-modified mRNAs during sperm maturation. Subsequently, western blot analysis was performed to determine its predominant expression in the adult mouse testis and a relative enrichment from the 2-week to 4-week testes, suggesting the active expression in spermatocytes and spermatids (Fig. 1A,B). Immunostaining of the mouse testis and epididymis further confirmed that IGF2BP3 exhibited low-level expression in spermatogonia, showed relative

accumulation from pachytene spermatocytes to step 12 spermatids, but not in epididymal spermatozoa (Fig. 1C–F; Appendix Fig. S1B). Moreover, the rare occurrence of IGF2BP3 in mouse testicular somatic cells was observed at both transcriptional and protein levels (Appendix Fig. S1C,D). Overall, these results demonstrated the specific localization of IGF2BP3 from pachytene spermatocytes to early stage of elongating spermatids in adult mouse testes.

## IGF2BP3 deficiency impairs sperm development and male fertility

Then, gene knockout mice were generated to evaluate the function of IGF2BP3 in spermatogenesis. Structurally, IGF2BP3 possesses two RNA recognition motifs (RRM) and four K-homology (KH) domains, with the KH domains playing a crucial role in RNA binding and regulation (Degrauwe et al, 2016). Therefore, Cas9 mRNA and gRNA targeting the KH1 domain in exon 6 of mouse *Igf2bp3* locus were co-injected into pronucleus zygotes to produce founder mouse with frameshift mutants (Fig. EV1A). Both female and male heterozygous knockout mice (*Igf2bp3*⁺/⁻) were fertile to obtain homozygous knockout mice (*Igf2bp3*⁻/⁻), of which the IGF2BP3 deletions were validated in testes via western blot and immunostaining analysis (Fig. EV1B,C). Moreover, off-target effects of CRISPR-Cas9 were not observed in *Igf2bp3*⁻/⁻ testis (Dataset EV1). Meanwhile, the expression of IGF2BP1 as well as IGF2BP2 did not show any perturbations in *Igf2bp3*⁻/⁻ testis, excluding the redundancy effect of the IGF2BPs family (Fig. EV1C).

Furthermore, adult *Igf2bp3*⁻/⁻ males exhibited sub-fertility or even sterility, with testes approximately half the size of those in *Igf2bp3*⁺/⁺ (wild-type, WT) and *Igf2bp3*⁺/⁻ mice (Fig. 2A–C). The concentration and motility of epididymal sperm from *Igf2bp3*⁻/⁻ males were significantly declined compared to those from WT and heterozygous knockout mice (Figs. 2D,E and EV1D,E; Movies EV1–3). Besides, an increase in the rates of malformation and DNA damage was observed in *Igf2bp3*⁻/⁻ spermatozoa compared to controls (Fig. 2F,G). Collectively, male mice lacking IGF2BP3 displayed composite phenotypes resembling human oligoasthenoteratozoospermia. HE staining revealed the presence of all types of spermatogenic cells and somatic cells within *Igf2bp3*⁻/⁻ seminiferous tubules (Fig. 2H). However, a significantly higher number of cells with abnormal nuclei were detected in the central region of *Igf2bp3*⁻/⁻ seminiferous tubules, accompanied with a decrease in the cross-sectional area of each tubule (Figs. 2H and EV1F,G). Furthermore, the loss of haploid spermatids (including round spermatids (RS), elongating spermatids (ES), and condensed spermatids (CS)) was found in *Igf2bp3*⁻/⁻ testes without significantly affecting their proportion of spermatogonia, spermatocytes, and Sertoli cells (Figs. 2I and EV1H; Appendix Fig. S2A–D). Consistently, TUNEL signals predominantly accumulated in ES and CS of *Igf2bp3*⁻/⁻ mice compared to those of *Igf2bp3*⁺/⁻ mice (Fig. 2J,K), indicating the necessity of IGF2BP3 in the spermatid maturation.

## IGF2BP3-deletion disturbs histone-to-protamine exchange in mouse spermatids

Subsequently, the ultrastructure of spermatozoa was examined using electron microscopy, revealing nuclear compaction defects in IGF2BP3-deleted sperm (Fig. 3A). The histone-to-protamine

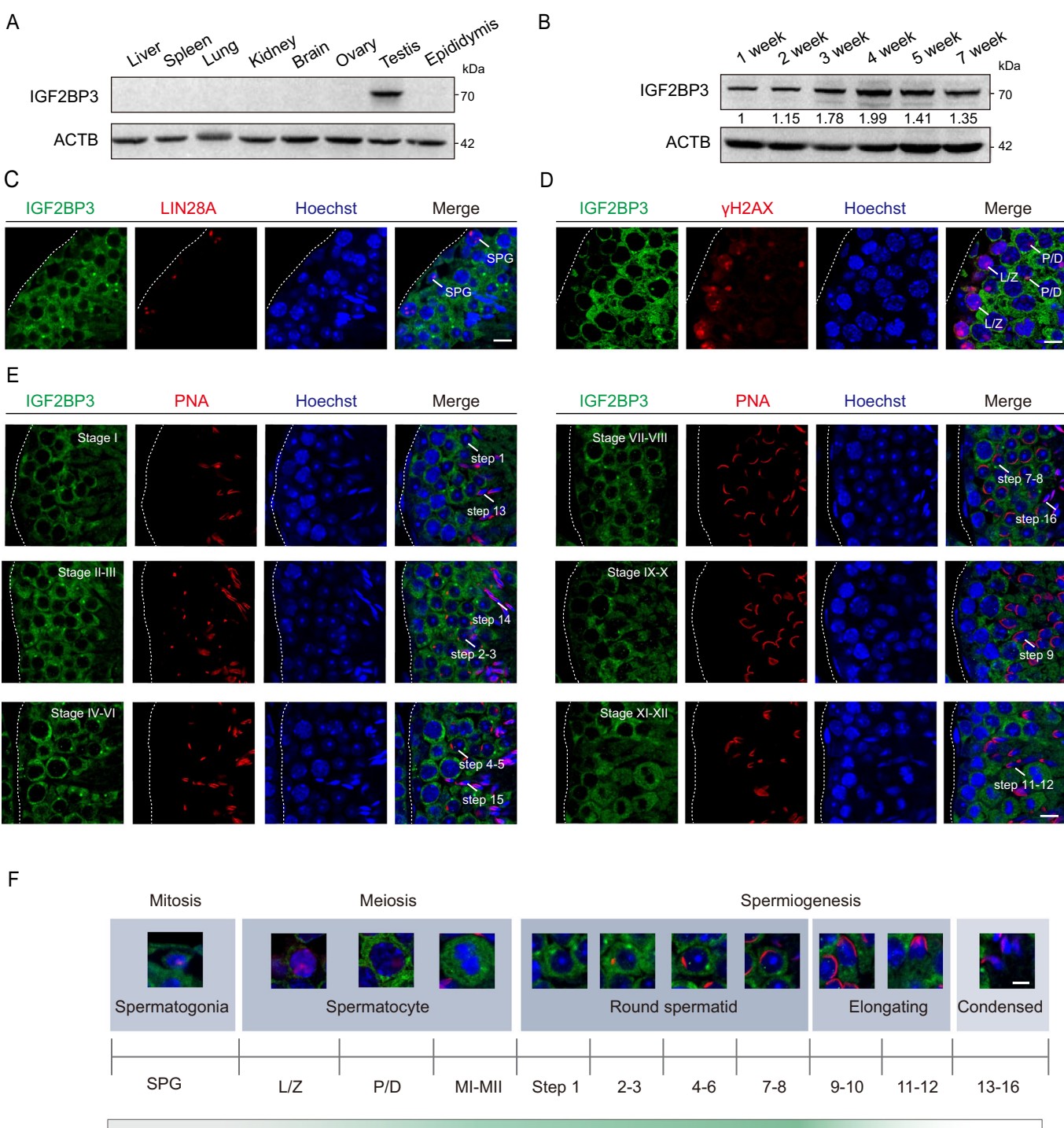

**Figure 1.  IGF2BP3 is expressed from pachytene spermatocytes to elongating spermatids in mouse testes.**

(**A, B**) Western blotting analysis of the protein levels of IGF2BP3 in adult mouse tissue extracts (**A**), and mouse testes at different postnatal weeks (**B**) with ACTB as the internal control. The values below each band represent the relative expression levels of IGF2BP3. (**C**) Immunofluorescence of IGF2BP3 (green) and LIN28A (red) in adult testicular paraffin sections from 8-week-old mice. SPG, spermatogonia. Scale bar, 10 μm. (**D**) Immunofluorescence of IGF2BP3 (green) and γH2AX (red) in adult testicular paraffin sections from 8-week-old mice. L/Z leptotene or zygotene spermatocytes, P/D pachytene or diplotene spermatocytes. Scale bar, 10 μm. (**E**) Immunofluorescence of IGF2BP3 (green) and PNA (red) in adult testicular paraffin sections showing with distinct stages of the seminiferous epithelium cycles from 8-week-old mice. The signals of IGF2BP3 in different steps of spermatids were indicated, respectively. Scale bars, 10 μm. (**F**) Scheme of IGF2BP3 expression dynamics during spermatogenesis. Green marks the protein level of IGF2BP3. SPG spermatogonia, L/Z leptotene or zygotene spermatocytes, P/D pachytene or diplotene spermatocytes, MI-MII metaphase I to II. Scale bars, 5 μm. Dotted borders in (**C–E**) demarcates the basement membrane of the seminiferous tubule. Source data are available online for this figure.

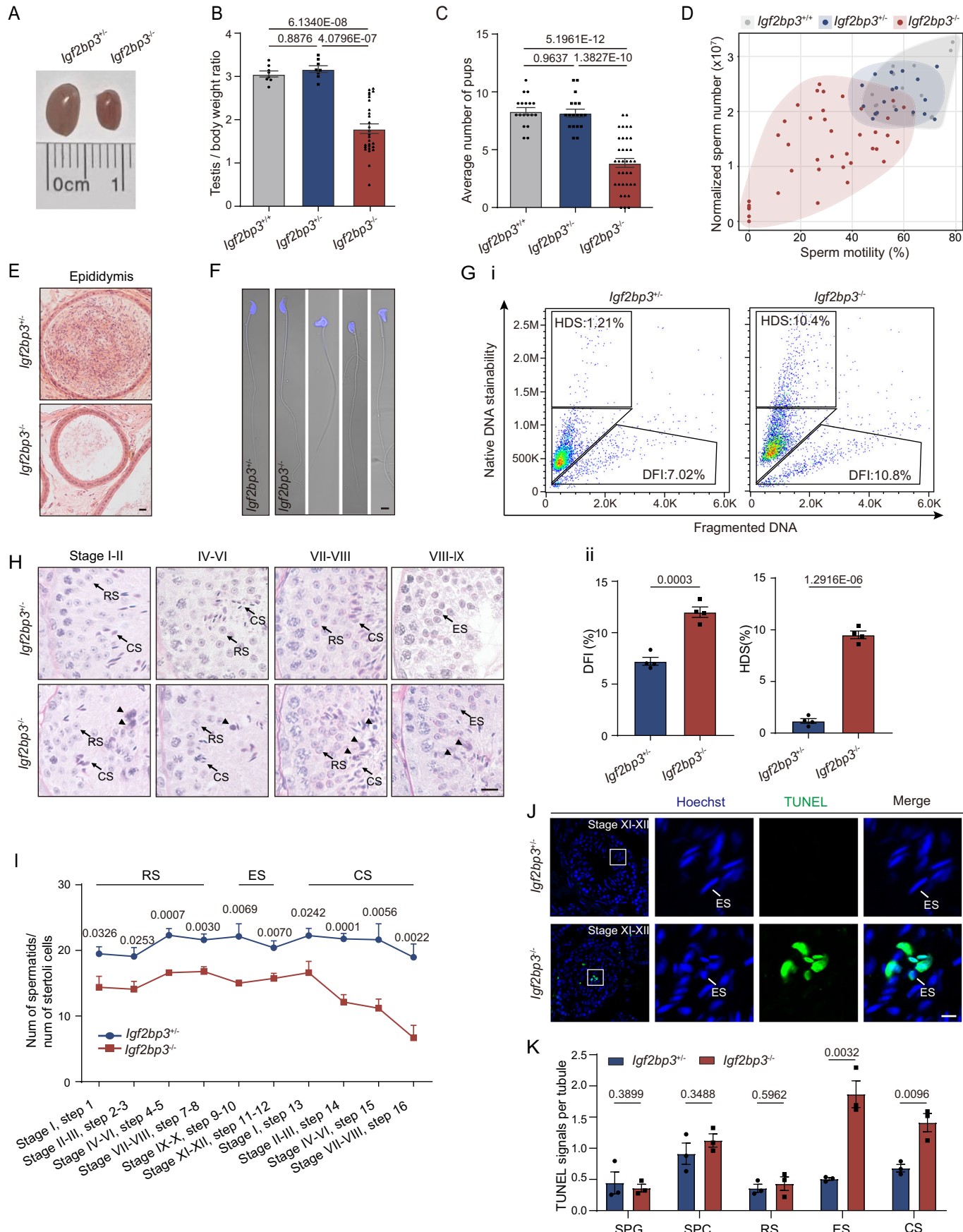

**Figure 2. IGF2BP3 deficiency leads to spermatogenesis defects and spermatids loss.**

(A) Morphology of representative testes from adult *Igf2bp3*⁺/⁻ and *Igf2bp3*⁻/⁻ mice. (B) Bar plot showing the ratios of testis weight (mg) to body weight (g) between adult *Igf2bp3*⁺/⁺ (n = 8), *Igf2bp3*⁺/⁻ (n = 8), and *Igf2bp3*⁻/⁻ (n = 28) mice. P values were calculated by one-way ANOVA. Each bar represents the mean ± SEM from biological replicates. (C) Bar plot showing the litter size of adult *Igf2bp3*⁺/⁺ (n = 18), *Igf2bp3*⁺/⁻ (n = 18), and *Igf2bp3*⁻/⁻ (n = 36) males mating with wild-type females. Each male mated with at least two females and the plugged females were counted. P values were calculated by one-way ANOVA. Each bar represents the mean ± SEM from biological replicates. (D) Scatter plot showing the percentage of motile sperm and the corresponding concentration of epididymal spermatozoa from *Igf2bp3*⁺/⁺ (n = 16), *Igf2bp3*⁺/⁻ (n = 18), and *Igf2bp3*⁻/⁻ (n = 39) mice. (E) H&E staining of caudal epididymis sections from adult *Igf2bp3*⁺/⁻ and *Igf2bp3*⁻/⁻ males. Scale bar, 20 μm. (F) Representative DIC micrograph images of spermatozoa from adult *Igf2bp3*⁺/⁻ and *Igf2bp3*⁻/⁻ mice with nuclei counterstained with Hoechst (blue). Scale bar, 20 μm. (G) Flow cytometry analysis (i) and its corresponding statistical results (ii) of the DNA fragmentation index (DFI) and high DNA stainability (HDS) of epididymis spermatozoa from 8-week-old *Igf2bp3*⁺/⁻ and *Igf2bp3*⁻/⁻ mice. Unpaired two-tailed t test. Each error bar represents the mean ± SEM from biological replicates. (H) PAS staining of adult testicular paraffin sections showing with distinct stages of the seminiferous epithelium cycles from 8-week-old *Igf2bp3*⁺/⁻ and *Igf2bp3*⁻/⁻ mice. Arrowheads indicate cells with abnormal agglutinated nuclei. RS round spermatids, ES elongating spermatids, CS condensed spermatids. Scale bar, 10 μm. (I) Ratios between spermatids and Sertoli cells in tubule cross sections of specific stages of seminiferous epithelial cycles and corresponding spermatid development steps were shown. Unpaired two-tailed t test. Error bars, n = 4 biological replicates, mean ± SEM. (J) TUNEL assay of adult testicular paraffin sections from *Igf2bp3*⁺/⁻ and *Igf2bp3*⁻/⁻ mice. Scale bar, 5 μm. (K) Bar plot showing the numbers of TUNEL-positive spermatogenic cells per tubule in *Igf2bp3*⁺/⁻ and *Igf2bp3*⁻/⁻ testes. SPG spermatogonia, SPC spermatocytes. Unpaired two-tailed t test. Each error bar represents the mean ± SEM from three biological replicates. Source data are available online for this figure.

exchange is a crucial process facilitating nuclear agglutination during spermatid maturation in testes, thereby safeguarding the paternal genome against damage and mutagenesis (Gou et al, 2017; Rathke et al, 2014). However, *Igf2bp3* knockout (*Igf2bp3*-KO) epididymal spermatozoa exhibited increased retention of histones (H2A, H2B, H3, and H4) and decreased accumulation of protamines (PRM1 and PRM2) compared to control spermatozoa, confirming the defects of histone-to-protamine replacement in *Igf2bp3*⁻/⁻ testes (Fig. 3B,C).

During histone-to-protamine exchange, somatic histones are initially replaced by testis-specific histone variants accompanied with histone modification from round spermatids, followed by the transition proteins (TNPs) incorporation into spermatid nuclei prior to the ultimate replacement of protamines (PRMs) in late spermatids, achieving highly condensed genomic structures (Barral et al, 2017; Wang et al, 2019). To investigate the defects within this process, high-purity (95%) round spermatids from adult *Igf2bp3*⁺/⁻ and *Igf2bp3*⁻/⁻ testes were isolated for subsequent analysis, respectively (Fig. EV2A–D). We found that mRNAs encoding testis-specific histone variants (*H1t*, *Hils1*, *H2al1*, and *H2afb1*), transition proteins (*Tnp1* and *Tnp2*), and protamines (*Prm1* and *Prm2*) were significantly downregulated in *Igf2bp3*-KO round spermatids compared to control round spermatids (Figs. 3D and EV2E). Accordingly, histone modifications facilitating the eviction of histones, such as H4K5ac, H4K8ac, and H4K12ac (H4K5/8/12ac) (Goudarzi et al, 2016; Shiota et al, 2018), showed substantial decline in *Igf2bp3*⁻/⁻ RS and ES compared to those in control (Figs. 3E and EV2F); meanwhile, immunofluorescence staining and western blot analysis also demonstrated a significantly reduced accumulation of TNP2, PRM1, and PRM2 in *Igf2bp3*-null haploid cells, especially the ES and CS (Figs. 3F,G and EV2G–K). Thus, genetic ablation of *Igf2bp3* disrupted the histone-to-protamine replacement during spermiogenesis.

## IGF2BP3 participates in spermiogenesis by repressing RNA translation

To explore the regulatory property of IGF2BP3 in spermiogenesis, we performed enhanced crosslinking and immunoprecipitation sequencing (eCLIP-seq) on adult mouse testes, and the eCLIP-seq data fully satisfied the stringent quality control standards from the ENCODE Project and region-based fold enrichment analysis

(Fig. EV3A,B; Appendix Fig. S3A). A total of 5,095 overlapping target genes were identified from two replicate eCLIP-seq IP and sized-match INPUT (SMInput) pairs for further analysis, which contained 16,579 and 28,494 peaks, respectively (Fig. EV3C). Approximately 96% of the IGF2BP3-binding sites were located within protein-coding transcripts (mRNAs) and were highly concentrated in 3′ untranslated regions (3′UTRs) (Figs. 4A,B and EV3B). Considering the predominant expression of IGF2BP3 in spermatogenic cells, we characterized 3,007 transcripts as germ cell-specific targets with high binding affinity of IGF2BP3 by integrating eCLIP-seq and scRNA-seq data from mouse testes (Fig. 4C; Appendix Fig. S3B; Dataset EV2) (Zhao et al, 2021).

Due to the cytoplasmic localization of IGF2BP3 in testes and the initial defects in round spermatids, we performed RNA-seq and ribosome nascent-chain complex sequencing (RNC-seq) on haploid round spermatids from *Igf2bp3*⁺/⁻ and *Igf2bp3*⁻/⁻ testes to dissect the post-transcriptional and translational regulatory mechanism(s) underlying male infertility caused by IGF2BP3 knockout (Fig. 4D; Appendix Fig. S3C–F). RNA-seq identified 376 downregulated and 174 upregulated differentially expressed genes in *Igf2bp3*⁻/⁻ RS compared to control RS, respectively. However, transcriptome and translatome analysis revealed that 426 targets out of 2141 genes exhibited elevated translation efficiency (TE), while 37 targets in 543 genes showed decreased TE in *Igf2bp3*⁻/⁻ RS compared to the control RS (Figs. 4E,F and EV3E). Notably, both of these gene clusters were significantly associated with 'microtubule-based process' and 'chromatin organization', aligning well with the observed phenotypes in *Igf2bp3*-KO sperm (Fig. EV3D; Dataset EV3).

## IGF2BP3 prevents the protein overproduction of *Dot1l* and *Hdac11* during spermatid maturation

Considering the dramatic changes at the translational level, we conducted GO analyses on 426 targets of IGF2BP3 with upregulated TE to further dissect the key targets of IGF2BP3 contributing to spermatids loss in *Igf2bp3*⁻/⁻ testes, and it indicated the defects related to 'chromatin organization' and 'microtubule-based movement' in IGF2BP3-KO testes once again (Figs. 4F,G and EV3E; Dataset EV4). A comprehensive analysis to the genes associated with 'chromatin organization' was further performed due to the observed failure in chromatin condensation in IGF2BP3-

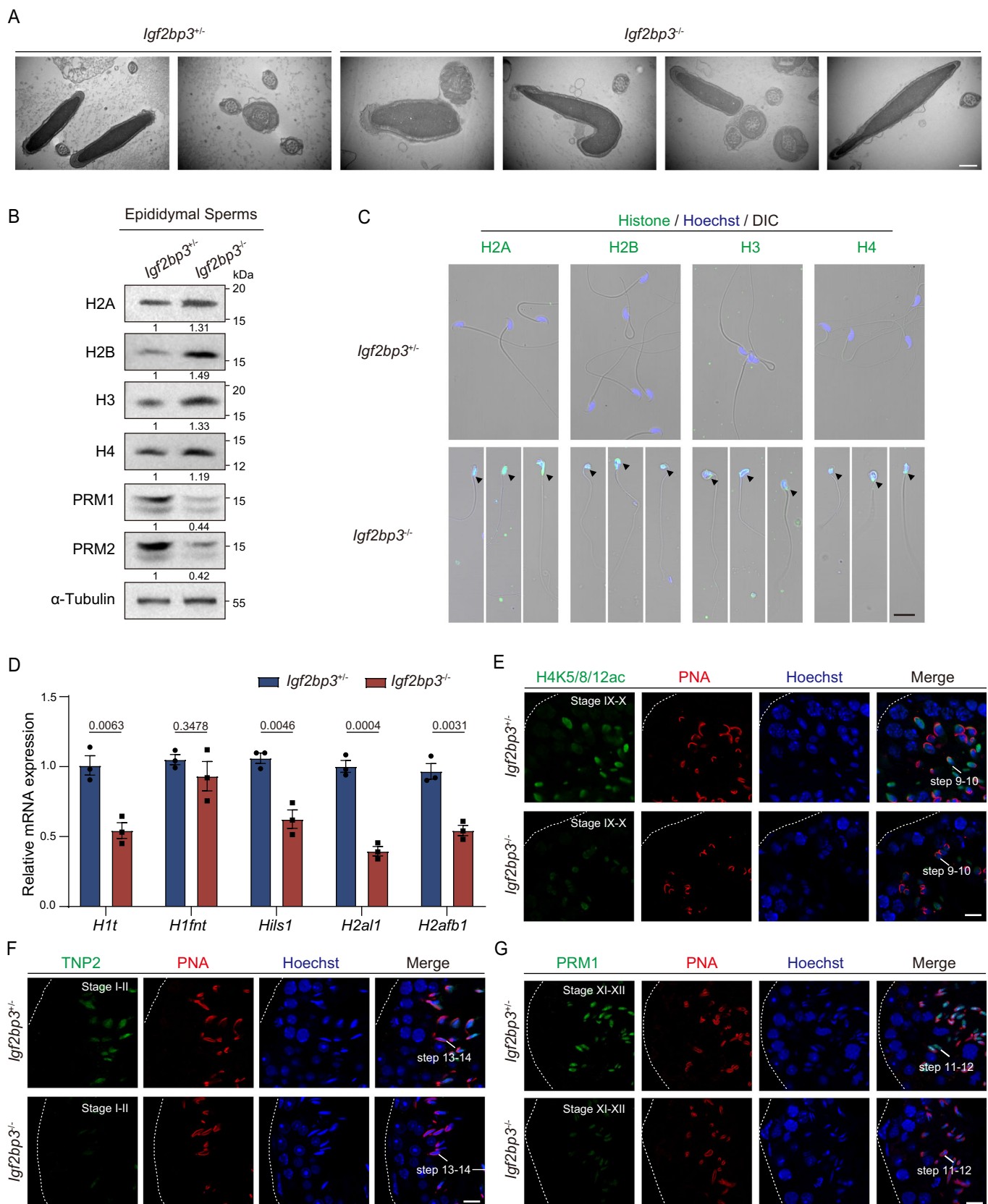

**Figure 3.   IGF2BP3 deficiency leads to impaired histone-to-protamine exchange.**

(A) Transmission electron microscopy analysis of epididymal spermatozoa showing impaired nuclear compaction in *Igf2bp3*⁻/⁻ compared to *Igf2bp3*⁺/⁻ sperm. Scale bar, 500 nm. (B) Western blotting analysis of the protein levels of H2A, H2B, H3, H4, PRM1 and PRM2 in epididymal spermatozoa from *Igf2bp3*⁺/⁻ and *Igf2bp3*⁻/⁻ mice. The values below each band represent the relative expression levels of each protein, normalized using α-Tubulin as a loading control. (C) Immunofluorescence of H2A, H2B, H3, and H4 (green) in spermatozoa from adult *Igf2bp3*⁺/⁻ (top) and *Igf2bp3*⁻/⁻ (bottom) mice. Arrowheads indicate the histone as shown. (D) qPCR analysis of the relative expression levels of *H1t*, *H1fnt*, *Hils1*, *H2al1*, and *H2afb1* normalized to *β-Actin* in round spermatids from adult *Igf2bp3*⁺/⁻ and *Igf2bp3*⁻/⁻ testes. Unpaired two-tailed *t* test. Each error bar represents the mean ± SEM from three biological replicates. (E) Immunofluorescence of H4K5/8/12/ac (green) and PNA (red) in adult testicular paraffin sections from 8-week-old *Igf2bp3*⁺/⁻ and *Igf2bp3*⁻/⁻ mice. Scale bar, 10 μm. (F) Immunofluorescence of TNP2 (green) and PNA (red) in adult testicular paraffin sections from 8-week-old *Igf2bp3*⁺/⁻ and *Igf2bp3*⁻/⁻ mice. Scale bar, 10 μm. (G) Immunofluorescence of PRM1 (green) and PNA (red) in adult testicular paraffin sections from 8-week-old *Igf2bp3*⁺/⁻ and *Igf2bp3*⁻/⁻ mice. Scale bar, 10 μm. Dotted borders in (E–G) demarcates the basement membrane of the seminiferous tubule. Source data are available online for this figure.

KO spermatozoa. And 37 genes in this term were categorized into three clusters exhibiting dynamic expression during spermatogenesis (Fig. EV3F). The significant binding enrichment of IGF2BP3 towards transcripts actively expressed in spermatocytes and spermatids within clusters 2 and 3 was validated in testes, especially for *Hdac11*, *Dot1l*, and *Chd5* (Fig. 4H; Appendix Fig. S3G).

Notably, DOT1L and HDAC11 proteins were overproduced in *Igf2bp3*-deficient RS and ES, consistent with their translational efficiency elevation in IGF2BP3-KO RS without excessive transcription (Figs. 4I,J and EV3G,H). Immunofluorescence staining further substantiated the co-localization of IGF2BP3 with DOT1L and HDAC11 in late spermatocytes to the early elongating spermatids, indicating the synchronized transcription and translation of *Dot1l* and *Hdac11* during spermiogenesis (Appendix Fig. S3H). It is well known that DOT1L is the sole H3K79 methyltransferase in mammals (Feng et al, 2002; Lin et al, 2023), and HDAC11 belongs to the histone deacetylases family (Gao et al, 2002; Villagra et al, 2009). Thus, the aberrant accumulation of DOT1L and HDAC11 was strongly associated with the increased H3K79me2 and decreased H4K5/8/12ac levels in *Igf2bp3*⁻/⁻ RS and ES (Figs. 3E, 4K, 4L, EV2F, and 3I).

## IGF2BP3 mediates RNA translation repression by interacting with YBX2 in spermiogenesis

To elucidate the potential co-factors of IGF2BP3 involved in RNA translational repression, we performed the immunoprecipitation of IGF2BP3 complex in mouse round spermatids, followed by comprehensive mass spectrometry analysis. YBX2, a well-known RNA translational repressor in testes, was identified as the top partner alongside IGF2BP3 (Fig. 5A,B; Dataset EV5). Besides, Co-IP analysis also confirmed their co-localization without the interference of RNAs in testes, suggesting a direct collaboration in regulating RNA metabolism (Fig. 5C). Considering the established role of YBX2 as an RBP, we overlapped 3,007 germ cell-specific eCLIP-seq targets of IGF2BP3 with 245 CLIP-seq targets of YBX2, however, there was limited concurrence between these two clusters (Appendix Fig. S4A) (Xu et al, 2009). Meanwhile, although the level of YBX2 remained unchanged in *Igf2bp3*-KO testes, its binding capacity to *Dot1l* and *Hdac11* was significantly reduced compared to that in control testes (Fig. 5D; Appendix Fig. S4B–D). Together, these results established that the RNA binding capacity of IGF2BP3 was not affected by the presence of YBX2, suggesting a predominantly translational regulatory role of YBX2 in the IGF2BP3-YBX2 complex during spermatogenesis.

Since IGF2BP3 primarily binds to the 3′UTR of its targets and suppresses their translational efficiency in testes, we constructed bidirectional expression vectors containing CopGFP with the 3′UTR of the interested RNAs in one direction, while mCherry was included as a transfection control in the opposite direction to investigate the regulatory mechanism of IGF2BP3-YBX2 complex quantitatively (Fig. 5E). To avoid the interference of endogenous IGF2BP3, bidirectional vectors with mouse *Igf2bp3* or *Ybx2* expression vectors were transfected into IGF2BP3-deleted HEK293T cells (Appendix Fig. S4E,F). Notably, mouse IGF2BP3 and YBX2 exhibited mutual interaction even in the absence of RNA in HEK293T cells, which was consistent with their binding mode observed in testis (Fig. 5F). We subsequently observed a simultaneous increase in both *CopGFP* and its protein abundance solely regulated by IGF2BP3, suggesting an RNA stabilization role without YBX2. However, IGF2BP3-YBX2 complex significantly reduced the translation efficiency (protein-to-RNA ratio) of CopGFP under the regulation of *Dot1l* 3′UTR, with *Actb* (non-target of IGF2BP3) serving as a negative control. These results demonstrated the RNA translation repression mediated by IGF2BP3-YBX2 complex, reflecting a similar mechanism within testes (Fig. 5G–I; Appendix Fig. S4G–I).

## IGF2BP3 recognizes and regulates *Dot1l* and *Hdac11* dependent on their m⁶A modification during spermiogenesis

Given the substantial enrichment of m⁶A-modified RNAs in post-meiotic germ cells, the relationship between m⁶A-containing RNAs and IGF2BP3 recognition during spermiogenesis was further investigated. First, the affinity of IGF2BP3 as an m⁶A binding protein in mouse testes was supported by the RNA pulldown, RIP LC-MS/MS, and dot blotting analyses using testicular or purified IGF2BP3 (Figs. 6A and EV4A–D). Moreover, the m⁶A motifs 'GAACA' and 'AAACC' were identified in the consensus sequence of overlapped IGF2BP3-binding peaks (Fig. EV4E). Together with 87% of germ cell specific targets containing m⁶A modification, it demonstrated the recognition of IGF2BP3 to m⁶A RNAs in mouse testes (Figs. 6A and EV4F) (Lin et al, 2017). Specifically, IGF2BP3 exhibited a tendency to bind to the m⁶A-enriched regions in 3′UTR of *Dot1l* and *Hdac11* in round spermatids (Fig. 6B,C).

Moreover, compared to target RNAs of IGF2BP3 in *Igf2bp3*⁺/⁻ RS, as m⁶A-modified peaks increased, there was a corresponding higher decline in RNA abundance and elevation in RNA translation efficiency in *Igf2bp3*⁻/⁻ RS (Figs. 6D,E and EV4G). Accordingly,

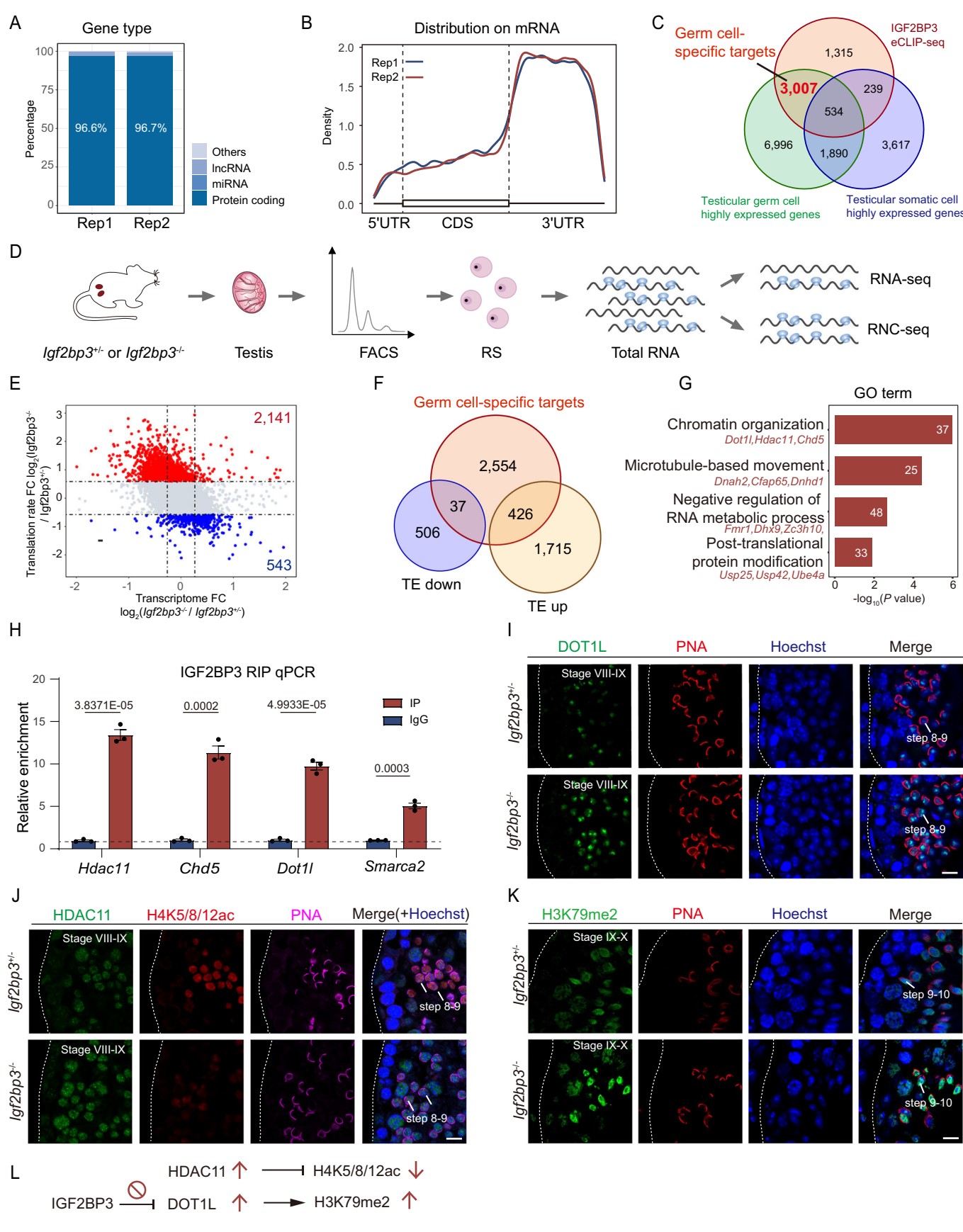

◀ **Figure 4. IGF2BP3 deletion leads to the overproduction of Histone-modifying enzymes HDAC11 and DOT1L in IGF2BP3-KO spermatids.**

(A) Distribution of IGF2BP3 peaks ($\log_2$(fold change) > 2 and $-\log_{10}$(P value) >2) across distinct gene types annotated by Ensembl GRCm38 v92. (B) Density plot showing the distribution of IGF2BP3-binding peaks ($\log_2$(fold change) > 2 and $-\log_{10}$(P value) >2) within different gene regions. CDS coding sequences, UTR untranslated regions. (C) Venn diagram showing the overlap between IGF2BP3 eCLIP targets of mouse testes (in the current study), germ cell-specific expressed genes, and testicular somatic cell-specific genes identified in mouse testicular scRNA-seq data (Zhao JX et al, 2021). (D) Scheme view of experimental design for investigating mRNA translation rates with polysomal and RNA-seq in RS of adult *Igf2bp3*$^{+/-}$ and *Igf2bp3*$^{-/-}$ mouse testes. (E) Scatter plot showing mRNA-level changes (x axis) against TE changes (y axis) between adult *Igf2bp3*$^{-/-}$ and *Igf2bp3*$^{+/-}$ round spermatids. TE-up genes in *Igf2bp3*$^{-/-}$ round spermatids are shown in red and TE-down genes are in blue. (F) Venn diagram showing the overlap between 3007 germ cell-specific targets of IGF2BP3, 2141 and 543 genes with up- and downregulated TE in *Igf2bp3*$^{-/-}$ RS compared to the control RS. (G) The GO terms enriched in 426 targets of IGF2BP3 with upregulated TE in *Igf2bp3*$^{-/-}$ RS compared to the control RS. (H) IGF2BP3 RIP-qPCR analysis of the relatively enrichment of indicated transcripts in adult WT mouse testes. Unpaired two-tailed *t* test. Error bars, n = 3 biological replicates, mean ± SEM. (I) Immunofluorescence of DOT1L (green) and PNA (red) in adult testicular paraffin sections from 8-week-old *Igf2bp3*$^{+/-}$ and *Igf2bp3*$^{-/-}$ mice. Scale bar, 10 μm. (J) Immunofluorescence of HDAC11 (green), H4K5/8/12ac (red), and PNA (magenta) in adult testicular paraffin sections from 8-week-old *Igf2bp3*$^{+/-}$ and *Igf2bp3*$^{-/-}$ mice. Scale bar, 10 μm. (K) Immunofluorescence of H3K79me2 (green) and PNA (red) in adult testicular paraffin sections from 8-week-old *Igf2bp3*$^{+/-}$ and *Igf2bp3*$^{-/-}$ mice. Scale bar, 10 μm. (L) Illustration of the levels of DOT1L, HDAC11, H3K79me2, and H4K5/8/12ac during spermiogenesis in *Igf2bp3*$^{+/-}$ and *Igf2bp3*$^{-/-}$ spermatids. Dotted borders in (I–K) demarcates the basement membrane of the seminiferous tubule. Source data are available online for this figure.

*Mettl3* (an m⁶A-methyltransferase) knockdown in GC-2 cells resulted in decreased levels of m⁶A modification on *Dot1l*, accompanied by reduced interaction between IGF2BP3 and *Dot1l* at the same region without significantly affecting their expression (Figs. 6F,G and EV4H–J). More importantly, when we inserted a mutant m⁶A motif-enriched 3′UTR of mouse *Dot1l* or *Hdac11* into our previously established 3′UTR-regulated expression vector transfected IGF2BP3-deleted HEK293T cells, exogenous mouse IGF2BP3 exhibited much lower binding affinity towards CopGFP mRNA compared to one with WT m⁶A motif-enriched 3′UTR (Fig. 6H,I); normalizing the CopGFP protein to the mCherry protein and then to the corresponding mRNA levels, the results showed that the IGF2BP3-YBX2 complex reduced the translation efficiency of CopGFP containing the m⁶A motif modification in the 3′UTR of *Dot1l* and *Hdac11* (Fig. 6J–L; Appendix Fig. S5A–D).

In addition, to determine whether the regulation of *Dot1l* primarily depends on the m⁶A binding activity of IGF2BP3 rather than that of YBX2, we constructed a modified IGF2BP3 with the motif GXXG (which corresponds to the m⁶A binding regions) in the KH3-4 domains being mutated to GEEG (Fig. EV4K) (Fakhar et al, 2024; Huang et al, 2018; Huttelmaier et al, 2005). Co-IP analysis revealed that the mutation in the KH3-4 domains of IGF2BP3 did not affect their mutual interaction with YBX2 even in the absence of RNAs as the scaffold (Fig. EV4L). Furthermore, neither in the presence nor absence of YBX2 cooperation could the mutant mouse IGF2BP3 exert translational repression on CopGFP regulated by the mouse *Dot1l* 3′UTR containing the WT m⁶A motif, in the context of human IGF2BP3-deleted HEK293T cells (Fig. EV4M–O). Overall, these findings collectively revealed a fundamental mechanism by which IGF2BP3, with the assistance of YBX2, sequesters m⁶A-modified RNAs (e.g., *Dot1l* and *Hdac11*) from polysomes to maintain their moderate translation, which is essential for spermiogenesis and male fertility (Fig. 6M).

### DOT1L and HDAC11 down-regulation rescues sperm developmental defects in *Igf2bp3*-deficient males

Furthermore, we investigated whether the aberrant accumulation of DOT1L and HDAC11 primarily contributes to the pathogenesis of *Igf2bp3*-KO sperm. To this end, siRNAs targeting *Dot1l* or *Hdac11* were micro-injected into *Igf2bp3*$^{-/-}$ testes to specifically knockdown their proteins (Fig. 7A). Compared to the negative control siRNA (siNC), siRNA targeting *Dot1l* (si*Dot1l*-1/2) significantly

reduced DOT1L and H3K79me2 levels in *Igf2bp3*$^{-/-}$ testes; additionally, siRNA targeting *Hdac11* (si*Hdac11*-1/2) effectively downregulated HDAC11 and increased H4K5/8/12ac levels in *Igf2bp3*$^{-/-}$ testes (Fig. EV5A–E). Besides, co-administration of si*Dot1l*-1 and si*Hdac11*-1 also reduced DOT1L, HDAC11, and H3K79me2 levels, and increased the level of H4K5/8/12ac in *Igf2bp3*$^{-/-}$ testes (Figs. 7B–F and EV5F–H). Moreover, the histone retention and protamine reduction in *Igf2bp3*$^{-/-}$ epididymal spermatozoa could partially be rescued via the co-injection of si*Dot1l*-1 and si*Hdac11*-1 (Fig. 7G). Interestingly, computer-aided sperm analysis (CASA) showed that both sole and combined injections of si*Dot1l* and si*Hdac11* significantly improved the motility and concentration of epididymal spermatozoa from *Igf2bp3*$^{-/-}$ testes, verifying the detriment of excessive DOT1L and HDAC11 in sperm developmental defects resulting from IGF2BP3 deletion (Figs. 7H,I and EV5I,J; Movies EV4–6). Most importantly, fertility was restored in 5 out of 16 infertile *Igf2bp3*$^{-/-}$ male mice following si*Dot1l*/*Hdac11*-1 injection within two months, whereas no such recovery was observed in the siNC-treated group (n = 13) (Fig. 7J). Collectively, we have established the IGF2BP3-*Dot1l*/*Hdac11* axis in histone-to-protamine transition and strongly validated its pivotal role in sperm development.

## Discussion

In this study, we identified the predominant expression and function of IGF2BP3 in mouse spermatids, genetic ablation of mouse IGF2BP3 resulted in abnormal morphology, decreased intensity, and impaired motility of spermatozoa. Mechanistic analysis emphasized the translational repression of IGF2BP3-YBX2 complex to m⁶A-modified RNAs essential for chromatin organization and spermatid maturation, including *Dot1l* and *Hdac11*. Their protein over-accumulation contributed to aberrant H3K79me2 and H4K5/8/12ac levels as well as the subsequent defects in histone-to-protamine exchange in IGF2BP3-KO mice. Based on the disclosed mechanism, siRNAs specifically downregulating the protein production of DOT1L and HDAC11 efficiently rescued defects of *Igf2bp3*-null spermatozoa, emphasizing the crucial involvement of IGF2BP3-*Dot1l*/*Hdac11* axis in sperm development and male fertility.

IGF2BP3 has been largely investigated in various cancers, such as promoting cancer progression and indicating a worse prognosis

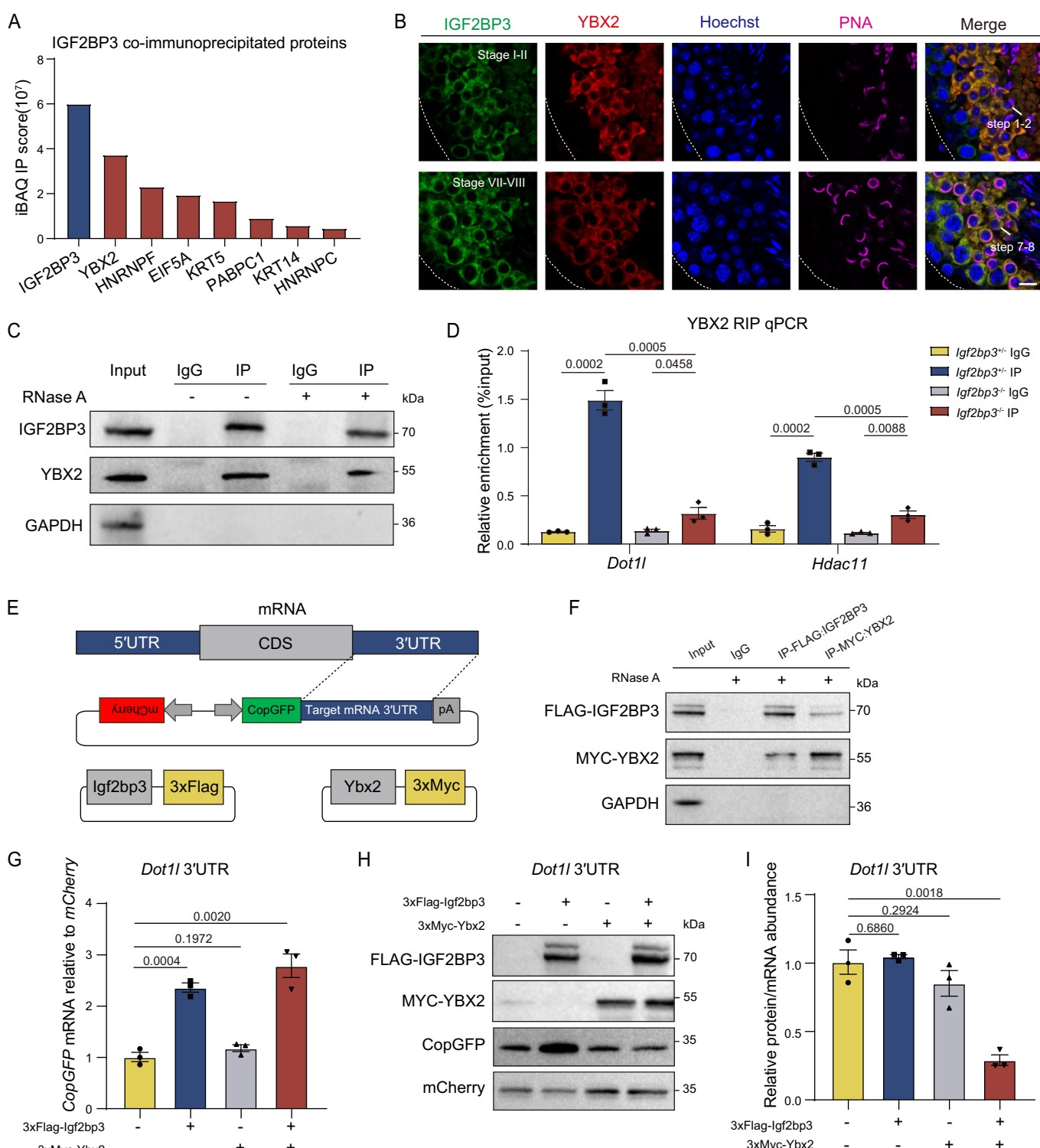

for patients (Ennajdaoui et al, 2016; Tang et al, 2023; Tran et al, 2022). Besides, IGF2BP3 also drives placental development (Li et al, 2014). However, its functions and regulatory role in the reproductive system remained largely unknown (Mancarella and Scotlandi, 2019). In the current study, IGF2BP3 ablation in testis led to sperm developmental defects and the protein overproduction associated with chromatin organization and histone modification, such as DOT1L and HDAC11, which are responsible for H3K79 methylation and histone deacetylation in mammals. Yet the role of HDAC11 in spermatids remains unknown, it is widely recognized

**Figure 5.  IGF2BP3 represses RNA translation via the interacting with YBX2.**

(A) Histogram representations of the top interacting partners of IGF2BP3 in adult WT mouse RS. (B) Immunofluorescence of IGF2BP3 (green), YBX2 (red), and PNA (magenta) in adult testicular paraffin sections from 8-week-old mice. Scale bar, 10 μm. Dotted borders demarcates the basement membrane of the seminiferous tubule. (C) Western blotting analysis of IGF2BP3 and YBX2 in the adult mouse RS lysates (input), and the lysate immunoprecipitation with anti-IgG or IGF2BP3 antibodies treated with RNase A (+) or not (−), respectively. (D) YBX2 RIP-qPCR analyses of *Dot1l* and *Hdac11* in adult *Igf2bp3*$^{+/-}$ and *Igf2bp3*$^{-/-}$ testes. Unpaired two-tailed $t$ test. Error bars, $n = 3$ biological replicates, mean ± SEM. (E) Schematic of bidirectional expression vectors. CopGFP ligates the 3′UTR of IGF2BP3 target gene, and mCherry is used as a transfection control. (F) Western blotting analysis of FLAG-IGF2BP3 and MYC-YBX2 in the Flag-*Igf2bp3* and Myc-*Ybx2* co-transfected HEK293T cell lysates (input), and the lysate immunoprecipitation with anti-IgG, anti-FLAG or anti-MYC antibodies treated with RNase A (+), respectively. (G) qPCR analyses of the relative levels of *CopGFP* mRNAs normalized to *mCherry* mRNAs. Cell lines were treated with 2 μg/ml actinomycin D for 2 h. Unpaired two-tailed $t$ test. Error bars, $n = 3$ biological replicates, mean ± SEM. (H) Western blotting analysis of the protein levels of CopGFP under the regulation of *Dot1l* 3′UTR with the overexpression of FLAG-tagged IGF2BP3 or MYC-tagged YBX2. The level of mCherry is set as the internal control. (I) Histogram showing the ratios of CopGFP proteins (normalized to mCherry proteins) to the *CopGFP* mRNAs (normalized to *mCherry* mRNAs), corresponding to (G, H). Unpaired two-tailed $t$ test. Error bars, $n = 3$ biological replicates, mean ± SEM. Source data are available online for this figure.

that histone deacetylases and acetyltransferases are concurrently expressed during spermiogenesis, indicating the precise dynamics of histone acetylation to open the chromatin structure to facilitate the eviction of histones in sperm maturation (Jiang et al, 2018; Shiota et al, 2018; Shogren-Knaak et al, 2006; Wang et al, 2018). Moreover, although DOT1L-knockdown lead to H3K79me2 loss resulted in failed histones replacement and aberrant protamine recruitment in mouse sperm (Blanco et al, 2023; Lin et al, 2023; Malla et al, 2023), depletion of H3K79me2 at specific locus could promoted the hyperacetylation of lysine on histone 4, thereby facilitating cell fate transition (Wille et al, 2023). Consistently, a significant increase in H4K5/8/12ac was observed in adult *Igf2bp3*$^{-/-}$ testes solely injected with siRNA targeting *Dot1l* (si-*Dot1l*) (Fig. EV5C,E). This suggested that the aberrate elevated DOT1L and accumulated H3K79me2 in IGF2BP3-KO testes may serve as an additional barrier to H4 hyperacetylation, potentially impeding the subsequent histone-to-protamine transition.

Phenotypically, we observed variations in fertility among individual *Igf2bp3*$^{-/-}$ male mice, ranging from sub-fertility to infertility, along with differences in sperm motility and sperm count. The environmental context and redundancy of histone modifications, resulting from the upregulation of DOT1L and HDAC11, likely contributed to these fertility variations. Despite the variability, our experimental design ensured that the molecular mechanism identified accurately reflected the genuine regulatory role of IGF2BP3 in spermatogenesis (see the Methods section). Additionally, a subset of *Igf2bp3*-KO sperms exhibited reduced motility and disorganized microtubules. Interestingly, genes involved in cilium and microtubule movement (such as *Cfap65*, *Dnah2*, and *Dnhd1*) were also targets of IGF2BP3 and translationally upregulated in round spermatids when lacking IGF2BP3 (Fig. 4G). However, it had been demonstrated that most of these genes were transcribed in late spermatocytes to round spermatids and translated at elongating spermatids (Hwang et al, 2021; Tan et al, 2022; Wang et al, 2021b), indicating a decoupling between transcription and translation of these RNAs. It suggested that IGF2BP3 may also play a role in safeguarding target RNAs from degradation and premature translation to guarantee timing translation at transcriptionally inert elongating spermatids. Thereby, IGF2BP3 would orchestrate the RNA stability and translation in round spermatids, ensuring precise dosage and timely synthesis of specific proteins for sperm development.

Mechanistically, we established the regulatory role of IGF2BP3 as an m$^6$A recognition protein during spermiogenesis, a stage where the involvement of m$^6$A binding proteins had not been previously evidenced. Notably, the CA-rich element, which is essential for IGF2BPs binding in previous works, was also identified in the binding motifs of IGF2BP3 in our study (Fig. EV4E) (Biswas et al, 2019; Conway et al, 2016; Hafner et al, 2010; Schneider et al, 2019). Additionally, the potential m$^6$A-induced RNA structural remodeling in facilitating IGF2BP3 binding had yet been distinguished in our study (Sun et al, 2019). Therefore, it remains to be further elucidated whether these mechanisms underlie IGF2BP3's function in mouse testes. Besides, we discovered that the translation repression of IGF2BP3 in testes relied on its cooperative binding with YBX2, an RBP facilitating mRNA storage in RNP and suppressing RNA translation in round spermatids (Kleene, 2016; Yang et al, 2005a). Similarly, IGF2BP3 was also present in the chromatoid bodies (CBs), a specialized form of RNP or P-body in round spermatids (Fig. 1E). Therefore, these may underlie the partitional regulation of IGF2BP3 to specific RNAs in testicular germ cells, which is highly consistent with previous mechanism revealed in HeLa cells, demonstrating that IGF2BP3 can drive hyper-m$^6$A methylated RNAs into P-body to negatively regulate their translation (Shan et al, 2023). In addition, despite the limited binding capacity of YBX2 to target RNAs of IGF2BP3 directly, our study expanded the regulatory repertoire of a proportion of YBX2 in sperm development and provided a comprehensive insight into the underlying mechanism responsible for male infertility resulting from YBX2 deletion (Yang et al, 2005b).

In conclusion, our present findings identified a pivotal role of IGF2BP3 in spermiogenesis and provided insights into the regulatory mechanisms governing chromatin organization and histone-to-protamine transition during sperm development in mice. Together with the elucidated IGF2BP3-YBX2 complex in the translation fine-tuning of m$^6$A-modified RNAs, our study highlighted the importance of a highly orchestrated RNA metabolic program in male germ cells.

# Methods

**Reagents and tools table**

| Reagent/resource | Reference or source | Identifier or catalog number |
|---|---|---|
| **Experimental models** | | |
| C57BL/6 (*M. musculus*) | Southern Medical University | N/A |

| Reagent/resource | Reference or source | Identifier or catalog number |
| --- | --- | --- |
| KM (*M. musculus*) | Southern Medical University | N/A |
| HEK293T cells (*H. sapiens*) | Procell Life Science Technology | CL-0005 |
| GC-2 cells (*M. musculus*) | Procell Life Science Technology | CL-0593 |
| **Recombinant DNA** | | |
| pHIV-EGFP | Addgene | #21373 |
| pCDH-CMV-mCherry-T2A-Puro | Addgene | #72264 |
| pCDH-EF1-copGFP | Addgene | #73030 |
| pHIV-Igf2bp3-3xFLAG-Puro | This study | N/A |
| pHIV-mutIgf2bp3-3xFLAG-Puro | This study | N/A |
| pHIV-Ybx2-3xMYC-Bsd | This study | N/A |
| pCDH-CMV-mCherry-EF1-copGFP | This study | N/A |
| pCDH-Actb-3′UTR-repoter | This study | N/A |
| pCDH-Dot1l-3′UTR-repoter | This study | N/A |
| pCDH-Dot1l-3′UTR-m⁶AWT | This study | N/A |
| pCDH-Dot1l-3′UTR-m⁶AMUT | This study | N/A |
| pCDH-Hdac11-3′UTR-m⁶AWT | This study | N/A |
| pCDH-Hdac11-3′UTR-m⁶AMUT | This study | N/A |
| **Antibodies** | | |
| anti-m6A antibody | Synaptic Systems | 202003 |
| horseradish peroxidase-conjugated secondary antibody | ABCAM | ab6721 |
| m6A antibody | Proteintech | 68055-1-Ig |
| Rabbit monoclonal anti-IGF2BP3 | Abcam | ab177477 |
| Rabbit polyclonal anti-IGF2BP3 | Bethyl Laboratories | A303-426A |
| Rabbit monoclonal anti-Lin28A | Abcam | ab279647 |
| Rabbit monoclonal anti-gamma H2A.X | Abcam | ab81299 |
| Rabbit anti-SOX9 | Abclonal | A19710 |
| Mouse monoclonal anti-GATA4 | Santa Cruz | sc-25310 |
| Rabbit monoclonal anti-IGF2BP1 | Cell Signaling Technology | 14672S |
| Rabbit monoclonal anti-IGF2BP2 | Cell Signaling Technology | 8482S |
| Rabbit monoclonal anti-Histone H2A | Abcam | ab177308 |

| Reagent/resource | Reference or source | Identifier or catalog number |
| --- | --- | --- |
| Mouse monoclonal anti-Histone H2B | Abcam | ab52484 |
| Rabbit polyclonal anti-H3 | Proteintech | 17168-1-AP |
| Mouse monoclonal anti-H4 | Cell Signaling Technology | 2935S |
| Mouse monoclonal anti-protamine P1 | Briar Patch Biosciences | Hup1N |
| Mouse monoclonal anti-protamine P2 | Briar Patch Biosciences | Hup2B |
| Mouse monoclonal anti-Ac-Histone H4 | Santa Cruz | sc-377520 |
| Mouse monoclonal anti-TNP2 | Santa Cruz | sc-393843 |
| Rabbit polyclonal anti-Histone H3(di methyl K79) | Abcam | ab3594 |
| Rabbit monoclonal anti-HDAC11 | Abcam | ab246512 |
| Rabbit monoclonal anti-DOT1L | Abcam | ab239358 |
| Mouse monoclonal anti-YBX2 | Santa Cruz | sc-393840 |
| Rabbit monoclonal anti-MYC tag | Proteintech | 16286-1-AP |
| Mouse monoclonal anti-FLAG tag | Proteintech | 60002-1-Ig |
| Rabbit monoclonal anti-mCherry | Proteintech | 26765-1-AP |
| Rabbit polyclonal anti-CopGFP | Bioss Antibodies | bs-33867R |
| Mouse polyclonal anti-Beta Actin | Proteintech | 20536-1-AP |
| Rabbit monoclonal anti-alpha Tubulin | Abcam | ab176560 |
| Rabbit polyclonal anti-METTL3 | Proteintech | 15073-1-AP |
| Mouse polyclonal anti-m6A | Proteintech | 68055-1-Ig |
| PNA-647 | ThermoFisher | L32460 |
| PNA-488 | ThermoFisher | L21409 |
| **Oligonucleotides and other sequence-based reagents** | | |
| PCR primers | This study | DataSet EV1 |
| shNC | CAACAAGATGAAGAGCACCAA | |
| sh*Mettl3* | GCAAATATGTTCACTATGAAA | |
| siNC | UUCUCCGAACGUGUCACGU | |
| si*Dot1l*-1 | GCUGACCUACAAUGACCUG | |
| si*Dot1l*-2 | CUCGGUUUACACAGCUUCA | |
| si*Hdac11*-1 | AGAGUCGUUUGCUGUUCAU | |
| si*Hdac11*-2 | CCACCAUCAUUGAUCUCGA | |
| **Chemicals, enzymes and other reagents** | | |
| T7 transcription kit | Vazyme | TR101-01 |

| Reagent/resource | Reference or source | Identifier or catalog number |
|---|---|---|
| Cas9 protein | Invitrogen | 91318767 |
| ClonExpress II One Step Cloning Kit | Vazyme | C112-01 |
| PrimeSTAR GXL DNA Polymerase | Takara | R050A |
| RIPA | Solarbio | R0010 |
| peanut agglutinin | PNA, Thermo Fisher | L32460 and L21409 |
| TUNEL BrightGreen Apoptosis Detection Kit | Vazyme | A112-01 |
| acridine orange hydrochloride (AO) staining kit | Solarbio | CA1142 |
| RNase A | Promega | A7973 |
| FastAP | Thermo Scientific | EF0652 |
| T4 PNK | NEB | M0201L |
| T4 RNA ligase | NEB | M0437M |
| proteinase K | NEB | P8107S |
| Superscript III | Thermo Fisher | 18080044 |
| MyONE Silane beads | Thermo Scientific | 37002D |
| Q5 PCR master mix | NEB | M0492L |
| ice-cold IP lysis buffer | Thermo Scientific | 87787 |
| DTT | TargetMol | T5370 |
| Protease inhibitors cocktail | TargetMol | C0001 |
| Recombinant RNasin® Ribonuclease Inhibitor | Promega | N2515 |
| DNase I | NEB | M0303S |
| Cycloheximide | CST | 2112 |
| TRIzol reagent | TIANGEN | Y1809 |
| Hiscript III Reverse Transcriptase | Vazyme | R302-01 |
| AceQ qPCR SYBR Green Master Mix | Vazyme | Q121-02 |
| Streptavidin magnetic beads | TargetMol | C0089 |
| RiboMinus Eukaryote Kit | Thermo | A15020 |
| RNA Clean & Concentrator Kits | Zymo Research | R1017 |
| mRNA Capture Beads | Vazyme | N403-01 |
| His6-IGF2BP3 | Hubei Ipodix Biotechnology | PA1000-8906M |
| His Tag Beads | TargetMol | C0123B |
| rSAP enzyme | NEB | M0371S |
| SuperEnhanced ECL | GBCBIO | G3308 |
| Recombinant RNasin® Ribonuclease Inhibitor | Promega | N2515 |
| IgG control | Proteintech | B900620 |

| Reagent/resource | Reference or source | Identifier or catalog number |
|---|---|---|
| protein G magnetic beads | Invitrogen | 10003D |
| proteinase K | Invitrogen | 25530049 |
| gDNA wiper | Vazyme | R423-01 |
| DMEM medium | GIBCO | 11965118 |
| Fetal bovine serum | VISTECH | SE100-011 |
| **Software** | | |
| cutadapt v4.5 | https://doi.org/10.14806/ej.17.1.200 | |
| Bowtie2 v2.3.5.1 | https://doi.org/10.1038/nmeth.1923 | |
| STAR v2.7.10b | https://doi.org/10.1093/bioinformatics/bts635 | |
| Clipper v2.1.2 | https://github.com/AngusJohnson/Clipper2 | |
| HOMER v4.10.0 | https://doi.org/10.1016/j.molcel.2010.05.004 | |
| BedTools v2.29.2 | https://doi.org/10.1093/bioinformatics/btq033 | |
| Trim Galore v0.6.6 | https://github.com/FelixKrueger/TrimGalore | |
| Hisat2 v2.1.0 | https://doi.org/10.1038/s41587-019-0201-4 | |
| StringTie v2.1.7 | https://doi.org/10.1038/nbt.3122 | |
| R v4.0.3 | https://www.r-project.org/ | |
| DESeq2 v1.22.1 | https://doi.org/10.1186/s13059-014-0550-8 | |
| gProfiler | https://biit.cs.ut.ee/gprofiler/gost | |
| GraphPad Prism 9.0 | http://www.graphpad.com | |
| ImageJ | https://imagej.net | |
| **Other** | | |
| Chemiluminescence detection system | Tanon-5200 | |
| Sperm Motility Analysis System | SMAS; DITECT | |
| FACS | MoFlo XDP, Beckman | |
| Hitachi-H7500 transmission electron microscope | Hitachi | |
| Agilent D1000 | Agilent Technologies | |
| HiSeq4000 | Illumina | |
| DNBSEQ-T7 | MGI Tech | |
| LTQ ORBITRAP Velos mass spectrometer | Thermo Fisher Scientific | |
| Triple-quadrupole mass spectrometer | SCIEX Triple Quad 7500 | |
| Mascot server | Matrix Science Ltd, UK | |

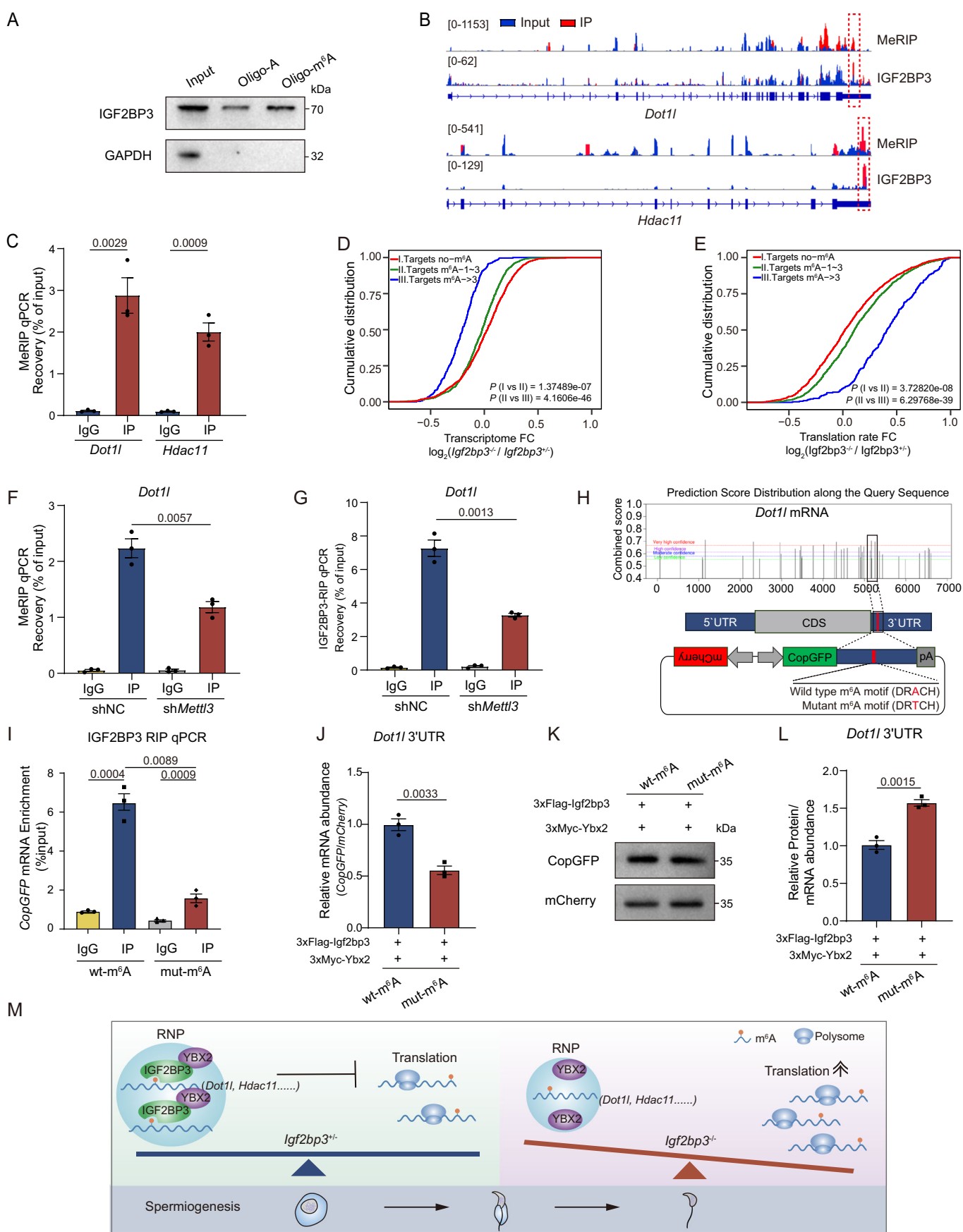

**Figure 6. IGF2BP3 recognition and regulation to m⁶A-modified RNA during spermatogenesis.**

(A) The IGF2BP3 pulldown assay was performed by incubating mouse testicular lysates with biotinylated RNA probes (m⁶A or non-m⁶A). (B) Integrative Genomics Viewer showing the distribution of m⁶A-modified peaks and IGF2BP3-binding peaks in indicated transcripts from round spermatid m⁶A-seq data (Hong S et al, 2022) and mouse testicular IGF2BP3 eCLIP-seq data. Blue and red representing peaks in the input and IP groups, respectively. Red dashed boxes showing the regions containing both m⁶A-modified and IGF2BP3-binding peaks. (C) MeRIP-qPCR analysis of indicated transcripts in round spermatids. Unpaired two-tailed *t* test. Error bars, *n* = 3 biological replicates, mean ± SEM. (D, E) Cumulative distribution of RNA abundance (D) and TE (E) changes between *Igf2bp3*⁻/⁻ and *Igf2bp3*⁺/⁻ round spermatids. Each line showing the IGF2BP3 targets without m⁶A-modified sites (red) and targets with one-three (green) and more than three (blue) m⁶A-modified sites. *P* values were calculated using two-sided Wilcoxon test. (F) MeRIP-qPCR analysis of *Dot1l* mRNAs in GC-2 cells treated with sh*Mettl3* or control shRNA. Unpaired two-tailed *t* test. Error bars, *n* = 3 biological replicates, mean ± SEM. (G) RIP qPCR analysis showing the binding preference of IGF2BP3 to *Dot1l* mRNA in GC-2 cells treated with sh*Mettl3* or control shRNA. Unpaired two-tailed *t* test. Error bars, *n* = 3 biological replicates, mean ± SEM. (H) m⁶A sites within *Dot1l* mRNAs predicted by SRAMP online software (top). Schematic of bidirectional expression vectors (bottom). CopGFP ligates the m⁶A-enriched region of *Dot1l* 3′UTR with the overexpression of FLAG-tagged IGF2BP3, and mCherry is used as a transfection control. (I) RIP-qPCR analysis showing the binding preference of FLAG-IGF2BP3 to *Dot1l* mRNA in HEK293T. Unpaired two-tailed *t* test. Error bars, *n* = 3 biological replicates, mean ± SEM. (J) qPCR analysis of the relative levels of *CopGFP* mRNA under the regulation of the m⁶A motif in the *Dot1l* 3′UTR, normalized to *mCherry* mRNA. Cell lines were treated with 2 μg/ml actinomycin D for 2 h. Unpaired two-tailed *t* test. Error bars, *n* = 3 biological replicates, mean ± SEM. (K) Western blotting analysis of CopGFP protein levels in FLAG-tagged IGF2BP3 and MYC-tagged YBX2 overexpressed HEK293T under the control of the m⁶A motifs in the *Dot1l* 3′UTR. The level of mCherry is set as the internal control. (L) Histogram showing the ratios of CopGFP proteins (normalized to mCherry proteins) to the *CopGFP* mRNAs (normalized to *mCherry* mRNAs), corresponding to Fig. 6J,K. Unpaired two-tailed *t* test. Error bars, *n* = 3 biological replicates, mean ± SEM. (M) Summary of the regulatory mechanism of IGF2BP3 in spermatids development. IGF2BP3-YBX2 complex binds a proportion of m⁶A-modified RNAs to ensure their balance between RNA storage in mRNPs and translation in polysomes. While, the ablation of IGF2BP3 leads to the removal of RNP protection and protein overproduction or pre-translation in RS. Source data are available online for this figure.

## Ethics approval and consent to participate

Animal procedures were performed according to the ethical guidelines of the South Medical University Animal Ethics Committee (L2021162).

## Mice

Mice used in the study were C57BL/6 and KM strains and maintained under the SPF animal facility at Southern Medical University. They were subjected to a 12-h light/dark cycle, with temperature controlled between 20 and 25 °C and humidity maintained at 40–70%. Mice were free to access water and food. The experimental procedures were approved and supported by the Ethics Committee on Use and Care of Animals of Southern Medical University.

*Igf2bp3* knockout mice were generated using CRISPR/Cas9 technology, following the method described previously (Shalem et al, 2014). Briefly, a sgRNA (AATGGCTCCAACAAACTGGG) was transcribed in vitro using the T7 transcription kit, followed by incubation with Cas9 protein. These two components were co-injected simultaneously into C57BL/6 pronuclear zygotes, which were then transferred into the pseudopregnant KM mouse. Offspring genotyping was performed to obtain founder mice (F0) with an *Igf2bp3*-deletion by genomic PCR. To generate *Igf2bp3*⁻/⁻ mice, *Igf2bp3*⁺/⁻ males were mated with *Igf2bp3*⁺/⁻ females, while for expedited production of *Igf2bp3*⁻/⁻ males, sub-fertile *Igf2bp3*⁻/⁻ males were mated with fertile *Igf2bp3*⁻/⁻ females. The off-target effect was excluded by sequencing the PCR-amplified potential off-target sites as predicted by Off-Spotter (Dataset EV1) (Pliatsika and Rigoutsos, 2015). To minimize variability and more accurately reflect the mechanisms by which IGF2BP3 regulates spermatogenesis, *Igf2bp3*⁻/⁻ male mice with impaired fertility were selected for the experiments presented in Figs. 3–5 and associated supplemental figures.

## Plasmids

### Expression vectors

To construct pHIV-Igf2bp3-3xFLAG-Puro and pHIV-Ybx2-3xMYC-Bsd, 3xFLAG-Puro sequences were PCR-amplified using the primer pair pPuro-F/R; 3xMYC-Bsd sequences were PCR-amplified using the primer pair pBsd-F/R. After purification, the PCR products were homologous recombined with the ClonExpress II One Step Cloning Kit, and inserted into BmahI and ClaI sites of pHIV-EGFP. The complementary DNAs (cDNAs) of *Igf2bp3* and *Ybx2* amplifying with PrimeSTAR GXL DNA Polymerase were inserted into the XbaI site of pHIV-3xFLAG-Puro and pHIV-3xMYC-Bsd through homologous recombination kit, respectively.

### Reporter vectors

To construct the pCDH-CMV-mCherry-EF1-copGFP vector, pCMV-F and pmCherry-R primers were applied to amplify the CMV-mCherry sequence with pCDH-CMV-mCherry-T2A-Puro. The PCR product was inserted into ClaI site of pCDH-EF1-copGFP by homologous recombination. To obtain the pCDH-Actb-3′UTR-repoter or pCDH-Dot1l-3′UTR-repoter, the primer pairs pDot1l-3′UTR-F/R and pActb-3′UTR-F/R were utilized for mouse genomic PCR. Then, the PCR products were purified and inserted into SalI sensitive site of pCDH-CMV-mCherry-EF1-copGFP vector. To generate pCDH-Dot1l-3′UTR-m⁶AWT, pCDH-Dot1l-3′UTR-m⁶AMUT, pCDH-Hdac11-3′UTR-m⁶AWT and pCDH-Hdac11-3′UTR-m⁶AMUT, DNA fragments containing wild-type and mutant Dot1l-3′UTR-m⁶A/ Hdac11-3′UTR-m⁶A were synthesized by Ruibiotech and cloned into the SalI site of pCDH-CMV-mCherry-EF1-copGFP.

All plasmids were sequenced to verify the successful construction; the above primers and DNA sequences were listed in Dataset EV1.

## Western blotting analysis

Western blotting analysis was performed as previously described (Wang et al, 2021a). Cells or testis tissues extracts were prepared using cold RIPA for 30 min. Protein lysates were separated by SDS-PAGE and transferred to nitrocellulose membranes. Membranes were blocked with 5% skim milk in TBST (20 mM Tris, 150 mM NaCl, pH 7.6 with 0.05% Tween-20) and incubated with the primary antibody overnight. The next day, membranes were washed with TBST buffer for 5 min and incubated with HRP-conjugated secondary antibodies at room temperature for 1 h.

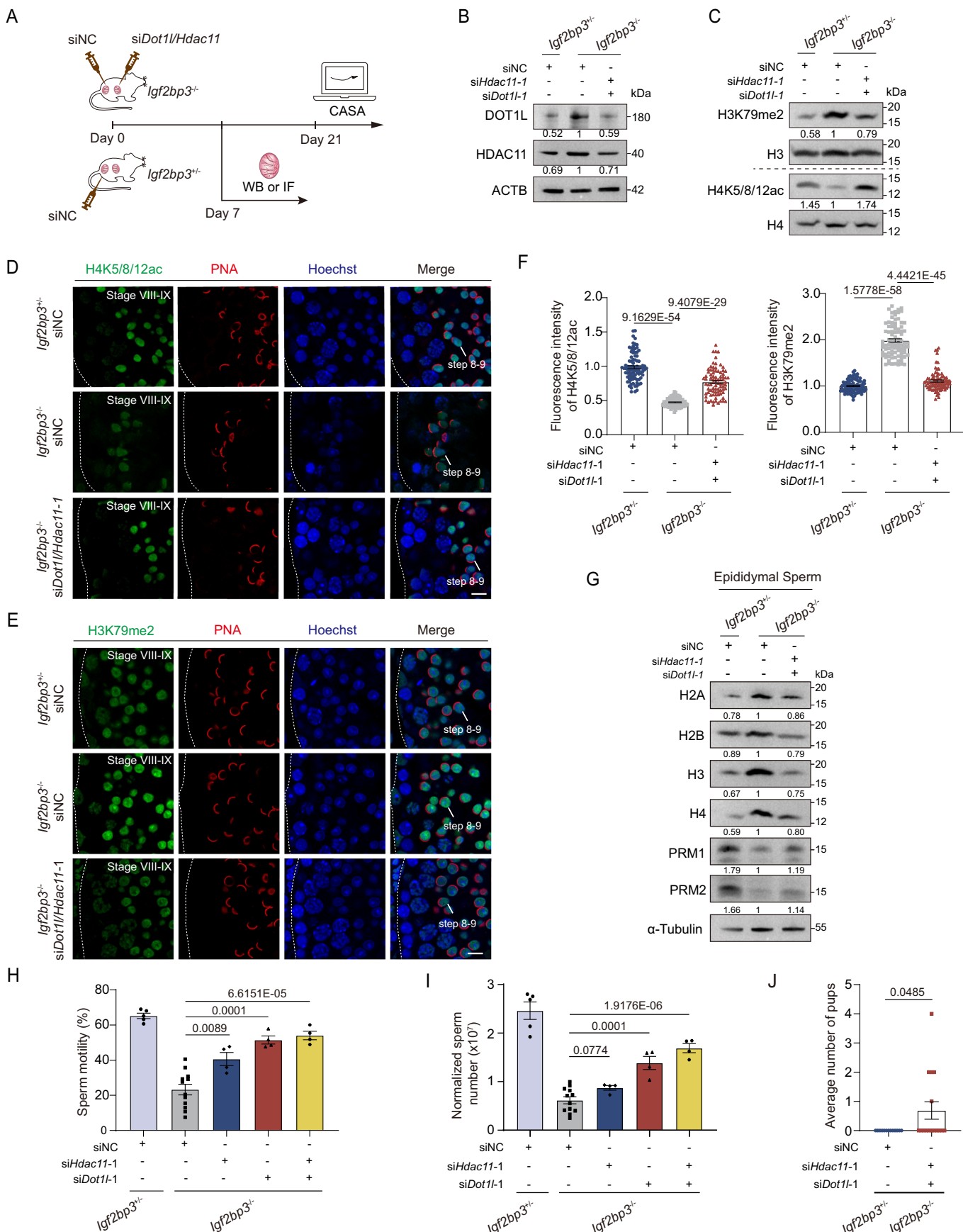

**Figure 7.  Testicular DOT1L and HDAC11 down-regulation rescue the sperm development disorder in IGF2BP3-KO testes.**

(A) Scheme diagram showing the siRNA micro-injection into *Igf2bp3*$^{+/-}$ and *Igf2bp3*$^{-/-}$ mice. (B, C) Western blotting analysis of the protein levels of DOT1L, HDAC11, H3K79me2 and H4K5/8/12ac from adult *Igf2bp3*$^{+/-}$ and *Igf2bp3*$^{-/-}$ mouse testes injected with negative control siRNA (siNC), or siRNAs targeting *Dot1l* and *Hdac11* (si*Dot1l/ Hdac11*-1). The values below each band represent the relative expression levels of each protein. ACTB serves as the internal control of DOT1L and HDAC11, H3 serves as the internal control of H3K79me2, H4 serves as the internal control of H4Ac. (D) Immunofluorescence of H4K5/8/12ac (green) and PNA (red) in paraffin sections of adult *Igf2bp3*$^{+/-}$ and *Igf2bp3*$^{-/-}$ mouse testes injected with siNC or si*Dot1l/Hdac11*-1. Scale bar, 10 μm. (E) Immunofluorescence of H3K79me2 (green) and PNA (red) in paraffin sections of adult *Igf2bp3*$^{+/-}$ and *Igf2bp3*$^{-/-}$ mouse testes injected with siNC or si*Dot1l/Hdac11*-1. Scale bar, 10 μm. (F) Quantification of fluorescence intensity of H4K5/8/12ac (left) and H3K79me2 (right) in paraffin sections of adult *Igf2bp3*$^{+/-}$ and *Igf2bp3*$^{-/-}$ mouse testes injected with siNC or si*Dot1l/Hdac11*-1, corresponding to (D, E). Unpaired two-tailed *t* test. Error bars, n = over 88 cells from three biological replicates, mean ± SEM. (G) Western blotting analysis of the protein levels of H2A, H2B, H3, H4, PRM1, and PRM2 in epididymal spermatozoa from adult *Igf2bp3*$^{+/-}$ and *Igf2bp3*$^{-/-}$ mouse testes injected with siNC or si*Dot1l/Hdac11*-1. The values below each band represent the relative expression levels of each protein, normalized using α-Tubulin as the control. (H, I) CASA of the percentage of motile sperm (H) and the corresponding concentration of epididymal spermatozoa (I) from adult *Igf2bp3*$^{+/-}$ and *Igf2bp3*$^{-/-}$ mouse injected with siNC or si*Dot1l/Hdac11*-1. Unpaired two-tailed *t* test. Each bar represents the mean ± SEM from four biological replicates. (J) Fertility recovery of infertile *Igf2bp3*$^{-/-}$ mouse injected with siNC or si*Dot1l/Hdac11*-1. Unpaired two-tailed *t* test. Each bar represents the mean ± SEM. Dotted borders in (D, E) demarcates the basement membrane of the seminiferous tubule. Source data are available online for this figure.

Signals were visualized by an enhanced chemiluminescence detection system, and analyzed by ImageJ.

## Immunostaining and TUNEL staining

Immunostaining was performed as described previously (Wang et al, 2018). Cells and testicular paraffin sections were blocked with 5% bovine serum albumin (BSA) for 1 h and then incubated with primary antibodies overnight at 4 °C. Subsequently, the samples were washed three times with PBS, followed by incubation with secondary antibodies or/and peanut agglutinin (PNA) for 1 h at room temperature. The nuclei were counterstained with 10 μg/ml Hoechst 33342 for 15 min at room temperature, followed by washing with PBS.

For TUNEL staining, assays were performed using TUNEL BrightGreen Apoptosis Detection Kit following the manufacturer's instructions. Images were captured with a ZEISS LSM880 confocal microscope.

## Fertility evaluation

The fertility evaluation was performed by mating one 8-week-old male with two 8-week-old WT female mice (C57BL/6). *Igf2bp3*$^{+/+}$ and *Igf2bp3*$^{+/-}$ male mice were used as control groups. In total, the litter size of *Igf2bp3*$^{-/-}$ male mice (n = 36), *Igf2bp3*$^{+/-}$ male mice (n = 18), and *Igf2bp3*$^{+/+}$ male mice (n = 18) mating with WT females over six months were recorded, respectively.

## Sperm counts and motility analysis

Male mice were sacrificed by cervical dislocation. Spermatozoa were removed from cauda epididymis, washed and incubated in DMEM medium for 30 min. Sperm motility and other parameters were quantitatively analyzed by an automated Sperm Motility Analysis System.

## Immunostaining of epididymis spermatozoa

Cauda epididymis was collected from one euthanized male mouse, several incisions were made and placed in 1 ml of PBS. Spermatozoa were centrifuged and fixed with 4% PFA, then smeared on a glass slide and dried in air. The slides were washed with PBS and stored at 4 °C. On the second day, the slides were permeabilized with 0.3% TritonX-100 in PBS three times, blocked with 5% BSA for 1 h, and

then incubated with primary antibodies overnight at 4 °C. The following steps were performed as the immunostaining of cells and testicular paraffin sections described previously.

## FACS analysis of round spermatids and epididymis spermatozoa

Round spermatids were isolated from adult *Igf2bp3*$^{+/-}$ and *Igf2bp3*$^{-/-}$ testes by FACS as described previously (Lima et al, 2016; Wang et al, 2021a). The analysis to cauda epididymal spermatozoa was performed by FACS following previous protocol (Evenson, 2013, 2022). Briefly, spermatozoa were isolated from the cauda epididymis, washed, and incubated in DMEM medium for 30 min. After centrifugation, the sperm pellets were resuspended in 1 ml of 75% alcohol and incubated overnight at 4 °C. Subsequently, an acridine orange hydrochloride (AO) staining kit was utilized for sperm staining. Following a wash with AO Stain Buffer (1x), the sperm density was adjusted to $10^6$/ml using AO Stain Buffer (1×). The stained sperm suspension was then filtered through 40-μm nylon mesh followed by flow cytometry analysis after a dark incubation at room temperature for 15 min. Damaged sperm with single-stranded DNA and normal sperm with double-stranded DNA were distinguished using FACS based on 647-APC fluorescence and 488-FITC fluorescence, respectively.

## Spermatozoa scan by transmission electron microscopy (TEM)

Cauda epididymis obtained from adult mice were dissected into 1 to 2 mm$^3$ fragments and fixed in a solution of 2.5% glutaraldehyde within the cacodylate buffer (0.1 M, pH 7.4) at 4 °C overnight. Following fixation, the samples were embedded in Eponate 812 and sectioned into ultrathin slices, following with uranyl acetate and lead citrate staining. Epididymal sections were visualized by using a Hitachi-H7500 transmission electron microscope.

## Immunostaining of spermatocyte chromosome spreads

Spermatocyte spreads of the testis samples were prepared as described previously (Peters et al, 1997). Briefly, the testes were dissected and the seminiferous tubules were washed with PBS. Then, the tubules were immersed in hypotonic extraction buffer (30 mM Tris, 50 mM sucrose, 17 mM trisodium citrate dihydrate, 5 mM EDTA, 2.5 mM DTT and 1 mM PMSF) for 30-60 min. The

tubules were then collapsed in 0.1 M sucrose (pH 8.2) before being transferred onto a clean glass slide and subjected to repeated pipetting until obtaining a homogeneous suspension. The cell suspension was then deposited onto slides coated with 4% PFA and 0.3% TritonX-100 (pH 9.2) and oven dried for 2 h before twice washed with 0.4% Photo-Flo 200. Finally, the slides were air-dried at room temperature before proceeding with immunofluorescence staining.

## eCLIP

eCLIP of mouse testes was performed following previous protocol with some modifications (Van Nostrand et al, 2016; Xu et al, 2019).

1. Two testes from adult C57BL/6 mice were gently triturated using a loose glass pestle to ensure mild mechanical digestion. The testis suspension was crosslinked three times with 400 mJ/cm$^2$ UV at 254 nm on ice.
2. Total RNAs were fragmented by 4 ng/µl RNase A to achieve the Input sample (2% of the lysate) was retained, and the protein-RNA complexes were immunoprecipitated with the IGF2BP3 antibody (Abcam) to obtain the IP samples, followed by thoroughly washing.
3. RNA dephosphorylation was performed using FastAP and T4 PNK, followed by ligation of a barcoded RNA adapter to the 3' end using T4 RNA ligase. The RNA-protein complexes were resolved on a gel and transferred onto a nitrocellulose membrane.
4. The region spanning from 70 kDa to 150 kDa containing RNA was digested with proteinase K, followed by RNA reverse transcription using Superscript III. Purification of the samples was carried out using MyONE Silane beads, and qPCR was performed to determine optimal PCR cycle number.
5. Library amplification was conducted using Q5 PCR master mix, then the quality control was performed with an Agilent D1000 following with sequencing on the HiSeq4000.

## RIP qPCR

The procedure was modified from the previous study (Wu et al, 2019). Two testes from adult C57BL/6 mice were dissected and homogenized in 2 ml ice-cold IP lysis buffer, 0.5 mM 1,4-dithiothreitol (DTT), 1× protease inhibitors cocktail, 100 U/ml Recombinant RNasin® Ribonuclease Inhibitor. Supernatants of the lysate were collected, one supplemented with 4 µg IgG antibody and 50 µl beads, another one supplemented with 5 µg IP antibody and 50 µl beads, both were incubated overnight at 4 °C with rotation. Three times washing with wash buffer (IP lysis buffer; 0.5 mM DTT; 1 mM PMSF), followed by the incubation in DNase buffer containing DNase I for 10 min at 37 °C. After twice washing, co-immunoprecipitated RNAs were recovered by TRIzol method and reverse transcribed for subsequent qPCR using SYBR Green master mix. The IP and IgG groups were normalized to the level of *Gapdh*. Primers we used were listed in Dataset EV1.

## RNA-seq and RNC-seq of round spermatids

The multi-ribosomal components of round spermatids were separated using a continuous sucrose gradient ultracentrifugation, and then subjected to RNA-seq and RNC-seq as previously described (Huang et al, 2022; Tang et al, 2020; Zhang et al, 2017).

1. Cells were treated with 100 µg/ml cycloheximide to block translational elongation during round spermatids isolation by FACS. Approximately 2 million round spermatids were isolated from each mouse, and then dissolved in the lysis buffer (15 mM Tris-HCl pH 7.5, 200 mM NaCl, 15 mM MgCl$_2$, 1% TritonX-100 (v/v), 100 µg/ml CHX, 1× proteinase inhibitor cocktail, 40 U/ml RNasin) and incubated on ice for 8 min. The supernatant was collected after centrifugation.
2. 10% of the extract was designated as input control for RNA-seq, the remaining extract was fractionated into 22 portions in a 5% to 50% sucrose gradient by centrifuging at 36,000 rpm for 2.5 h at 4 °C. RNAs associated with polysomes (fractions 16–22) were isolated for RNC-seq.
3. The RNAs were purified through poly(A) selection and subsequently utilized for full-length library construction using the STRT-seq method without 3' terminal cDNAs enrichment (Zhao et al, 2021). The libraries were sequenced on a DNBSEQ-T7 instrument.

## Sequencing data analysis

For eCLIP-seq data:

eCLIP reads were processed and quality control (QC) was performed according to the Encyclopedia of DNA Elements (ENCODE) data processing protocol for eCLIP reads as previously described (https://www.encodeproject.org/eclip/) (Van Nostrand et al, 2016).

1. First, reads were demultiplexed according to their inline barcodes using a custom script, which also modifies each read name to include the read unique molecular identifier (UMI) (https://github.com/YeoLab/eclipdemux/blob/master/eclipdemux_package/demux.py).
2. Next, reads were trimmed using cutadapt (v4.5) and reads from rRNA were removed using Bowtie2 (v2.3.5.1).
3. Surviving reads were then mapped with STAR (v2.7.10b), to mm10 assembly to obtain genome alignments.
4. PCR duplicate removal was then performed with a custom script based on UMI sequences placed inside each read name (https://github.com/YeoLab/eCLIP/blob/master/bin/barcodecollapsepe.py).
5. Read 2 for each IP and size-matched INPUT (SMInput) BAM file was used to call enriched peak clusters with Clipper (v2.1.2). These clusters were then normalized against SMInput reads. Regions passing a $-\log_{10}(P)$ significance of at least 2 and a $\log_2$(fold change) cutoff of 2 were deemed as significantly IGF2BP3-bound for each replicate.
6. Peak annotation was performed using HOMER (V4.10.0) annotatePeaks.pl script. Motif analysis was performed using HOMER findMotifsGenome.pl script on overlapping peaks from two replicates.
7. BedTools (v2.29.2) was utilized to quantify read counts across annotated gene regions, including coding exons (CDS), introns, and 3'UTRs. Least-squares regression was subsequently applied to assess inter-sample correlations. Only regions with at least 10

reads in one of IP or SMInput were considered, and significance was determined by Yates' Chi-Square test or Fisher's Exact test.

For RNA-seq and RNC-seq data:

1. Trim Galore (v0.6.6) was used to trim the 3' adapter in the raw reads. Low-quality reads with Phred quality score <25 were removed.
2. Next, the trimmed reads were aligned to the mouse reference genome assembly build GRCm38 (mm10) using the Hisat2 (v2.1.0).
3. Mapped reads were then assembled into transcripts under the guidance of reference annotation with StringTie (v2.1.7). The gene expression level was normalized by using fragments per kilobase of transcript per million mapped reads (FPKM) method.
4. Differential expression analysis was performed within R (v4.0.3) using DESeq2 (v1.22.1), applying a dual threshold on the $|\log_2$ (fold change)$| > 1.5$ and corresponding statistical significance ($P$ value $< 0.05$). Translational efficiencies (TE) were calculated as: TE = (FPKM in RNC-seq) / (FPKM in RNA-seq). Genes with FPKM > 0.1 were kept for downstream analysis.
5. Gene Ontology (GO) term enrichment analysis for biological processes was performed using gProfiler.

## Quantitative real-time PCR (qPCR)

Total RNA was extracted using the TRIzol reagent. Reverse transcription was performed by Hiscript III Reverse Transcriptase. qPCR was conducted utilizing the qPCR SYBR Green Master Mix and Bio-Rad CFX96 Real-Time System (Bio-Rad CFX Manager 3.1 software). Relative mRNA expression was quantified employing the Delta-Delta CT method and normalized to *β-Actin*. Primers in qPCR assays were listed in Dataset EV1.

## IGF2BP3 IP mass spectrometry of round spermatids

The experimental procedure was adapted from the previous study (Tan et al, 2023b). Adult mouse testes were enzymatically digested, followed by the round spermatids collection using flow cytometry. The collected RS were subsequently lysed with IP lysis buffer and divided into two portions, each portion was then incubated with either IgG or IGF2BP3 antibody-conjugated beads at 4 °C overnight. The immunoprecipitated proteins were digested with trypsin at 37 °C overnight. Peptides were sequentially extracted with 5% formic acid/50% acetonitrile and 0.1% formic acid/75% acetonitrile, followed by the analytical capillary column isolation and spraying into an LTQ ORBITRAP Velos mass spectrometer equipped with a nano-ESI ion source. Identified peptides were annotated against the IPI (International Protein Index) mouse protein database on the Mascot server.

## RNA binding protein pulldown

RNA binding protein pulldown was performed as previously describe with some modifications (Wang et al, 2014; Wu et al, 2019). Biotin-labeled RNA oligonucleotides containing adenosine or $m^6A$ were synthesized by Ruibiotech. An adult mouse testis was homogenized resuspended in 1 mL of lysis buffer on ice, followed

by centrifugation. To the resulting lysate, 1 µg of each RNA oligonucleotide was added, and the lysate, pre-washed with streptavidin magnetic beads, was incubated overnight at 4 °C. Subsequently, 200 µL of pre-washed streptavidin beads were added, and the mixture was incubated for 4 h at 4 °C. The beads were washed five times with wash buffer, then boiled in loading buffer for subsequent western blot. The RNA oligonucleotides sequences were listed in Dataset EV1.

## In vivo RIP LC-MS/MS

The testicular lysis was performed as previously described for RIP qPCR. 50 µL of the supernatant was retained as the Input. The remaining supernatant was incubated with 10 µg of IGF2BP3 antibody-protein A/G beads at 4 °C overnight by rotation. After incubation, 50 µl of supernatant was collected and saved as FT. The beads were washed five times with lysis buffer and saved as the IP fraction. TRIzol was added to all three fractions (Input, FT, and IP) to purify the RNA. Three portions were further purified by rRNA depletion using the RiboMinus Eukaryote Kit with size selection of RNA greater than 200 nt using RNA Clean & Concentrator Kit. The $m^6A$ content in each of the Input, FT, and IP samples was quantified by LC-MS/MS.

## In vitro RIP LC-MS/MS

mRNA was purified from mouse testes by using the mRNA Capture Beads, with 0.2 µg retained as the Input. The purified mRNA (0.8 µg) was incubated with His6-IGF2BP3 (final concentration 500 nM) at 4 °C for 2 h to form the RBP complex. Next, 100 µl of His Tag Beads were washed and added to the solution, followed by rotation at 4 °C for 2 h. The beads were washed, and 500 µl of TRIzol was added to extract the RNA, saved as the IP fraction. The $m^6A$ content in both the Input and IP samples was quantified by LC-MS/MS.

## Quantitative analysis of $m^6A$ by LC–MS/MS

LC-MS/MS was performed as previously described with some modifications (Kang et al, 2024). 50 ng of non-ribosomal RNA or mRNA from mouse testes was digested with 1 U of nuclease P1 and 1 U of rSAP enzyme. The digested samples were then introduced into an LC-MS/MS system. Nucleosides were detected using a triple-quadrupole mass spectrometer in positive ion multiple reaction-monitoring (MRM) mode. Nucleosides were quantified based on retention time and ion mass transitions (268–136 for A; 282–150 for $m^6A$). The nucleoside levels were quantified by comparison with a standard curve generated from pure nucleoside standards prepared from the same batch of samples. The $m^6A$ content was calculated as the ratio of the corrected A value to the calibration concentration.

## $m^6A$ dot blot

The indicated amount of total RNA or synthesized RNA oligonucleotide was denatured in a threefold volume of RNA incubation buffer (65.7% formamide, 7.77% formaldehyde, and $1.33 \times$ MOPS) at 65 °C for 5 min, followed by chilling on ice and mixing with onefold volume of $20 \times$ Saline Sodium Citrate (SSC)

buffer. The RNAs were loaded onto a nitrocellulose filter membrane with Bio-Dot Apparatus. After UV crosslinking, the membrane was stained with 0.02% methylene blue in 0.3 M sodium acetate. Subsequently, the membrane was washed with PBST and blocked with 5% non-fat milk in PBST before incubating with anti-m6A antibody overnight at 4 °C. After incubating with horseradish peroxidase-conjugated secondary antibody, the membrane was visualized using SuperEnhanced ECL.

### MeRIP qPCR

MeRIP–seq was performed as described previously, with minor modifications (Tan et al, 2023b; Xiao et al, 2019). In brief, the total RNAs from GC-2 cells or round spermatids were fragmented and purified by using zinc acetate and TRIzol, separately. RNAs were then incubated overnight at 4 °C in IPP buffer containing Recombinant RNasin Ribonuclease Inhibitor with IgG control or m6A antibody, followed by incubation with protein G magnetic beads and three times washing with IPP buffer as well as the proteinase K digestion. Bound RNAs were extracted and reverse transcribed via HiScript IV RT SuperMix containing gDNA wiper. Recovery rate of specific transcript was calculated by comparing the relative mRNA abundance in the IP group to the input group. Normalization was performed using *Gapdh* for both the IP and IgG groups. Primers used are listed in Dataset EV1.

### Cell culture and transfection

HEK293T and GC-2 cell line was cultured in DMEM medium supplemented with 10% fetal bovine serum. All cells were cultured in 37 °C humidified incubators with 5% $CO_2$ and routinely tested for mycoplasma contamination. IGF2BP3-KO cells were generated by utilizing the CRISPR-Cas9 system, and plasmids were transfected into cells with polyethylenimine (PEI).

### Microinjection of siRNA into mouse testes

We performed the siRNA transfection as previously described (Tan et al, 2023a). Briefly, 4-week-old mice were anesthetized by pentobarbital sodium, and the testes were exteriorized through abdominal incisions. One testis was injected with 4 µl dilutions of siRNA (20 µM) targeting *Dot1l* or *Hdac11* chemically modified with 2-methoxyethyl and cholesterol, while another testis from the same mouse received 4 µl negative control siRNA dilutions at the same concentration as control. After recovery, the mice were maintained on their regular diet. Seven days after the siRNA injection, testes were harvested for the subsequent experiments. Twenty-one days after the injection, epididymides were harvested for subsequent CASA testing.

### Statistical analysis

GraphPad Prism 5 was used to analyze the data of qPCR, western blotting, immunostaining, TEM, FACS, RIP LC-MS/MS, and EMSA. Statistical differences were assessed using Student's *t* test or ordinary one-way ANOVA test. The standard error of the mean (SEM) was employed to evaluate the biological significance. *P* value < 0.05 was considered statistically significant.

## Data availability

The eCLIP-seq, RNA-seq and RNC-seq data are deposited in the Gene Expression Omnibus (GEO) under the accession number GSE256055. The raw data of eCLIP-seq has also been deposited in the National Genomics Data Center of the China National Center for Bioinformation database and can be accessed by reviewers via the private link below: https://ngdc.cncb.ac.cn/gsa/s/Y30kRQCZ. All other data supporting the findings of this study are available from the Lead Contact upon reasonable request.

The source data of this paper are collected in the following database record: biostudies:S-SCDT-10_1038-S44318-025-00659-y.

## Peer review information

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

## Acknowledgements

This work was supported by grants from the National Key R&D Program of
China (2022YFA0806303 to YZ, 2024YFA1802602 to MW,
2022YFA1106200 to JZ), the National Natural Science Foundation of China
(32170866 and 32370840 to MW, 32370911 and 32170869 to GC,
U22A20278 to YZ), the project of Guangzhou Science and Technology
Program (SL2023A04J02344 to MW), the Natural Science Foundation of
Shenzhen (JCYJ20210324120212033 and JCYJ20230808105421043 to GC),
Shenzhen Medical Research Fund (B2402013 to GC). We thank Professors
Xiao-Yang Zhao and Yawei Gao for giving advices to this project, Dr. Jia Yu and
Fang Wang for providing sh*Mettl3* plasmids and eCLIP-seq experiment. We
also thank Beckman for supporting the FACS.

## Author contributions

**Dazhuang Wang**: Investigation; Writing—original draft; Writing—review and
editing. **Zhenyi Huang**: Resources; Software; Investigation; Writing—original
draft; Writing—review and editing. **Yichun Zhou**: Investigation; Writing—
original draft. **Peiyan Chen**: Investigation; Writing—original draft. **Gang Chang**:
Funding acquisition; Investigation; Writing—original draft. **Liwei Ke**:
Investigation. **Congying Jing**: Investigation. **Haojie Yang**: Investigation. **Jiexiang
Zhao**: Funding acquisition; Investigation. **Shaofang Ren**: Investigation. **Yi
Zheng**: Funding acquisition; Investigation. **Yuhan Chen**: Investigation. **Yufan
Xiang**: Investigation. **Jun Liu**: Investigation. **Mei Wang**: Data curation;
Supervision; Funding acquisition; Writing—original draft; Project
administration; Writing—review and editing.

Source data underlying figure panels in this paper may have individual
authorship assigned. Where available, figure panel/source data authorship is
listed in the following database record: biostudies:S-SCDT-10_1038-S44318-
025-00659-y.

## Disclosure and competing interests statement

The authors declare no competing interests.

# Expanded View Figures

**Figure EV1. The strategy of *Igf2bp3*-KO mice and its spermatids loss in testis, related to Fig. 2.**

(A) Schematic structures showing RNA-binding domains within IGF2BP3 proteins (top). Schematic diagram of CRISPR/Cas9 for generating the *Igf2bp3* knockout mice (middle). Sanger sequencing of the targeting locus of *Igf2bp3* in wild-type and *Igf2bp3*$^{-/-}$ mice (bottom). (B) Immunofluorescence of IGF2BP3 (top) and secondary antibody (bottom) in adult testicular sections from 8-week-old *Igf2bp3*$^{+/-}$ and *Igf2bp3*$^{-/-}$ mice. Scale bar, 20 μm. (C) Western blotting analysis of the protein levels of IGF2BP1, IGF2BP2, and IGF2BP3 in *Igf2bp3*$^{+/+}$, *Igf2bp3*$^{+/-}$, and *Igf2bp3*$^{-/-}$ testes. Both the N-terminal (ab177477) and C-terminal (A303-426A) recognizing antibodies of IGF2BP3 were utilized in this assay. ACTB serves as a loading control. (D, E) Computer-aided sperm analysis (CASA) of the sperm motility (D) and sperm number (E) in *Igf2bp3*$^{+/+}$ ($n = 16$), *Igf2bp3*$^{+/-}$ ($n = 18$), and *Igf2bp3*$^{-/-}$ ($n = 39$) mice. *P* values are calculated by one-way ANOVA. Each bar represents the mean ± SEM from biological replicates. (F) Quantitative comparison of abnormal nuclei per tubule between adult *Igf2bp3*$^{+/-}$ and *Igf2bp3*$^{-/-}$ testes. At least 20 tubules of each mouse were calculated. Unpaired two-tailed *t* test. Error bars, $n = 4$ biological replicates, mean ± SEM. (G) PAS staining of the adult testicular sections from 8-week-old *Igf2bp3*$^{+/+}$, *Igf2bp3*$^{+/-}$ and *Igf2bp3*$^{-/-}$ mice. Arrowheads indicate cells with abnormal agglutinated nuclei (left). Scale bar, 40 μm. Lumen area statistics on PAS staining of paraffin sections of *Igf2bp3*$^{+/+}$, *Igf2bp3*$^{+/-}$ and *Igf2bp3*$^{-/-}$ adult testes (right). *P* values are calculated by one-way ANOVA. Error bars, $n = 3$ biological replicates, mean ± SEM. (H) Flow cytometry analysis (left) and bar plot (right) showing the distribution of each ploidy population within adult *Igf2bp3*$^{+/-}$ and *Igf2bp3*$^{-/-}$ testes. Unpaired two-tailed Student's *t* test. Each error bar represents the mean ± SEM from 5 biological replicates.

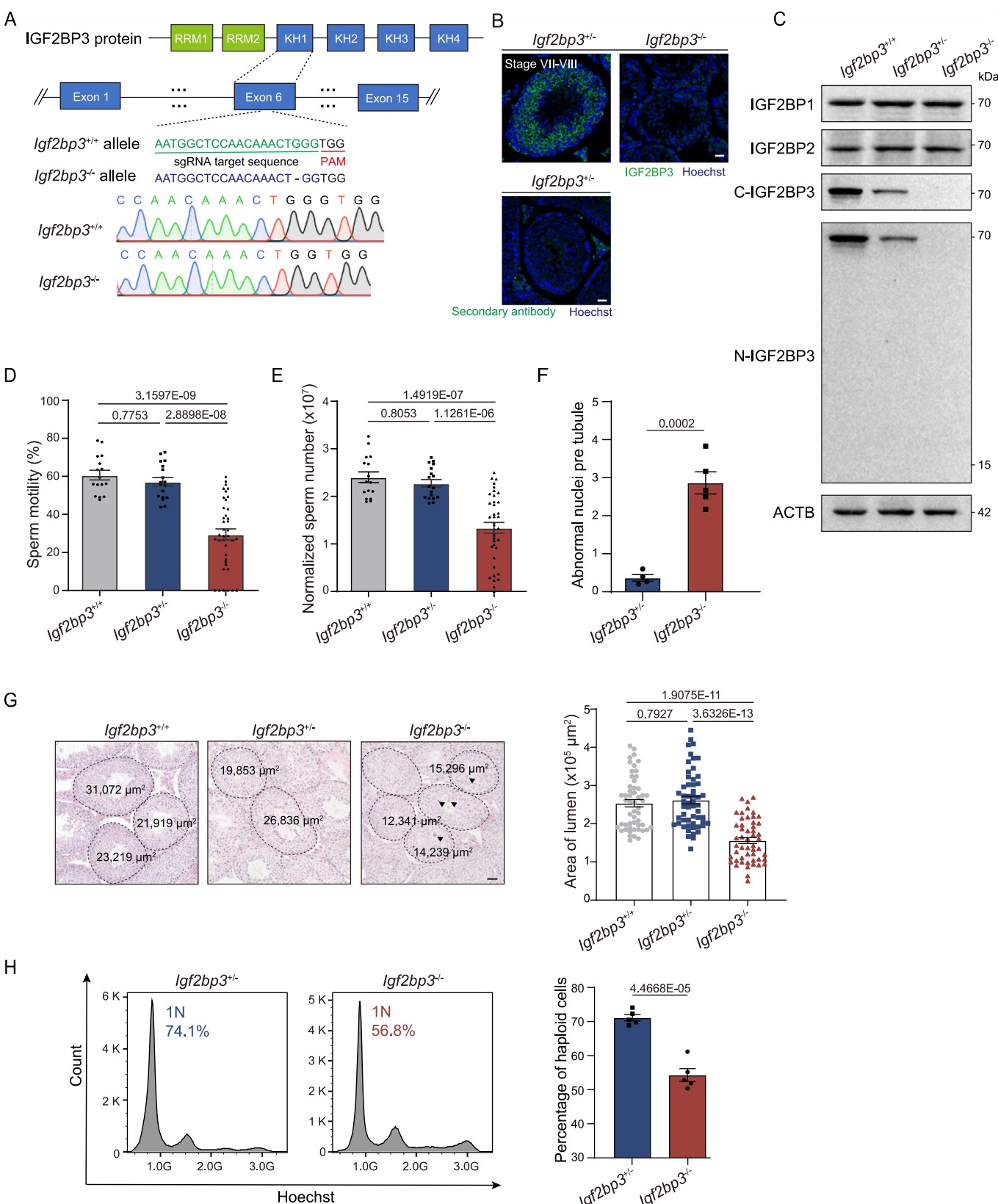

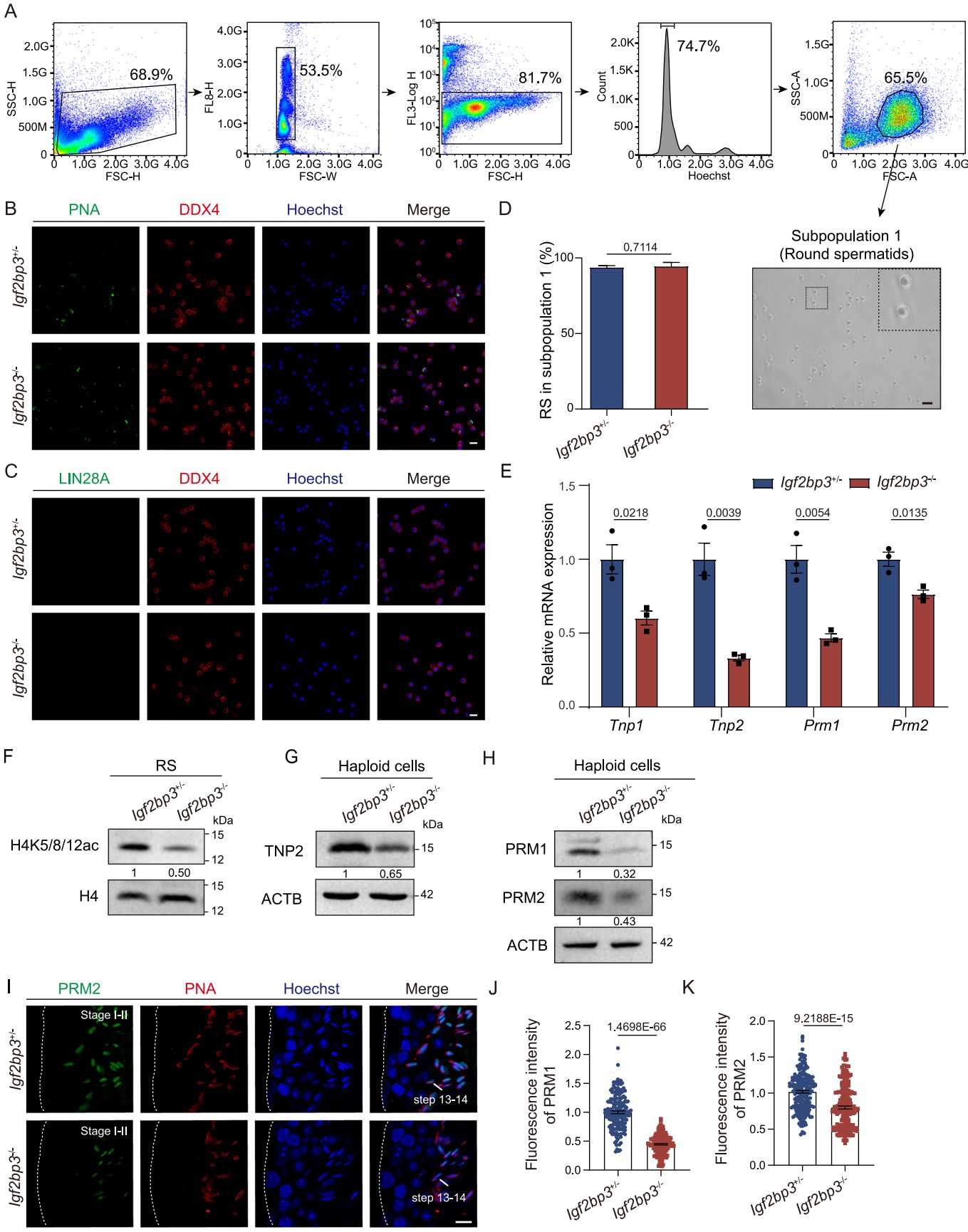

◄

**Figure EV2. The down-regulation of transition proteins and protamines in *Igf2bp3*-KO spermatids, related to Fig. 3.**

(A) Representative FACS strategy for isolating round spermatids from *Igf2bp3*$^{+/-}$ and *Igf2bp3*$^{-/-}$ testes. Isolated round spermatids showing in the bright field image. Scale bar, 80 μm. (B, C) Immunostaining of PNA or LIN28A (green), and DDX4 (red) in round spermatids sorted from *Igf2bp3*$^{+/-}$ and *Igf2bp3*$^{-/-}$ testes, respectively. Scale bar, 20 μm. (D) Percentage of round spermatids in Subpopulation 1 sorted from adult *Igf2bp3*$^{+/-}$ and *Igf2bp3*$^{-/-}$ testes. Unpaired two-tailed *t* test. Error bars, *n* = 3 biological replicates, mean ± SEM. (E) qPCR analysis of the relative expression levels of *Tnp1*, *Tnp2*, *Prm1*, and *Prm2* normalized to *β-Actin* in *Igf2bp3*$^{+/-}$ and *Igf2bp3*$^{-/-}$ RS. Unpaired two-tailed *t* test. Error bars, *n* = 3 biological replicates, mean ± SEM. (F) Western blotting analysis of the protein levels of H4K5/8/12ac in *Igf2bp3*$^{+/-}$ and *Igf2bp3*$^{-/-}$ RS. The values below each band represent the relative expression levels of each protein, normalized using H4 as a loading control. (G, H) Western blotting analysis of the protein levels of TNP2 (G), PRM1 and PRM2 (H) in haploid cells from adult *Igf2bp3*$^{+/-}$ and *Igf2bp3*$^{-/-}$ testes. The values below each band represent the relative expression levels of each protein, normalized using ACTB as a loading control. (I) Immunofluorescence of PRM2 (green) and PNA (red) in adult testicular paraffin sections from 8-week-old *Igf2bp3*$^{+/-}$ and *Igf2bp3*$^{-/-}$ mice. Scale bar, 10 μm. Dotted borders demarcates the basement membrane of the seminiferous tubule. (J, K) Quantification of fluorescence intensity of PRM1 (Fig. 3G) and PRM2 (Fig. EV2I) in paraffin sections of adult *Igf2bp3*$^{+/-}$ and *Igf2bp3*$^{-/-}$ mouse testes. Unpaired two-tailed *t* test. Error bars, *n* = over 150 cells from 3 biological replicates, mean ± SEM.

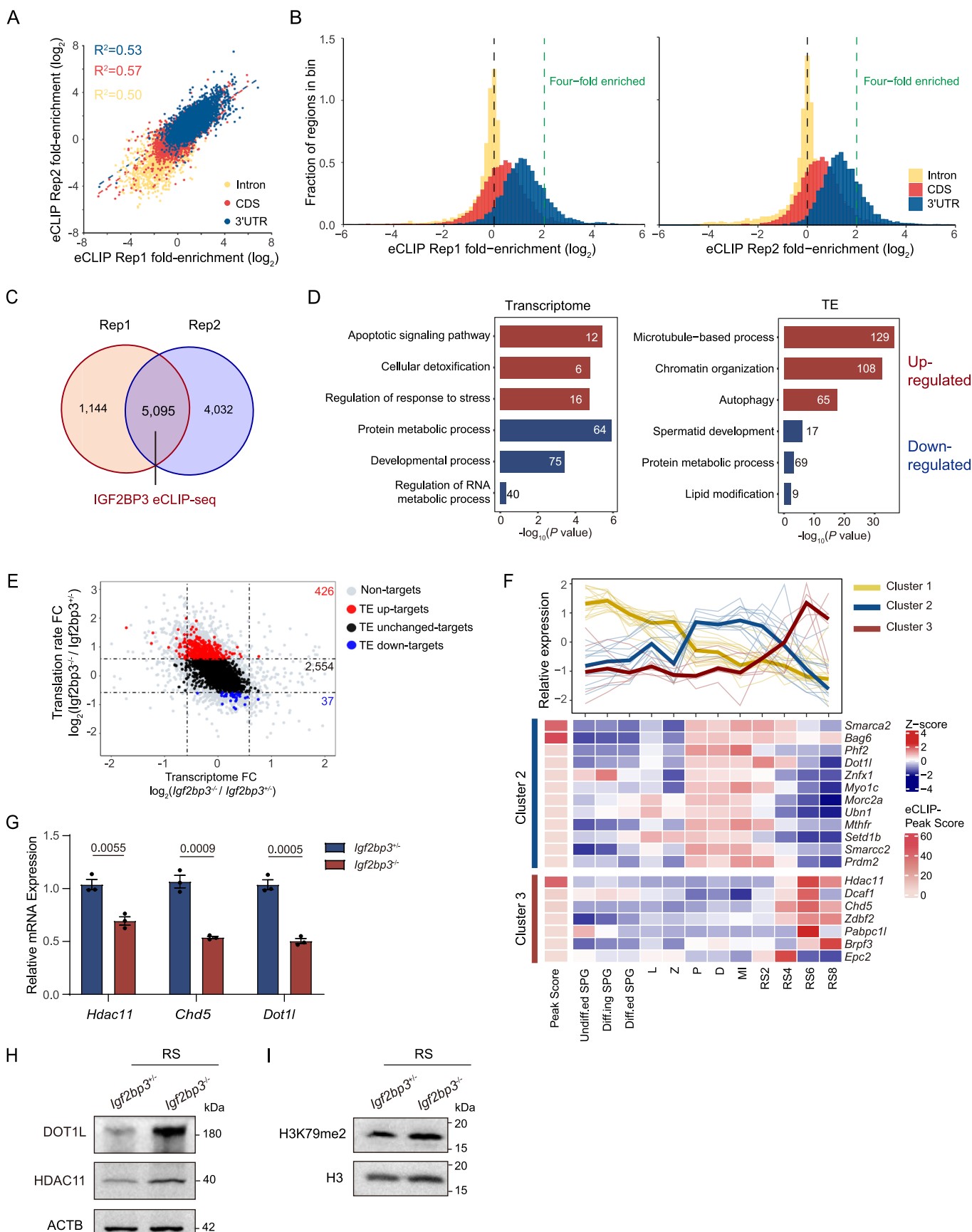

◀ **Figure EV3. Testicular eCLIP-seq of IGF2BP3 and translation efficiency changes between *Igf2bp3*<sup>+/−</sup> and *Igf2bp3*<sup>−/−</sup> round spermatids, related to Fig. 4.**

(A) eCLIP fold enrichment comparison across replicates and samples. Fold enrichment of IGF2BP3 IP1 over paired SMInput1 (*x* axis) versus IGF2BP3 IP2 over paired SMInput2 (*y* axis). Dotted lines and $R^2$ values indicate least-squares regression performed separately for each region type. CDS, coding sequences; UTR, untranslated regions. Rep, replicates. (B) Histogram of region-based fold enrichment for IGF2BP3 (each compared to its paired SMInput). Left, Rep1; right, Rep2. (C) Venn diagram showing the 5,095 overlapping target genes from two IGF2BP3 eCLIP-seq biological replicates. (D) Top GO terms in biological process categories of down- and upregulated genes at transcriptional-level (left) and TE-level (right). Fisher's exact test with g:Profiler was used in GO enrichment analysis. (E) Scatter plot showing mRNA-level changes (*x* axis) against translational efficiencies (TE) changes (*y* axis) between adult *Igf2bp3*<sup>−/−</sup> and *Igf2bp3*<sup>+/−</sup> round spermatids. Red dots indicate TE up-targets. Blue dots indicate TE down-targets. Black dots indicate TE unchanged-targets. Gray dots indicate non-targets of IGF2BP3. (F) Clustering analysis of 37 TE upregulated targets associated with 'chromatin organization' related to Fig. 4G (top). Heatmap showing the genes in cluster 2 and cluster 3 (bottom). (G) qPCR analysis of the relative expression levels of indicated transcripts in round spermatids from adult *Igf2bp3*<sup>+/−</sup> and *Igf2bp3*<sup>−/−</sup> testes. Unpaired two-tailed *t* test. Error bars, *n* = 3 biological replicates, mean ± SEM. (H) Western blotting analysis of the protein levels of DOT1L and HDAC11 in *Igf2bp3*<sup>+/−</sup> and *Igf2bp3*<sup>−/−</sup> RS. ACTB serves as a loading control. (I) Western blotting analysis of the protein levels of H3K79me2 in *Igf2bp3*<sup>+/−</sup> and *Igf2bp3*<sup>−/−</sup> RS. H3 serves as a loading control.

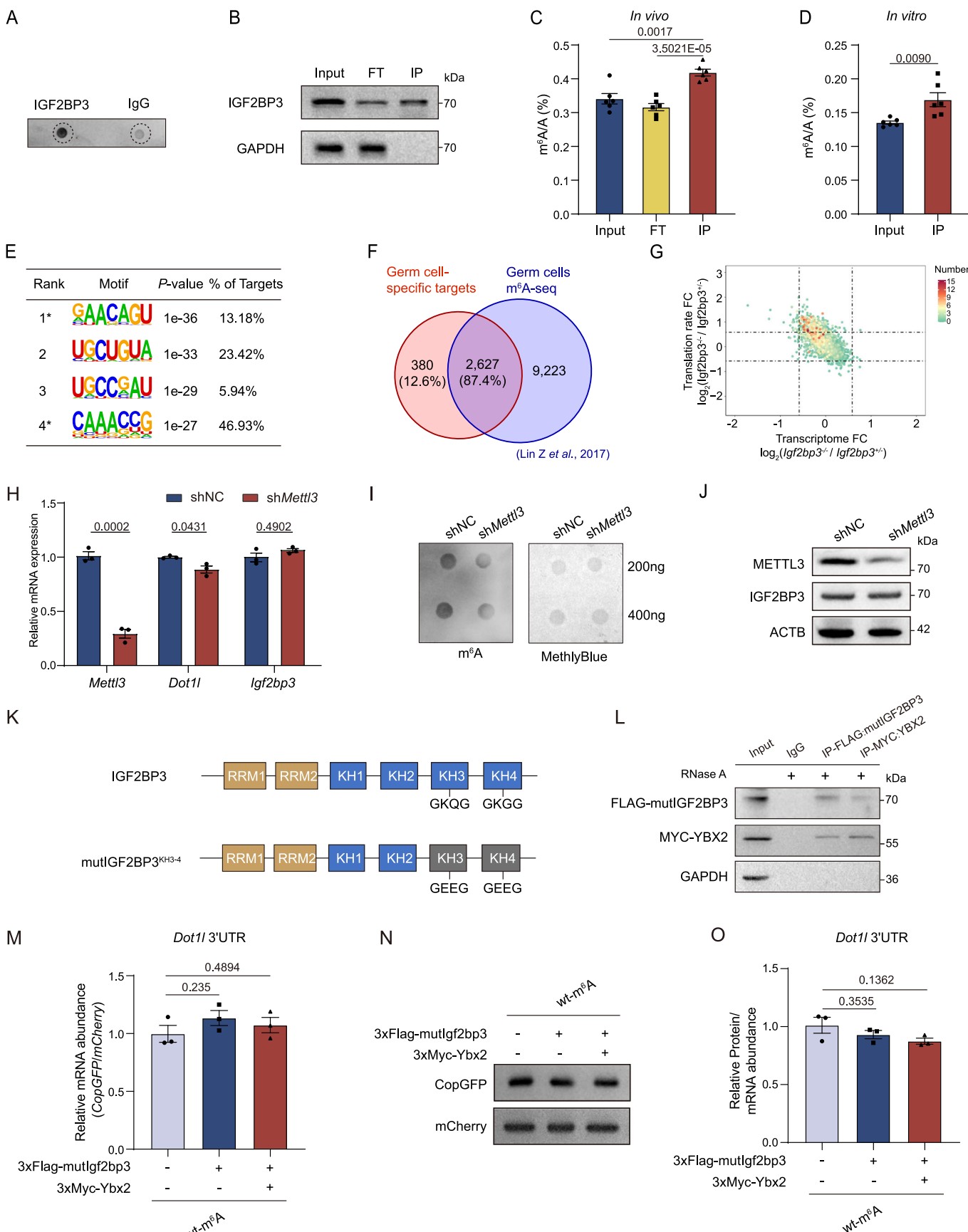

◀

**Figure EV4.  IGF2BP3 recognizes and regulates m⁶A-containing RNAs in mouse testes, related to Fig. 6.**

(A) Dot blotting analysis of the m⁶A level in IGF2BP3-bound RNAs isolated from adult testes with IgG as the control. (B) Western blotting analysis using adult mouse testicular lysates prior to RIP. (C) RIP LC-MS/MS showing m⁶A enrichment in IGF2BP3-bound RNAs while depleted in the flow-through (FT) portion (mouse testes) ($n = 6$, two technical replicates over three biologically replicates). Unpaired two-tailed $t$ test. Error bars, mean ± SEM. (D) LC-MS/MS showing m⁶A enrichment in His6-IGF2BP3 bound mRNA portion. ($n = 6$, two technical replicates over three biologically replicates). Unpaired two-tailed $t$ test. Error bars, mean ± SEM. (E) Top consensus sequences on overlapping IGF2BP3-bound peaks using HOMER. The motifs marked with * represent m⁶A motifs. (F) Venn diagram showing the overlap between 3,007 germ cell-specific targets of IGF2BP3 identified in Fig. 4C and m⁶A-modified mRNA in testicular germ cells (Lin et al, 2017). (G) Scatter plot showing the number of m⁶A-modified sites of IGF2BP3 targets with the mRNA-level changes ($x$ axis) against TE changes ($y$ axis) between adult $Igf2bp3^{-/-}$ and $Igf2bp3^{+/-}$ round spermatids. (H) qPCR analyses of the relative expression levels of $Mettl3$, $Dot1l$ and $Igf2bp3$ mRNA normalized to $\beta$-$Actin$ in GC-2 cells treated with sh$Mettl3$ or control shRNA. Unpaired two-tailed $t$ test. Error bars, $n = 3$ biological replicates, mean ± SEM. (I) Dot blotting analysis of the global m⁶A level of RNA extracted from $Mettl3$-knockdown or control GC-2 cells. (J) Western blotting analysis of the protein levels of METTL3 and IGF2BP3 in GC-2 cells treated with sh$Mettl3$ or control shRNA. ACTB serves as a loading control. (K) Schematic structures showing RNA-binding domains within IGF2BP proteins and a summary of IGF2BP variants used in this study. Yellow boxes are RRM domains, blue boxes are wild-type KH domains with GxxG motifs, and grey boxes are inactive KH domains with GxxG converted to GEEG. (L) Western blotting analysis of FLAG-mutIGF2BP3 and MYC-YBX2 in the Flag-mutant $Igf2bp3$ and Myc-$Ybx2$ co-transfected HEK293T cell lysates (input), and the lysate immunoprecipitation with anti-IgG, anti-FLAG or anti-MYC antibodies treated with RNase A (+), respectively. (M) qPCR analyses of the relative levels of $CopGFP$ mRNAs normalized to $mCherry$ mRNAs. Cell lines were treated with 2 µg/ml actinomycin D for 2 h. Unpaired two-tailed $t$ test. Error bars, $n = 3$ biological replicates, mean ± SEM. (N) Western blotting analysis of the protein levels of CopGFP under the regulation of $Dot1l$ 3′UTR with the overexpression of FLAG-tagged mutant IGF2BP3 or MYC-tagged YBX2. The level of mCherry is set as the internal control. (O) Histogram showing the ratios of CopGFP proteins (normalized to mCherry proteins) to the $CopGFP$ mRNAs (normalized to $mCherry$ mRNAs), corresponding to Fig. EV4M,N. Unpaired two-tailed $t$ test. Error bars, $n = 3$ biological replicates, mean ± SEM.

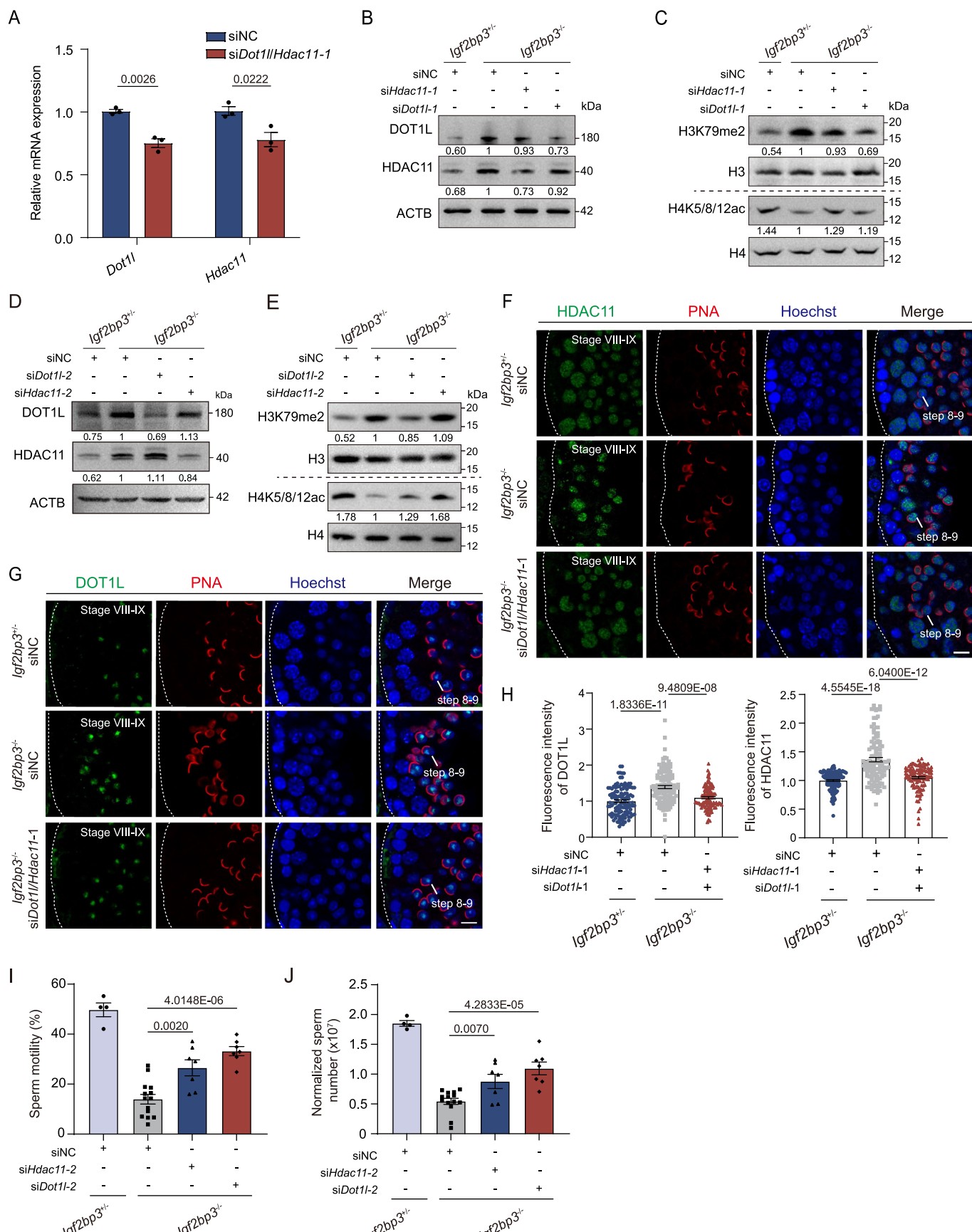

◀ **Figure EV5. Testicular DOT1L and HDAC11 down-regulation rescue the sperm developmental disorders in IGF2BP3-KO testes, related to Fig. 7.**

(A) qPCR analyses of the relative expression levels of *Dot1l* and *Hdac11* mRNAs normalized to *β-Actin* in RS from *Igf2bp3*$^{-/-}$ mouse testes injected with siNC or si*Dot1l*/*Hdac11*-1. Unpaired two-tailed *t* test. Error bars, $n = 3$ biological replicates, mean ± SEM. (B–E) Western blotting analysis of the protein levels of DOT1L, HDAC11, H3K79me2 and H4K5/8/12ac from adult *Igf2bp3*$^{+/-}$ and *Igf2bp3*$^{-/-}$ mouse testes injected with negative control siRNA (siNC), siRNA targeting *Dot1l* (si*Dot1l*-1/2) or siRNA targeting *Dot1l* (si*Hdac11*-1/2). The values below each band represent the relative expression levels of each protein. ACTB serves as the internal control of DOT1L and HDAC11, H3 serves as the internal control of H3K79me2, H4 serves as the internal control of H4Ac. (F, G) Immunofluorescence of HDAC11 or DOT1L (green) and PNA (red) in paraffin sections of adult *Igf2bp3*$^{+/-}$ and *Igf2bp3*$^{-/-}$ mouse testes injected with siNC or si*Dot1l*/*Hdac11*-1. Scale bar, 10 μm. Dotted borders demarcates the basement membrane of the seminiferous tubule. (H) Quantification of fluorescence intensity of DOT1L (left) and HDAC11 (right) in paraffin sections of adult *Igf2bp3*$^{+/-}$ and *Igf2bp3*$^{-/-}$ mouse testes injected with siNC or si*Dot1l*/*Hdac11*-1, corresponding to Fig. EV5F,G. Unpaired two-tailed *t* test. Error bars, $n =$ over 88 cells from 3 biological replicates, mean ± SEM. (I, J) CASA of the percentage of motile sperm (I) and the corresponding concentration of epididymal spermatozoa (J) from adult *Igf2bp3*$^{+/-}$ and *Igf2bp3*$^{-/-}$ mouse injected with siNC, si*Dot1l*-2 or si*Hdac11*-2. Unpaired two-tailed *t* test. Each bar represents the mean ± SEM from 7 biological replicates.

