## [Peer Review File · The EMBO Journal]

IGF2BP3 recognizes m⁶A to regulate histone-to-protamine replacement during mouse sperm development

Dazhuang Wang, Zhenyi Huang, Yichun Zhou, Peiyan Chen, Gang Chang, Liwei Ke, Congying Jing, Haojie Yang, Jiexiang Zhao, Shaofang Ren, Yi Zheng, Yuhan Chen, Yufan Xiang, Jun Liu, and Mei Wang

Corresponding author: Mei Wang (wangmei94@smu.edu.cn)

Review Timeline:

Submission Date:	9th Jun 25
Editorial Decision:	7th Jul 25
Revision Received:	18th Sep 25
Editorial Decision:	22nd Oct 25
Revision Received:	2nd Nov 25
Accepted:	14th Nov 25

Editor: Ieva Gailite

Transaction Report:

Dear Dr. Wang,

Thank you for submitting your manuscript for consideration by The EMBO Journal. We have now received comments from a full set of reviewers, which are included below for your information.

As you will see, reviewers #1 and #3 are generally positive in their evaluation and appreciate the contribution of the study to the research field. Furthermore, all reviewers indicate a number of concerns regarding the experimental approach, data analysis and interpretation that would need to be addressed before they can support publication. While reviewer #2 is more critical regarding the novelty of the findings, this is not a concern from the editorial side.

From my side, I find these requests for revisions generally reasonable. Therefore, based on these positive assessments by two of the reviewers, I invite you to submit a revised manuscript in response to the comments by all reviewers. I think that it would be useful to discuss the revision in more detail via email or phone/videoconferencing - please let me know which option you prefer.

We generally allow three months as standard revision time, which can be extended to six months in the case of major revisions. Should you foresee a problem in meeting this deadline, please let us know in advance to discuss an extension.

As a matter of policy, competing manuscripts published during this period will not negatively impact on our assessment of the conceptual advance presented by your study. However, please contact me as soon as possible upon publication of any related work to discuss the appropriate course of action.

When preparing your letter of response to the referees' comments, please bear in mind that this will form part of the Review Process File and will therefore be available online to the community. For more details on our Transparent Editorial Process, please visit our website: <https://www.embopress.org/page/journal/14602075/authorguide#transparentprocess>. Please also see the attached instructions for further guidelines on preparation of the revised manuscript.

Please feel free to contact me if you have any further questions regarding the revision. Thank you for the opportunity to consider your work for publication. I look forward to discussing your revision.

With best regards,

Ieva

Ieva Gailite, PhD
Senior Scientific Editor
The EMBO Journal
Meyerohofstrasse 1
D-69117 Heidelberg
Tel: +4962218891309
i.gailite@embojournal.org

- a point-by-point response to the referees' comments, with a detailed description of the changes made (as a word file).
- a word file of the manuscript text.
- individual production quality figure files (one file per figure)

- a complete author checklist, which you can download from our author guidelines (<https://www.embopress.org/page/journal/14602075/authorguide>).

- Expanded View files (replacing Supplementary Information)

We realize that it is difficult to revise to a specific deadline. In the interest of protecting the conceptual advance provided by the work, we recommend a revision within 3 months (5th Oct 2025). Please discuss the revision progress ahead of this time with the editor if you require more time to complete the revisions.

Referee #1:

Modification of mRNA with N6-methyladenine (m6A) is known to be important for spermatogenesis, but its exact role is not well understood. In this study, Wang et al demonstrate that the testis-enriched m6A binding protein IGF2BP3 regulates translation of m6A-modified transcripts and that this function is important for the histone-to-protamine transition in sperm. They generate whole-body knockouts (KO) of *Igf2bp3* using CRISPR and find that KO males have multiple sperm defects and variable levels of subfertility and infertility, as well as increased expression of chromatin regulator proteins such as DOT1L and HDAC11. They then characterize IGF2BP3 RNA targets and translational effects in testes using eCLIP-seq, RNA-seq, and ribosome nascent-chain complex sequencing (RNC-seq), finding that IGF2BP3 binds to a set of target mRNAs and its loss appears to reduce transcript stability and increase translation efficiency for a subset of these targets. By IP-mass spec, they show that YBX2 interacts with IGF2BP3 and facilitates but is not essential for this activity. Among the mRNA targets are several chromatin regulators, including *Dot1l* and *Hdac11*, consistent with the observation that levels of the encoded proteins are elevated in the *Igf2bp3* KO, and knockdown of these targets by siRNA injection into testes partially rescues molecular and developmental defects. To support the contribution of the m6A modification to these effects, they mutate putative m6A target sites in the *Dot1l* 3'UTR and confirm reduced mRNA abundance and increased translation efficiency in an in vitro assay.

Overall, experiments are carefully done and well controlled, data is high quality and clearly presented, and each component of the model is supported with data. Together, the findings advance understanding of translational regulation and RNA modification in mammalian spermatogenesis, and fill an important gap in understanding the regulation of the histone-to-protamine transition. However, in some cases the data shown does not fully support the claim made. Additional quantitation and statistics should be added, especially for Western blots that are used to support claims about quantitative changes. Finally, more discussion of the complexity of some of the findings should be included.

Major points:

- 1) In Figure 1, the microscopy images are small and localization can be hard to see. A panel showing higher magnification for each combination of markers would be helpful.
- 2) According to Figure 1B, there is not much variation in IGF2BP3 expression in the first-wave time course, although it does look higher at 4 and 5 week time points. Specifically, there appears to be robust expression of IGF2BP3 at the 1 week time point, when spermatogonia are the only germ cell type present in testes. This would suggest there is reasonably strong expression of IGF2BP3 in spermatogonia in addition to spermatocytes and spermatids. If so, why is expression not seen in spermatogonia (LIN28 co-staining) in Figure 1C?
- 3) The fertility (average number of pups), sperm motility, and sperm count of the *Igf2bp3* KO males is quite variable as shown in Figure 2. Some discussion should be added about possible reasons for the variability, and how this functional variability might affect the interpretation of molecular studies in the manuscript.
- 4) In Figure 2H quantitation and statistics should be shown to support the claim about abnormal nuclei.

- 5) The Western blot shown in Figure 3B supports a central claim of the paper, that histone retention is increased in KO sperm. Although the result looks visually believable, quantitation of the signal with loading control normalization should be included to more robustly support this claim.
- 6) A Western blot showing reduction in PRM1/PRM2 in haploid cells as shown for H4ac and TNP2 in Figure 3G and S3H should be included for completeness.
- 7) The correlation between eCLIP-seq replicates is low (Figure S4B,D). This should be acknowledged. Also, presumably the 5,095 target genes overlapping between replicates (Figure S4D) is the set that was used for further analysis, but this should be stated more clearly.
- 8) The set of 534 genes that is highly expressed in both germ and somatic cells and are targets as identified by eCLIP-seq (Figure 4C) are also potentially real and interesting targets. Can the authors provide some more information on this set of genes?
- 9) Are the differences between IP and IgG conditions in the *Igf2bp3*^{-/-} condition in Figure 5D statistically significant? The result of the statistical test (significant or not) is important for interpreting the data and should be indicated in the figure. Similarly, in Figure 6I the significance of the difference between IgG and IP conditions with the mut-m6A construct should be shown.
- 10) The Western blots in Figures 7B,C and S8B,C are important for making an argument about protein quantity, so quantitation of bands should be shown. By IF (Fig S8D,E), changes in DOT1L and HDAC11 protein levels seem very modest. This change in protein level is referred to again in line 13.28-30: "Consistently, a significant increase in H4Ac was observed in adult *Igf2bp3*^{-/-} testes solely injected with siRNA targeting *Dot1l* (si-*Dot1l*) (Figure 8C). This increase is hard to see in Figure 8C and would be better supported with quantitation and statistical analysis of the data.
- 11) In Figure 7G, it looks like levels of histones are rescued by the double siRNA, but levels of PRM1 and PRM2 appear unchanged. Again, quantitation would be helpful here.
- Minor:
- 12) Line 4.6: typo: conservative conserved
- 13) Lines 7.3-5 (referring to Figure 3D, S3E): "We found that testis-specific histone variants, transition proteins, and protamines...". This should more accurately be stated as "We found that mRNAs encoding testis-specific histone variants, transition proteins, and protamines..."
- 14) Line 7.7-8 (Referring to Figure 3E, S3F): "histone modifications facilitating the eviction of histones, such as H4ac (H4K5ac, H4K8ac, H4K12ac)": for completeness, this list should also include H4K16ac.
- 15) Figure 4H: The dashed line seems misleading: why is it set at y=5? Why not y=1 as a baseline for fold change?
- 16) Figure S6A: I could not find the reference for the YBX2 CLIP-seq dataset (presumably PMID:19597149) in the References section. If this the dataset that was used the fact that the YBX2 dataset is also from testes is a relevant piece of information and should be specified in the text.

Please number lines continuously or add page numbers for easier reference.

Referee #2:

Wang et al. investigated the role of the m6A-binding protein IGF2BP3 in spermatogenesis using knockout mice. The authors found that deleting IGF2BP3 impaired the histone-to-protamine exchange process during spermatogenesis, resulting in male subfertility and infertility. IGF2BP3 was found to regulate the translation of specific targets, particularly *Dot1l* and *Hdac11*. However, the function of IGF2BP3 in germ cell development has previously been demonstrated in zebrafish (PMID: 34214072), which means that this work is not particularly novel. Furthermore, important functional experiments are lacking to support the main conclusion, particularly with regard to the primary defects and mechanism.

Major concerns:

1. Is the IGF2BP3 signal located in the chromatoid body, a germ cell-specific RNA processing center, in round spermatids? If so, are there any defects in the chromatoid body or the RNA processing proteins located in the chromatoid body? In my opinion, since knocking out IGF2BP3 does not affect the proportion of spermatogonia, spermatocytes and Sertoli cells, but specifically influences haploid spermatids, the defects in histone-to-protamine exchange in KO mice are probably related to the unique localization of IGF2BP3.
2. Since both IGF2BP3 and YBX2 can bind the 3'UTR of *Dot1l*, what impact does the m6A reading function of IGF2BP3 have on

the level of Dot11 protein?

3. As reported in previous literature, YBX2 binds to a series of gamete-specific mRNA, including PRM1 and PRM2 (<https://doi.org/10.1073/pnas.0404685102>). How many of these genes overlap with the CLIP-seq targets of YBX2 in this study? Has the binding capacity changed between YBX2 and PRM1 or PRM2?
4. Inconsistency in YBX2 cooperation and stage-specific phenotypes. YBX2 knockout causes defects as early as spermatocytes, whereas Igf2bp3 KO phenotypes manifest post-meiotically. The claim that YBX2's RNA-binding capacity is "completely lost" in Igf2bp3 KO without affecting spermatocytes requires clarification. Please discuss possible compensatory mechanisms (e.g., redundant m6A readers or RNA-binding proteins in spermatocytes). Test whether YBX2 localization or stability is altered in Igf2bp3 KO germ cells (e.g., by co-IP/WB/IF).
5. There is potential for crosstalk with other m6A readers. Due to the unique germline expression of IGF2BP3, it is necessary to determine whether its loss affects the expression or subcellular distribution of other m6A readers (e.g. YTHDF1/2/3 and YTHDC1/2). Simple WB/qRT-PCR analysis of key readers in WT vs. KO testes would address this.
6. The subcellular localisation of IGF2BP3 and its potential association with P-bodies. The punctate staining of IGF2BP3 observed in round spermatids strongly resembles that of cytoplasmic granules (e.g., P-bodies or chromatid bodies). As IGF2BP3 is known to regulate mRNA localisation to P-bodies (see, for example, the article 'm6A modification negatively regulates translation by switching mRNA from polysome to P-body via IGF2BP3' in *Molecular Cell*), this connection should be explored. Co-stain with P-body markers (e.g., DCP1A or others) to confirm the granule identity of IGF2BP3.
7. Assess whether Igf2bp3 KO disrupts granule formation or RNA localization (e.g. via RNA-FISH for target transcripts such as Dot11 or Hdac11). This would significantly strengthen the mechanistic model of translational repression.

Other concerns:

1. Are there any differences between subfertile and infertile males? What about the infertility ratio? Which type of male is used in the experiments?
2. In Figure 3, the immunofluorescence images of H2A/H2B/H3/H4 in sperm heads lack clarity. Add arrows/insets to highlight specific abnormalities (e.g., nuclear retention or mislocalization) in KO sperm.
3. Technical limitation: siRNA penetration across the blood-testis barrier (BTB) is notoriously inefficient in 4-weeks old mice. The sub-fertile phenotype of Igf2bp3 KO mice may not conclusively demonstrate functional recovery, as off-target effects or incomplete knockdown cannot be ruled out. Experimental rigor: (a) At least two independent siRNAs per target gene (Dot11 and Hdac11) must be used to confirm specificity. (b) Fertility restoration assays are critical: Assess whether siRNA-treated Igf2bp3 KO mice show improved fertility (e.g., litter size, sperm counts, or mating trials).

Referee #3:

Reviewer comments: IGF2BP3 Recognizes m6A to Regulate Histone-to-protamine Replacement during Mouse Sperm Development, Wang et al

General summary and importance:

The discovery of the reversible nature of m6A in RNA has inspired numerous studies to identify the biological role of m6A dynamics - including the characterization of the many m6A binding proteins that are required for m6A to have a role in gene regulation. In most model organisms studied the role of methylating A to m6A is particularly obvious during germ cell maturation, with "similar" phenotypes for male and female germ cells. Additionally, several m6A binding proteins affect fertility if mutated. In mammals this is also shown to be a key feature for mice lacking the m6A demethylase, ALKBH5.

This work was initiated with the identification of IGF2BP3 being relevant for m6A during spermatogenesis based on expression profiling, and Igf2bp3 mutation led to male sub/infertility. They identified the histone to protamine transition to be defect in Igf2bp3 mutant mice. Some of the analysis, e.g. the detailed characterization of the histone-protamine transition, the overlap of IGF2BP3 pulldown and MeRIP, and mutating the m6A motif of Dot11 and Hdac11 are particularly impressive, and most studies neatly combine molecular analysis with phenotypic characteristics.

Major comment:

My only major concern relates to the generation and identification of Igf2bp3 mutant mice. I cannot find information regarding which of the two different IGF2BP3 antibodies used for immunostaining of cells: The Bethyl antibody used is raised against the C-terminus (although described as A303-426A in the table I assume it is raised against the C-terminus aa 529-579 as described in Remark to the antibody table). The Abcam antibody (if the upper described in antibody table remark?) is raised against the N-terminus. Thus, they should differ in their ability to identify a truncated IGF2BP3 protein still containing the two RRM domains. The truncated protein, if at all present, would not be identified in the Western blot shown for sizes only corresponding to the wild-type protein. Was any mRNA analysis done to quantitate the truncated mRNA transcript (if having a premature stop in the interrupted mutant) and possible address its ability to be translated to a protein and provide a dominant-negative effect?

Minor comments:

Some language/grammar editing would improve the manuscript.

Line 26 Introduction page 3: I assume m6A, despite different levels, is present in all spermatogenic cells (delete almost).

Line 28 Introduction page 3: Correct singular/plural of m6A modifying enzymes.

Line 6 Introduction page 4: correct "conservative in" to "conserved between"

Line 24 Results page 4: delete "relatively" (since compared is also used in the sentence)

Lines 26 Results page 4: this sentence is very long and must be corrected grammatically.

Line 16 Results page 5: I don't agree that this is a good reason for not also delete the RRM domains. Do you have any additional information to provide on this (see also major comment above)?

Line 25 Results page 7: these results are very nice, and I wonder if you also have information regarding any conserved motif (like the m6A motif?)

Section starting at line 11 page 9: could you elaborate briefly on the possibility of the phenotype of Igf2bp3 mutant mice to depend on m6A binding and/or interaction with YBX2, idenpendently?

Any comments relating to the (sub/in)fertility of female Igf2bp3 mutant mice? Based on line 15 page 15 I assume they have normal fertility.

Point by point response to the Reviewers' questions:

Below we have listed the point-by-point response to each of the questions, with the answers highlighted in blue.

Referee #1:

Modification of mRNA with N6-methyladenine (m⁶A) is known to be important for spermatogenesis, but its exact role is not well understood. In this study, Wang et al demonstrate that the testis-enriched m⁶A binding protein IGF2BP3 regulates translation of m⁶A-modified transcripts and that this function is important for the histone-to-protamine transition in sperm. They generate whole-body knockouts (KO) of *Igf2bp3* using CRISPR and find that KO males have multiple sperm defects and variable levels of subfertility and infertility, as well as increased expression of chromatin regulator proteins such as DOT1L and HDAC11. They then characterize IGF2BP3 RNA targets and translational effects in testes using eCLIP-seq, RNA-seq, and ribosome nascent-chain complex sequencing (RNC-seq), finding that IGF2BP3 binds to a set of target mRNAs and its loss appears to reduce transcript stability and increase translation efficiency for a subset of these targets. By IP-mass spec, they show that YBX2 interacts with IGF2BP3 and facilitates but is not essential for this activity. Among the mRNA targets are several chromatin regulators, including *Dot11* and *Hdac11*, consistent with the observation that levels of the encoded proteins are elevated in the *Igf2bp3* KO, and knockdown of these targets by siRNA injection into testes partially rescues molecular and developmental defects. To support the contribution of the m⁶A modification to these effects, they mutate putative m⁶A target sites in the *Dot11* 3'UTR and confirm reduced mRNA abundance and increased translation efficiency in an in vitro assay.

Overall, experiments are carefully done and well controlled, data is high quality and clearly presented, and each component of the model is supported with data. Together, the findings advance understanding of translational regulation and RNA modification in mammalian spermatogenesis, and fill an important gap in understanding the regulation of the histone-to-protamine transition. However, in some cases the data

shown does not fully support the claim made. Additional quantitation and statistics should be added, especially for Western blots that are used to support claims about quantitative changes. Finally, more discussion of the complexity of some of the findings should be included.

Answer: We appreciate your positive comments and acknowledgement of our study's importance.

Major points:

1) In Figure 1, the microscopy images are small and localization can be hard to see. A panel showing higher magnification for each combination of markers would be helpful.

Answer: Thank you for your valuable suggestion. The images showing higher magnification of the IGF2BP3 staining pattern during spermatogenesis had been correspondingly highlighted (Figure 1 in response, Figure 1F in the revised manuscript).

Figure 1.

Scheme of IGF2BP3 expression dynamics during spermatogenesis. Green marks the protein level of IGF2BP3. SPG, spermatogonia; L/Z, leptotene or zygotene spermatocytes; P/D, pachytene or diplotene spermatocytes; MI-MII, metaphase I to II.

2) According to Figure 1B, there is not much variation in IGF2BP3 expression in the first-wave time course, although it does look higher at 4 and 5 week time points. Specifically, there appears to be robust expression of IGF2BP3 at the 1 week time point, when spermatogonia are the only germ cell type present in testes. This would

suggest there is reasonably strong expression of IGF2BP3 in spermatogonia in addition to spermatocytes and spermatids. If so, why is expression not seen in spermatogonia (LIN28 co-staining) in Figure 1C?

Answer: Thank you for your question. A quantitative analysis of IGF2BP3 protein levels in mouse testes at different postnatal weeks had been performed. The results revealed a gradual increase in IGF2BP3 expression from the first to the third postnatal week (Figure 2 in response, Figure 1B in the revised manuscript). As you pointed out, IGF2BP3 was already detectable in the first week, suggesting its presence in spermatogonia. This observation was corroborated by the relatively low expression levels observed in Lin28a-positive spermatogonia in Figure 1C in the previous manuscript. Although our study primarily focused on the role of IGF2BP3 in spermatids, we acknowledge that the weak signal detected in spermatogonia may imply a potential involvement of IGF2BP3 in spermatogonial self-renewal or differentiation.

Figure 2

Figure 2.

Western blotting analysis of the protein level of IGF2BP3 in mouse testes at different postnatal weeks with ACTB as the internal control. The values below each band represent the relative expression levels of IGF2BP3.

3) The fertility (average number of pups), sperm motility, and sperm count of the Igf2bp3 KO males is quite variable as shown in Figure 2. Some discussion should be added about possible reasons for the variability, and how this functional variability might affect the interpretation of molecular studies in the manuscript.

Answer: Thank you for your suggestion. Explanations and corresponding discussions of these observations are provided below and further elaborated in the Discussion section of the revised manuscript.

First, we had also observed variations in fertility among individual *Igf2bp3*^{-/-} male mice, ranging from sub-fertility to infertility, along with differences in sperm motility and sperm count. These observations were highly consistent with those reported in other gene knockout male mice exhibiting impaired fertility, such as *Chd5*^{-/-}, *Tex19.1*^{-/-}, *Tnp1*^{-/-}, and *H2a.b.3*^{-y} mice (Anuar *et al*, 2019; Li *et al*, 2014; Ollinger *et al*, 2008; Yu *et al*, 2000). Although the underlying causes of fertility variation observed in these knockout models may differ, all of these genes are highly expressed during spermiogenesis. This suggested that the deletion of key regulators enriched during spermiogenesis may lead to a common phenotype characterized by inherent fertile variability in different individuals. Furthermore, based on the explanations provided in these reports, we proposed that environmental context and redundancy of histone modifications resulting from the upregulated DOT1L and HDAC11 would serve as plausible explanations for the observed fertility variations in *Igf2bp3*^{-/-} male mice, especially considering that the genetic background (C57BL/6), which was used to minimize any genetic instability.

Moreover, our previous results indicated that approximately 65% of *Igf2bp3*^{-/-} male mice displayed sub-fertility and 10% of *Igf2bp3*^{-/-} male mice were completely infertile when mated with wild-type female mice (Figure 2C in the previous manuscript). To minimize variability and more accurately reflect the mechanisms by which IGF2BP3 regulates spermatogenesis, *Igf2bp3*^{-/-} male mice with impaired fertility were selected for the experiments presented in Figures 3 to 5. It is important to clarify that we could not specifically select sub-fertile or infertile male mice for the siRNA microinjection experiment, as it was conducted using 4-week-old mice in Figure 7. Thus, siNC and si*Dot1l/Hdac11* were injected into the left and right testes of the same knockout mouse, respectively. Although variations in fertility and sperm quality were observed

among IGF2BP3 knockout male mice, the well-structured experimental design ensures that the molecular mechanism elucidated in this study accurately reflects the genuine regulatory role of IGF2BP3 in spermatogenesis. To ensure clarity regarding the phenotypic variability of *Igf2bp3*^{-/-} male mice, we had elaborated on the corresponding information in Methods section in the revised manuscript.

4) In Figure 2H quantitation and statistics should be shown to support the claim about abnormal nuclei.

Answer: We appreciate your suggestion. A quantitative and statistical analysis of the abnormal cell nuclei corresponding to Figure 2H in the previous manuscript had been conducted, and the result strongly supported our conclusion that abnormal nuclei accumulated significantly in *Igf2bp3*^{-/-} testes compared to *Igf2bp3*^{+/-} testes (Figure 3 in response, Figure EV1F in the revised manuscript).

Figure 3

Figure 3.

Quantitative comparison of abnormal nuclei per tubule between adult *Igf2bp3*^{+/-} (n = 4) and *Igf2bp3*^{-/-} (n = 5) testes. At least 20 tubules of each mouse were calculated. Unpaired two-tailed *t*-test. *** *P* < 0.001. Each bar represents the mean ± SEM from biological replicates.

5) The Western blot shown in Figure 3B supports a central claim of the paper, that histone retention is increased in KO sperm. Although the result looks visually believable, quantitation of the signal with loading control normalization should be included to more robustly support this claim.

Answer: Thanks for your suggestion. The quantified results, normalized to the loading controls, have been added to Figure 3B of the revised manuscript (Figure 4 in response).

Figure 4

Figure 4.

Western blotting analysis of the protein levels of H2A, H2B, H3, H4, PRM1 and PRM2 in epididymal spermatozoa from *Igf2bp3*^{+/-} and *Igf2bp3*^{-/-} mice. The values below each band represent the relative expression levels of each protein, normalized using α -Tubulin as a loading control.

6) A Western blot showing reduction in PRM1/PRM2 in haploid cells as shown for H4ac and TNP2 in Figure 3G and S3H should be included for completeness.

Answer: Thanks for your suggestion. We have added the Western blotting results showing PRM1/PRM2 levels in haploid cells (Figure 5A in response, Figure EV2H in the revised manuscript). Consistently, quantitative analyses of PRM1/PRM2 levels corresponding to Figures 3G and S3H in the previous manuscript also demonstrated a significant reduction in these proteins in *Igf2bp3*^{-/-} elongated spermatids (Figures 5B and 5C in response, Figures EV2J-K in the revised manuscript).

Figure 5

Figure 5.

A. Western blotting analysis of the protein levels of PRM1 and PRM2 in haploid cells from *Igf2bp3*^{+/-} and *Igf2bp3*^{-/-} mice. The values below each band represent the relative expression levels of each protein, normalized using ACTB as a loading control.

B-C. Quantification of fluorescence intensity of PRM1 (B) and PRM2 (C) in paraffin sections of adult *Igf2bp3*^{+/-} and *Igf2bp3*^{-/-} mouse testes, corresponding to Figures 3G and EV2I in the revised manuscript. Unpaired two-tailed *t*-test. *****P* < 0.0001. Error bars, *n* = over 150 cells from 3 biological replicates, mean ± SEM.

7) The correlation between eCLIP-seq replicates is low (Figure S4B,D). This should be acknowledged. Also, presumably the 5,095 target genes overlapping between replicates (Figure S4D) is the set that was used for further analysis, but this should be stated more clearly.

Answer: Thanks for your suggestions. The correlation between eCLIP-seq replicates is indeed relatively low compared to that of RNA-seq or ChIP-seq data. Since the specific binding sites of RBPs identified through eCLIP-seq account for only a very small proportion of the entire genome, the overall correlation of read distribution at the global level may be low due to the high proportion of background signals, which has been widely acknowledged by its developers of this method, the Yeo lab (Figure 6A in response) (Conway *et al*, 2016; Van Nostrand *et al*, 2016). Nevertheless, this is still a robust approach for identifying direct binding sites of RBPs with high resolution. For instance, the correlation and the proportion of overlapping target genes between two eCLIP-seq replicates in our study were both comparable to those observed in replicates from the Yeo lab and the Bluma J Lesch lab, indicating that the

correlation patterns in eCLIP-seq data are broadly consistent across studies (Figure 6B in response) (Conway *et al.*, 2016; Griffin *et al.*, 2022). Besides, we are sorry for neglecting to provide a clear statement regarding 5,095 target genes overlapping between two eCLIP-seq replicates in Figure S4D for further analysis, and this information has now been added accordingly in the revised manuscript.

Figure 6.

A. Scatter plots show the correlation of region-based fold enrichment in eCLIP across different datasets. Dotted lines and R^2 values indicate least-squares regression performed separately for each region type.

B. The Venn diagram shows the overlap of target genes identified in two eCLIP-seq biological replicates from different datasets.

8)The set of 534 genes that is highly expressed in both germ and somatic cells and are targets as identified by eCLIP-seq (Figure 4C) are also potentially real and interesting targets. Can the authors provide some more information on this set of genes?

Answer: Thank you for your question. The binding affinity of IGF2BP3 to 534 genes that were highly expressed in both germ cells and somatic cells was examined. The

result showed that IGF2BP3 bound to these genes at a significantly low level compared to the 3,007 germ cell-specific genes mainly analyzed in our study. This observation was consistent with the very low expression of IGF2BP3 in testicular somatic cells (Figure 7 in response). Therefore, these findings suggested that the dysregulation of these 534 genes was unlikely to be the primary cause of the subfertility or infertility observed in *Igf2bp3*^{-/-} male mice.

Figure 7

Figure 7.

Aggregated IGF2BP3 eCLIP-seq signals at 534 target genes highly expressed in both germ and somatic cells upon 3,007 germ cell-specific target genes.

9) Are the differences between IP and IgG conditions in the *Igf2bp3*^{-/-} condition in Figure 5D statistically significant? The result of the statistical test (significant or not) is important for interpreting the data and should be indicated in the figure. Similarly, in Figure 6I the significance of the difference between IgG and IP conditions with the mut-m6A construct should be shown.

Answer: Thanks for your suggestions. We had confirmed that there were significant differences between IP and IgG conditions in both the *Igf2bp3*^{-/-} condition in Figure 5D and the mutant m⁶A vector samples in Figure 6I (Figures 8A and 8B in response, Appendix Figure S4A and Figure 5D in revised manuscript). On one hand, these results indicated that YBX2 had a relatively low affinity to binding *Dot1l* and *Hdac1l* RNAs in mouse testes, even in the absence of IGF2BP3. Considering that neither *Dot1l* nor *Hdac1l* was identified as the CLIP-seq targets of YBX2 in testes (Figure 8C in response) (Xu *et al*, 2009), it was likely that YBX2 indirectly bound these two

RNAs by collaborating with other RBPs in testes. Together, these findings expanded the known regulatory repertoire of YBX2 in sperm development, which aligned well with the discussion presented in our previous manuscript. On the other hand, these statistical analyses also indicated that IGF2BP3 was capable of recognizing and binding RNAs through non-m⁶A regions. This observation was highly consistent with both our motif analysis and previous studies, demonstrating that the CA-rich element was also essential for IGF2BP3 binding (Biswas *et al*, 2019; Conway *et al.*, 2016; Ketchum *et al*, 2018).

Figure 8

Figure 8.

A. Venn diagram showing the 3,007 germ cell-specific targets of IGF2BP3 (in the current study) and 212 CLIP-seq targets of YBX2 in mouse testes (Xu MG *et al.*, 2009).

B. YBX2 RIP-qPCR analyses of *Dot1l* and *Hdac11* in adult *Igf2bp3*^{+/-} and *Igf2bp3*^{-/-} testes. Unpaired two-tailed *t*-test. *** *P* < 0.001, ** *P* < 0.01, * *P* < 0.05. Error bars, n = 3 biological replicates, mean ± SEM.

C. RIP-qPCR analysis showing the binding preference of FLAG-IGF2BP3 to *Dot1l* mRNA in HEK293T. Unpaired two-tailed *t*-test. **** *P* < 0.0001, *** *P* < 0.001. Error bars, n = 3 biological replicates, mean ± SEM.

10) The Western blots in Figures 7B,C and S8B,C are important for making an argument about protein quantity, so quantitation of bands should be shown. By IF (Fig S8D,E), changes in DOT1L and HDAC11 protein levels seem very modest. This change in protein level is referred to again in line 13.28-30: "Consistently, a significant increase in H4Ac was observed in adult *Igf2bp3*^{-/-} testes solely injected with siRNA targeting Dot11 (si-Dot11) (Figure S8C). This increase is hard to see in Figure S8C and would be better supported with quantitation and statistical analysis of the data.

Answer: We thank the reviewer for these suggestions. Quantitative analyses had been added to Figures 7B, 7C, S8B, and S8C, further supporting the corresponding conclusions (Figures 9A-9D in response; Figures 7B, 7C, EV5B, and EV5C in the revised manuscript). Additionally, quantification and statistical analysis had been conducted for Figures S8D and S8E, confirming the significant changes in DOT1L and HDAC11 protein levels *in situ* under different experimental conditions (Figures 9E and 9F in response; Figure EV5H in the revised manuscript). Previously, the interaction between DOT1L and H4K5/8/12ac in mouse testes had been discussed in our manuscript. As you mentioned, we also observed the modest increase in H4K5/8/12ac level in adult *Igf2bp3*^{-/-} testes solely injected with si*Dot11* in Figure S8C. To further validate this observation, we performed additional Western blotting and quantitation analyses in *Igf2bp3*^{-/-} testes injected with either siNC or si*Dot11*-1. These results further supported the notion that DOT1L/H3K79me2 negatively regulates H4 acetylation during spermatogenesis (Figures 9G-9I in response).

Figure 9

Figure 9.

A-D. Western blotting analysis of the protein levels of DOT1L, HDAC11, H3K79me2 and H4K5/8/12ac was performed on adult *Igf2bp3^{+/+}* and *Igf2bp3^{-/-}* mouse testes injected with negative control siRNA (siNC), siRNA targeting *Dot1l* (siDot1l-1), siRNA targeting *Hdac11* (siHdac11-1) (C and D) or siRNAs targeting *Dot1l* and *Hdac11* (siDot1l/Hdac11-1) (A and B). The values below each band represent the relative expression levels of each protein. ACTB serves as the internal control of DOT1L and HDAC11, H3 serves as the internal control of H3K79me2, H4 serves as the internal control of H4K5/8/12ac.

E-F. Quantification of fluorescence intensity of DOT1L (E) and HDAC11 (F) in paraffin sections of adult *Igf2bp3^{+/+}* and *Igf2bp3^{-/-}* mouse testes injected with siNC or siDot1l/Hdac11-1, corresponding to Figures EV5F and EV5G in the revised manuscript. Unpaired two-tailed *t*-test. **** $P < 0.0001$. Error bars, $n =$ over 88 cells from 3 biological

replicates, mean \pm SEM.

G. Western blotting analysis of the protein level of DOT1L, HDAC11 from *Igf2bp3*^{-/-} mouse testes injected with negative control siRNA (siNC), or siRNA targeting *Dot1l* (si*Dot1l-1*). ACTB serves as the internal control.

H. Western blotting analysis of the protein level of H3K79me2 and H4K5/8/12ac from *Igf2bp3*^{-/-} mouse testes injected with siNC or si*Dot1l-1*. H3 serves as the internal control of H3K79me2, H4 serves as the internal control of H4K5/8/12ac.

I. The ratio of H4K5/8/12ac to total histone H4 was quantified by Western blotting from *Igf2bp3*^{-/-} mouse testes injected with siNC or si*Dot1l-1*. Unpaired two-tailed *t*-test. ** *P* < 0.01. Error bars, n = 3 biological replicates, mean \pm SEM.

11) In Figure 7G, it looks like levels of histones are rescued by the double siRNA, but levels of PRM1 and PRM2 appear unchanged. Again, quantitation would be helpful here.

Answer: Thanks for your suggestion. Quantitative analyses have been provided accordingly, further supporting our previous conclusion that the histone retention and protamine reduction in *Igf2bp3*^{-/-} epididymal spermatozoa could partially be rescued via the co-injection of si*Dot1l* and si*Hdac11* (Figure 10 in response, Figure 7G in the revised manuscript).

Figure 10

Figure 10.

Western blotting analysis of the protein level of H2A, H2B, H3, H4, PRM1, and PRM2 in epididymal spermatozoa from adult *Igf2bp3*^{+/-} and *Igf2bp3*^{-/-} mouse testes injected with siNC or *siDot11/Hdac11-1*. The values below each band represent the relative expression levels of each protein, normalized using α -Tubulin as the control.

Minor:

12) Line 4.6: typo: conservative \square conserved

Answer: We have corrected it as suggested in Line 77 in the revised manuscript.

13) Lines 7.3-5 (referring to Figure 3D, S3E): "We found that testis-specific histone variants, transition proteins, and protamines...". This should more accurately be stated as "We found that mRNAs encoding testis-specific histone variants, transition proteins, and protamines..."

Answer: We have corrected it as suggested in Line 171 page in the revised manuscript.

14) Line 7.7-8 (Referring to Figure 3E, S3F): "histone modifications facilitating the eviction of histones, such as H4ac (H4K5ac, H4K8ac, H4K12ac)": for completeness, this list should also include H4K16ac.

Answer: Thanks for your suggestion. As you mentioned, previous reports have demonstrated that H4K16ac is a representative form of histone H4 acetylation (Ketchum *et al.*, 2018; Qian *et al.*, 2013). However, in this study, we followed the established protocol for detecting testicular histone H4 acetylation used in other studies, employing the antibody (Santa Cruz, sc-377520) that specifically recognize H4K5ac, H4K8ac, and H4K12ac (Goudarzi *et al.*, 2016; Shiota *et al.*, 2018). To avoid ambiguity, H4K5/8/12ac had been clearly annotated in the revised manuscript.

15) Figure 4H: The dashed line seems misleading: why is it set at y=5? Why not y=1 as a baseline for fold change?

Answer: Thank you for your question. To address this, we have set $y=1$ as the fold change baseline in the previous Figure 4H (Figure 4H in the revised manuscript).

16) Figure S6A: I could not find the reference for the YBX2 CLIP-seq dataset (presumably PMID:19597149) in the References section. If this the dataset that was used the fact that the YBX2 dataset is also from testes is a relevant piece of information and should be specified in the text.

Answer: Thank you for your reminder. The reference for the YBX2 CLIP-seq dataset (PMID: 19597149) (Xu *et al.*, 2009) and the corresponding sample information have been included in the revised manuscript accordingly.

Please number lines continuously or add page numbers for easier reference.

Answer: We have accordingly provided this information in the revised manuscript.

Referee #2:

Wang *et al.* investigated the role of the m6A-binding protein IGF2BP3 in spermatogenesis using knockout mice. The authors found that deleting IGF2BP3 impaired the histone-to-protamine exchange process during spermatogenesis, resulting in male subfertility and infertility. IGF2BP3 was found to regulate the translation of specific targets, particularly Dot11 and Hdac11. However, the function of IGF2BP3 in germ cell development has previously been demonstrated in zebrafish (PMID: 34214072), which means that this work is not particularly novel. Furthermore, important functional experiments are lacking to support the main conclusion, particularly with regard to the primary defects and mechanism.

Answer: We sincerely appreciate your feedback. Prior to the initiation of our project, several studies had explored the potential involvement of IGF2BP3 in the development of primordial germ cells (PGCs) and early-stage embryos in zebrafish (Ren *et al.*, 2021; Vong *et al.*, 2021). However, no significant impairment in PGC or early embryonic development was observed in IGF2BP3 knockout mice. This suggested that the regulatory role of IGF2BP3 may differ between mammals and

zebrafish. Furthermore, IGF2BP3 knockout male mice predominantly exhibited impaired sperm development, indicating a novel function of IGF2BP3 in mouse spermatogenesis, a process in which RNA regulatory mechanisms are highly coordinated. Moreover, mRNA m⁶A modification and its binding proteins have been extensively investigated to regulate the mRNA metabolism and play crucial roles in spermatogenesis from spermatogonia to spermatocytes. However, the key binding protein involved in post-meiotic spermiogenesis and its specific regulatory mechanism governing spermatid maturation remain unveiled. Collectively, these findings, along with the specific expression of IGF2BP3 in mammalian post-meiotic spermatogenic cells, prompted us to comprehensively investigate its previously unexplored regulatory roles in RNA metabolism during mammalian spermatogenesis, which may provide new insights into sperm development and male infertility. Regarding the lack of functional experiments in previous manuscript, we have supplemented the relevant contents in accordance with your suggestions, as follows.

Major concerns:

1. Is the IGF2BP3 signal located in the chromatoid body, a germ cell-specific RNA processing center, in round spermatids? If so, are there any defects in the chromatoid body or the RNA processing proteins located in the chromatoid body? In my opinion, since knocking out IGF2BP3 does not affect the proportion of spermatogonia, spermatocytes and Sertoli cells, but specifically influences haploid spermatids, the defects in histone-to-protamine exchange in KO mice are probably related to the unique localization of IGF2BP3.

Answer: Thank you for your insightful questions. Testicular immunofluorescence co-staining of IGF2BP3 and DDX4 (a well-established marker of CBs) or YBX2 (the RNA processing protein present in CBs) first confirmed that a portion of the IGF2BP3 signals localized to the CBs of round spermatids, which are highly consistent with the previous mass spectrometry data on the composition of CBs (Figures 11A, 14A, and 14B in response) (Meikar *et al*, 2014). Although IGF2BP3 is present in CBs, these

granules were still formed in *Igf2bp3*^{-/-} round spermatids, as they accumulated in *Igf2bp3*^{+/-} RS (Figures 11B, 14A, and 14B in response). CBs are representative cytoplasmic RNPs essential for post-transcriptional mRNA regulation during spermatogenesis. Therefore, to assess the RNA aggregation capacity of CBs, newly synthesized RNAs were labeled with the nucleotide analog 5-ethynyluridine (EU), as described previously (Lehtiniemi *et al.*, 2022; Meikar *et al.*, 2014). The results revealed that EU-labeled CBs were also observed in *Igf2bp3*^{-/-} RS at levels comparable to those in *Igf2bp3*^{+/-} RS, suggesting that the RNA-aggregating capacity of CBs remains intact in the absence of IGF2BP3 (Figure 11C in response).

Although the specific components of RNA processing proteins (with the exception of YBX2 and DDX4) have not yet been fully examined in the *Igf2bp3*^{-/-} CBs, these structures represented canonical cytoplasmic ribonucleoprotein complexes (RNPs) in RS, as illustrated by the mechanistic model presented in Figure 6M of our previous manuscript. Therefore, we agree with your hypothesis that the unique localization of IGF2BP3 in round spermatids would be one of the explanations for the primary defects observed in IGF2BP3-deleted RS, rather than in other spermatogenic cells or Sertoli. In addition, as suggested by Reviewer 1, we also performed clustering analysis of IGF2BP3 eCLIP-seq binding signals for its 3,007 germ cell-specific target mRNAs. The results revealed a particularly strong binding affinity of IGF2BP3 to mRNAs that are highly expressed in round spermatids (Figures 11D and 11E in response). Thus, we speculated that the relatively low binding strength of IGF2BP3 to target genes in spermatogonia, spermatocytes, and somatic cells might also be one of the reasons why the knockout of IGF2BP3 did not have a significant impact on these cells. Collectively, both these hypotheses offer substantial insights into our comprehensive analysis of the regulatory mechanism of IGF2BP3 in spermatogenesis. Accordingly, these critical considerations have been thoroughly discussed in the Discussion section of the revised manuscript.

Figure 11

Figure 11.

A. Immunofluorescence of IGF2BP3 (green) and DDX4 (red) in adult testicular paraffin sections from 8-week-old mice. Arrowheads indicate CBs. Scale bar, 10 μ m.

B. Immunofluorescence of DDX4 (red) and PNA (green) in adult testicular paraffin sections from 8-week-old *Igf2bp3*^{+/-} and *Igf2bp3*^{-/-} mice. Scale bar, 10 μ m.

C. RS were incubated with ethynyl uridine (EU) for 10 h, and subsequently, synthesized RNA was visualized using the Click reaction (green). Arrowheads indicate EU-positive CBs.

D. Hierarchical clustering heatmap showing the expression profiles of 3,007 IGF2BP3 eCLIP-seq target genes across distinct testicular cells.

E. Metagene profiling of four gene clusters identified in Figure 11D. The left panel displays normalized IGF2BP3 eCLIP-seq signals from the IP fraction, while the right panel shows signals from SMIinput controls.

2. Since both IGF2BP3 and YBX2 can bind the 3'UTR of *Dot1l*, what impact does the m⁶A reading function of IGF2BP3 have on the level of *Dot1l* protein?

Answer: Thank you for your question. First, we speculated that you may get the conclusion that both IGF2BP3 and YBX2 could potentially bind the 3'UTR of *Dot1l* based on the result in Figure 5D in the previous manuscript. However, our data showed that YBX2 exhibited relatively low binding affinity for *Dot1l* RNA in mouse testes, even in the absence of IGF2BP3 (Figure 8B in response, Figure 5D in the revised manuscript). Furthermore, *Dot1l* was not identified as a target in YBX2 CLIP-seq data in mouse testes (see also Figure 8A in response) (Xu *et al.*, 2009), suggesting that YBX2 may not directly interact with *Dot1l* mRNA. Instead, it is possible that YBX2 indirectly associates with *Dot1l* through cooperation with other RBPs in testes. These findings collectively indicate that YBX2 does not directly bind to the 3'UTR of *Dot1l*.

Second, based on the above speculation, we think you may wonder whether the regulation of *Dot1l* primarily depends on the m⁶A binding activity of IGF2BP3 rather than that of YBX2. To address this, we constructed a modified IGF2BP3 with the GXXG motif (which corresponds to the m⁶A binding regions) in the KH3-4 domains mutated to GEEG (Figure 12A in response) (Fakhar *et al.*, 2024; Huang *et al.*, 2018; Huttelmaier *et al.*, 2005). Co-IP analysis revealed that the mutation in the KH3-4 domains of IGF2BP3 did not affect their mutual interaction with YBX2 even in the absence of RNAs as the scaffold (Figure 12B in response, Figure EV4L in the revised manuscript). More importantly, neither in the presence nor absence of YBX2 cooperation could the mutant mouse IGF2BP3 exert translational repression on CopGFP regulated by the mouse *Dot1l* 3'UTR containing the WT m⁶A motif, in the context of IGF2BP3-deleted HEK293T cells (Figures 12C-12E in response, Figures

EV4M-O in the revised manuscript). Together with our previous results, which demonstrated that overexpression of YBX2 alone had no obvious regulatory effect on the protein level of CopGFP (Figures 5G-5I in the previous manuscript), these findings further supported the established regulatory model in our study. In this model, the translational regulation of *Dot1l* primarily depends on the m⁶A binding activity of IGF2BP3 rather than that of YBX2. In this context, YBX2 predominantly functions as a translational regulatory cofactor in the IGF2BP3-YBX2 complex (Figure 6M of the previous manuscript).

Figure 12

Figure 12.

A. Schematic structures showing RNA-binding domains within IGF2BP proteins and a summary of IGF2BP variants used in this study. Yellow boxes are RRM domains, blue boxes are wild-type KH domains with GxxG motifs, and grey boxes are inactive KH domains with GxxG converted to GEEG.

B. Western blotting analysis of FLAG-mutIGF2BP3 and MYC-YBX2 in the Flag-mutant Igf2bp3 and Myc-Ybx2 co-transfected HEK293T cell lysates (input), and the lysate immunoprecipitation with anti-IgG, anti-FLAG or anti-MYC antibodies treated with RNase A (+), respectively.

C. qPCR analyses of the relative level of *CopGFP* mRNAs normalized to *mCherry* mRNAs. Cell lines were treated with 2 $\mu\text{g/ml}$ actinomycin D for 2 hours. Unpaired two-tailed *t*-test. N.S., not significant. Error bars, $n = 3$ biological replicates, mean \pm SEM.

D. Western blotting analysis of the protein levels of CopGFP under the regulation of Dot11 3'UTR with the overexpression of FLAG-tagged mutant IGF2BP3 or MYC-tagged YBX2. The level of mCherry was set as the internal control.

E. Histogram showing the ratios of CopGFP proteins (normalized to mCherry proteins) to the *CopGFP* mRNAs (normalized to *mCherry* mRNAs), corresponding to Figures 12C and 12D. Unpaired two-tailed *t*-test. N.S., not significant. Error bars, $n = 3$ biological replicates, mean \pm SEM.

3. As reported in previous literature, YBX2 binds to a series of gamete-specific mRNA, including PRM1 and PRM2 (<https://doi.org/10.1073/pnas.0404685102>). How many of these genes overlap with the CLIP-seq targets of YBX2 in this study? Has the binding capacity changed between YBX2 and PRM1 or PRM2?

Answer: Thank you for your question. Previously, 21 target genes of IGF2BP3 (including *Prm1* and *Prm2*) overlapping with the CLIP-seq targets of YBX2 have been observed in Figure S6A in the previous manuscript (see also Figure 8A in response). As you suggested, the YBX2 RIP-qPCR assay revealed that the binding capacity of YBX2 to *Prm1/2* remained unchanged in the absence of IGF2BP3, suggesting that testicular YBX2 may regulate *Prm1/2* in an IGF2BP3-independent manner (Figure 13 in response).

Figure 13

Figure 13.

YBX2 RIP-qPCR analyses of *Prm1* and *Prm2* in adult *Igf2bp3*^{+/-} and *Igf2bp3*^{-/-} testes. Unpaired two-tailed *t*-test. N.S., not significant, *****P* < 0.0001. Error bars, n = 3 biological replicates, mean ± SEM.

4. Inconsistency in YBX2 cooperation and stage-specific phenotypes. YBX2 knockout causes defects as early as spermatocytes, whereas *Igf2bp3* KO phenotypes manifest post-meiotically. The claim that YBX2's RNA-binding capacity is "completely lost" in *Igf2bp3* KO without affecting spermatocytes requires clarification. Please discuss possible compensatory mechanisms (e.g., redundant m6A readers or RNA-binding proteins in spermatocytes). Test whether YBX2 localization or stability is altered in *Igf2bp3* KO germ cells (e.g., by co-IP/WB/IF).

Answer: We sincerely appreciate your insightful comments. First, we would like to provide a clearer interpretation of YBX2 cooperation in mouse testes with IGF2BP3. In the previous manuscript, our statement that 'although the level of YBX2 remained unchanged in *Igf2bp3*-KO testes, where YBX2 no longer bound the target RNAs of IGF2BP3 (like *Dot11* and *Hdac11*) as it did in control testes' in Figure 5 did not imply that YBX2 completely lost its RNA-binding capacity in *Igf2bp3*-KO testes, but rather indicated that YBX2 lost its binding capacity to certain target RNAs of IGF2BP3 (such as *Dot11* and *Hdac11*) in *Igf2bp3*-KO testes. Therefore, to avoid any misunderstanding, we have revised the statement to: 'although the level of YBX2 remained unchanged in *Igf2bp3*-KO testes, its binding capacity to *Dot11* and *Hdac11* was significantly reduced compared to that in control testes'. Moreover, the limited overlap between the CLIP-seq targets (directly binding target genes) of IGF2BP3 and YBX2 also provided a plausible explanation for why YBX2 knockout mice exhibited a distinct phenotype in the testes compared to *Igf2bp3*-KO mice, which was highly consistent with your observations.

In addition, IF staining in the adult testes and WB analysis of round spermatids revealed that the localization and stability of YBX2 in *Igf2bp3*-KO spermatocytes and round spermatids were not significantly altered compared to those in WT germ cells

(Figure 14 in response). Collectively, these findings further suggested that the spermatids deficiency in IGF2BP3 KO testes was primarily attributed to the loss of IGF2BP3's RNA binding capacity, rather than the absence of YBX2.

Figure 14

Figure 14.

A. Immunofluorescence of YBX2 (red) and SYCP3 (green) in adult testicular paraffin sections from 8-week-old *Igf2bp3^{+/-}* and *Igf2bp3^{-/-}* mice. Arrow heads indicate the chromatoid bodies. Scale bar, 10 μ m.

B. Immunofluorescence of YBX2 (red) and PNA (green) in adult testicular paraffin sections from 8-week-old *Igf2bp3^{+/-}* and *Igf2bp3^{-/-}* mice. Arrow heads indicate the chromatoid bodies. Scale bar, 10 μ m.

C. Western blotting analysis of the protein level of YBX2 in RS from *Igf2bp3^{+/-}* and *Igf2bp3^{-/-}* mice. ACTB as a loading control.

5. There is potential for crosstalk with other m⁶A readers. Due to the unique germline

expression of IGF2BP3, it is necessary to determine whether its loss affects the expression or subcellular distribution of other m⁶A readers (e.g. YTHDF1/2/3 and YTHDC1/2). Simple WB/qRT-PCR analysis of key readers in WT vs. KO testes would address this.

Answer: Thank you for your suggestions. As shown in Figure S1A of the previous manuscript, the expression levels of all these m⁶A readers had been analyzed using our RNA-seq data from adult *Igf2bp3*^{+/-} and *Igf2bp3*^{-/-} round spermatids. The results indicated that only *Igf2bp1*, *Ythdc1*, and *Ythdf1* exhibited a slight alteration at the transcriptional level in the *Igf2bp3*-KO RS (Figure 15A in response). However, Western blotting analyses of adult *Igf2bp3*^{+/-} and *Igf2bp3*^{-/-} testes revealed no significant differences in the protein level of IGF2BP1, YTHDC1, and YTHDF1, thereby indicating no remarkable crosstalk between IGF2BP3 and other m⁶A readers during spermiogenesis (Figures 15B and 15C in response, Figure S2B in the previous manuscript).

Figure 15.

A. RNA-seq expression analysis of m⁶A readers in testes of *Igf2bp3*^{+/-} (blue) and *Igf2bp3*^{-/-} (red) mice. Data are presented as mean ± SEM. Statistical significance was determined by unpaired *t*-test: **P* < 0.05, ***P* < 0.01; N.S., not significant.

B. Western blotting analysis of the protein levels of IGF2BP1 and IGF2BP2 in *Igf2bp3*^{+/-},

Igf2bp3^{+/-}, and *Igf2bp3*^{-/-} testes. The values below each band represent the relative expression levels of each protein, normalized using ACTB as a loading control.

C. Western blotting analysis of the protein level of YTHDC1 and YTHDF1 in *Igf2bp3*^{+/-} and *Igf2bp3*^{-/-} mice testes. The values below each band represent the relative expression levels of each protein, normalized using ACTB as a loading control.

6. The subcellular localisation of IGF2BP3 and its potential association with P-bodies. The punctate staining of IGF2BP3 observed in round spermatids strongly resembles that of cytoplasmic granules (e.g., P-bodies or chromatid bodies). As IGF2BP3 is known to regulate mRNA localisation to P-bodies (see, for example, the article 'm⁶A modification negatively regulates translation by switching mRNA from polysome to P-body via IGF2BP3' in *Molecular Cell*), this connection should be explored. Co-stain with P-body markers (e.g., DCP1A or others) to confirm the granule identity of IGF2BP3.

Answer: We appreciate your comments and suggestions. Previous studies have revealed that the CB is a single, large, RNA-rich granule present in the haploid spermatids and functions as a specialized P-body essential for post-transcriptional mRNA regulation during spermatogenesis (Cassani & Seydoux, 2024). According to your advice, we conducted co-staining of IGF2BP3 and DDX4 (a marker of CB) in mouse testes, which confirmed the granule identity of IGF2BP3 in round spermatids (see also response to your Major concern 1 and Figure 11 in response). Notably, we had also observed this characteristic of IGF2BP3 in mouse testes, as shown in Figure 1 of our previous manuscript. And the possibility of a similar mechanism occurring in mouse RS and HeLa cells, where IGF2BP3 has been shown to drive hyper-m⁶A methylated RNAs into P-body to negatively regulate their translation has been thoroughly discussed in the Discussion section (Shan *et al*, 2023).

7. Assess whether *Igf2bp3* KO disrupts granule formation or RNA localization (e.g. via RNA-FISH for target transcripts such as *Dot11* or *Hdac11*). This would significantly strengthen the mechanistic model of translational repression.

Answer: We sincerely appreciate your important suggestions. Previously, we had revealed that the CB formation and wherein the RNA translocation were not be disrupted by the deletion of IGF2BP3 in round spermatids by DDX4 staining and EU culture experiment (see also response to your Major concern 1 and Figure 11 in response). Furthermore, to explore the potential regulatory role of testicular IGF2BP3 in cytoplasmic compartmentalization of its target RNAs, such as *Dot1l* or *Hdac11*, the destination gene-MS2-BoxB tethering reporter system is currently the available approach, as reported by Shan *et al* (Shan *et al.*, 2023). However, this method requires the use of both *Dot1l/Hdac11*-MS2-BoxB reporter mice and λ N-mCherry mice, which makes this experiment highly challenging to implement at present. Besides, due to the space limitations of this article, we believe that investigating the subcellular partition of RNA-binding proteins and their target RNAs, along with the potential mechanisms of RNA translational repression mediated by CB or other cytoplasmic organelles, and their interplay with dynamic enriched m⁶A- or other RNA-modifications during spermatogenesis, represents a highly promising and worthwhile endeavor that merits systematic exploration as an independent project in the future.

Other concerns:

1. Are there any differences between subfertile and infertile males? What about the infertility ratio? Which type of male is used in the experiments?

Answer: Thank you for your questions. In our previous results, when mated with wild-type female mice, adult *Igf2bp3*^{-/-} male mice exhibited variable fertility, with approximately 65% displaying sub-fertility and 10% being completely infertile (Figure 2C in the previous manuscript). Except for the testes weight and semen quality, no other significant differences were observed among these IGF2BP3-KO males prior to sacrifice (Figures 2D-2G). As is well known, semen quality must be significantly impaired before it substantially affects male fertility (Minhas *et al*, 2025; World Health Organization., 2010). Therefore, to minimize variability and more accurately reflect the mechanisms by which IGF2BP3 regulates spermatogenesis, *Igf2bp3*^{-/-} male mice with impaired fertility were selected for the experiments

presented in Figures 3 to 5. It is important to clarify that we could not specifically select sub-fertile or infertile male mice for the siRNA microinjection experiment, as it was conducted using 4-week-old mice in Figure 7. Therefore, the left and right testes of the same knockout mouse were injected with siNC and *siDot11/Hdac11*, respectively. Collectively, although some variations in fertility were observed among IGF2BP3 knockout mice, our data demonstrate a high level of reliability and reproducibility.

2. In Figure 3, the immunofluorescence images of H2A/H2B/H3/H4 in sperm heads lack clarity. Add arrows/insets to highlight specific abnormalities (e.g., nuclear retention or mislocalization) in KO sperm.

Answer: Thank you for your suggestion, and we have added arrows accordingly in the revised manuscript.

3. Technical limitation: siRNA penetration across the blood-testis barrier (BTB) is notoriously inefficient in 4-weeks old mice. The sub-fertile phenotype of *Igf2bp3* KO mice may not conclusively demonstrate functional recovery, as off-target effects or incomplete knockdown cannot be ruled out. Experimental rigor: (a) At least two independent siRNAs per target gene (*Dot11* and *Hdac11*) must be used to confirm specificity. (b) Fertility restoration assays are critical: Assess whether siRNA-treated *Igf2bp3* KO mice show improved fertility (e.g., litter size, sperm counts, or mating trials).

Answer: Thank you for your valuable comments and suggestions. First, we would like to sincerely apologize for any inadequacies in this experiment. Specifically, siRNAs were directly injected into the seminiferous tubules following Brinster's protocol, rather than being administered through the testis interstice by diffusion (Figure 16A in response) (Brinster & Zimmermann, 1994). Therefore, there was no requirement for the siRNAs to cross the BTB. In addition, 5'Cy3 was covalently conjugated to the siRNAs, which were chemically modified with 2'-OM (2-methoxyethyl) and cholesterol to enhance their stability and biodistribution, as reported in a previous

study (Tan *et al*, 2023). 24 hours post siRNA injection, FACS analysis revealed that approximately 34% of Cy3-positive haploid cells could be isolated from the 4-week-old testes (Figures 16B-16D in response). Therefore, this preliminary experiment provided us with a feasible protocol to efficiently knockdown DOTL1 and HDAC11 specifically in round spermatids by microinjecting siRNAs into 4-week-old *Igf2bp3*^{-/-} seminiferous tubules.

Second, based on your suggestions, additional independent siRNAs targeting *Dot1l* and *Hdac11* (*siDot1l-2*, n = 7; *siHdac11-2*, n = 7) were used in the siRNA microinjection experiments. Western blotting and sperm analyses confirmed that these siRNAs effectively contributed to the partial rescue of histone retention and protamine reduction, as well as improved motility and concentration of *Igf2bp3*^{-/-} epididymal spermatozoa, showing high consistency with the previous results obtained using *siDot1l-1* and *siHdac11-1* (Figures 16E-16H in response, Figures EV5D, EV5E, EV5I and EV5K in the revised manuscript).

Third, we tested the litter size with 9 to 16-weeks-old infertile *Igf2bp3*^{-/-} male mice injected with siNC (n = 13) and *siDot1l/Hdac11-1* (n = 16), separately. Notably, fertility was restored in 5 out of 16 infertile *Igf2bp3*^{-/-} male mice following *siDot1l/Hdac11-1* injection within two months, whereas no such recovery was observed in the siNC-treated group (Figure 16I in response, Figure 7I in the revised manuscript).

Collectively, despite the limited number of available *Igf2bp3*^{-/-} male mice within the revision timeframe, these findings suggested that siRNAs targeting *Dot1l* and *Hdac11* could partially restore fertility in infertile *Igf2bp3*^{-/-} males, thus emphasizing the essential role of IGF2BP3-*Dot1l/Hdac11* axis in sperm development and male fertility.

Figure 16

Figure 16.

A. Bright field images of siRNAs directly injected into the seminiferous tubules. Blue dye indicates the trypan blue.

B. Fluorescence and bright field images of 4-week-old testes 24-hours post siRNA injection.

Scale bar, 1 mm.

C. Representative FACS strategy for isolating the Cy3-positive haploid cells from testes injected with siRNA covalently bound by Cy3.

D. Fluorescence of Cy3⁺ or Cy3⁻ haploid cells from *Igf2bp3*^{-/-} mouse testes injected with siRNA covalently bound by Cy3 or not. Scale bar, 80 μ m.

E-F. Western blotting analysis of the protein level of DOT1L, HDAC11, H3K79me2 and H4K5/8/12ac from adult *Igf2bp3*^{+/-} and *Igf2bp3*^{-/-} mouse testes injected with negative control siRNA (siNC), siRNA targeting *Dot1l* (si*Dot1l*-2) or siRNA targeting *Hdac11* (si*Hdac11*-2). ACTB serves as the internal control of DOT1L and HDAC11, H3 serves as the internal control of H3K79me2, H4 serves as the internal control of H4K5/8/12ac.

G-H. CASA of the percentage of motile sperm (F) and the corresponding concentration of epididymal spermatozoa (G) from adult *Igf2bp3*^{+/-} and *Igf2bp3*^{-/-} mouse injected with siRNA (siNC), siRNA targeting *Dot1l* (si*Dot1l*-2) or siRNA targeting *Hdac11* (si*Hdac11*-2). Unpaired two-tailed *t*-test. N.S., not significant, **P* < 0.05, ***P* < 0.01, ****P* < 0.001, *****P* < 0.0001. Each bar represents the mean \pm SEM from 4 biological replicates.

I. Fertility recovery of infertile *Igf2bp3*^{-/-} mouse injected with siNC or si*Dot1l/Hdac11*-1. Unpaired two-tailed *t*-test. **P* < 0.05. Each bar represents the mean \pm SEM.

Referee #3:

Reviewer comments: IGF2BP3 Recognizes m6A to Regulate Histone-to-protamine Replacement during Mouse Sperm Development, Wang et al

General summary and importance :

The discovery of the reversible nature of m6A in RNA has inspired numerous studies to identify the biological role of m6A dynamics - including the characterization of the many m6A binding proteins that are required for m6A to have a role in gene regulation. In most model organisms studied the role of methylating A to m6A is particularly obvious during germ cell maturation, with "similar" phenotypes for male and female germ cells. Additionally, several m6A binding proteins affect fertility if mutated. In mammals this is also shown to be a key feature for mice lacking the m6A

demethylase, ALKBH5.

This work was initiated with the identification of IGF2BP3 being relevant for m6A during spermatogenesis based on expression profiling, and Igf2bp3 mutation led to male sub/infertility. They identified the histone to protamine transition to be defect in Igf2bp3 mutant mice. Some of the analysis, e.g. the detailed characterization of the histone-protamine transition, the overlap of IGF2BP3 pulldown and MeRIP, and mutating the m6A motif of Dot11 and Hdac11 are particularly impressive, and most studies neatly combine molecular analysis with phenotypic characteristics.

Answer: We appreciate your positive comments and acknowledgement of our study's importance.

Major comment:

My only major concern relates to the generation and identification of Igf2bp3 mutant mice. I cannot find information regarding which of the two different IGF2BP3 antibodies used for immunostaining of cells: The Bethyl antibody used is raised against the C-terminus (although described as A303-426A in the table I assume it is raised against the C-terminus aa 529-579 as described in Remark to the antibody table). The Abcam antibody (if the upper described in antibody table remark?) is raised against the N-terminus. Thus, they should differ in their ability to identify a truncated IGF2BP3 protein still containing the two RRM domains. The truncated protein, if at all present, would not be identified in the Western blot shown for sizes only corresponding to the wild-type protein. Was any mRNA analysis done to quantitate the truncated mRNA transcript (if having a premature stop in the interrupted mutant) and possible address its ability to be translated to a protein and provide a dominant-negative effect?

Answer: Thanks for your question. Firstly, we sincerely apologized for the lack of clarity regarding the two different IGF2BP3 antibodies used in the previous manuscript. The Bethyl antibody (catalog number: A303 - 426A) recognizes the C-terminal region of IGF2BP3 spanning amino acids 529 to 579, and it was only used in the Figure S2B of previous manuscript. And the Abcam antibody (catalog number:

ab177477) targets the N-terminal region of IGF2BP3, covering amino acids 1 to 250. This Abcam antibody was employed extensively in multiple experiments, including immunofluorescence staining, western blotting, RIP, and eCLIP assays. This information has been described clearly in the figure legends in the revised manuscript. Moreover, larger gel images of western blotting using C-IGF2BP3 and N-IGF2BP3 antibodies revealed the absence of the truncated IGF2BP3 protein in *Igf2bp3*^{-/-} testes (Figures 17A and 17B in response). Consistently, qPCR analysis targeting the *Igf2bp3* transcripts confirmed that the level of truncated *Igf2bp3* transcript in *Igf2bp3*^{-/-} testes was extremely low (Figure 17C in response). Therefore, the spermatogenesis disorders observed in IGF2BP3 knockout mice are unlikely to result from a dominant-negative effect.

Figure 17

Figure 17.

A. Schematically depicts the domain structures of full-length IGF2BP3 and truncated IGF2BP3.

B. Western blotting analysis of the protein levels of IGF2BP3 in *Igf2bp3*^{+/+}, *Igf2bp3*^{+/-}, and *Igf2bp3*^{-/-} testes. Both the N-terminal and C-terminal antibodies recognizing IGF2BP3 were

utilized in this assay. ACTB serves as a loading control. The dashed box indicates the potential position of the shortened form of IGF2BP3.

C. qPCR analysis of the relative expression level of *Igf2bp3* normalized to β -Actin in adult *Igf2bp3*^{+/-} and *Igf2bp3*^{-/-} testes. Unpaired two-tailed *t*-test. *****P* < 0.0001. Each error bar represents the mean \pm SEM from three biological replicates.

Minor comments:

Some language/grammar editing would improve the manuscript.

Line 26 Introduction page 3: I assume m6A, despite different levels, is present in all spermatogenic cells (delete almost).

Answer: We have corrected it as suggested in Line 65 in the revised manuscript.

Line 28 Introduction page 3: Correct singular/plural of m6A modifying enzymes.

Answer: We have corrected it as suggested in Line 67 in the revised manuscript.

Line 6 Introduction page 4: correct "conservative in" to "conserved between"

Answer: We have corrected it accordingly in Line 78 in the revised manuscript.

Line 24 Results page 4: delete "relatively" (since compared is also used in the sentence)

Answer: We have corrected it as suggested in Line 99 in the revised manuscript.

Lines 26 Results page 4: this sentence is very long and must be corrected grammatically.

Answer: We have corrected it accordingly in Line 101-106 in the revised manuscript.

Line 16 Results page 5: I don't agree that this is a good reason for not also delete the RRM domains. Do you have any additional information to provide on this (see also major comment above)?

Answer: We sincerely appreciate your question. In addition to the rationale for

employing the gene knockout strategy to generate *Igf2bp3*^{-/-} mouse provided in the previous manuscript, there were several other important considerations as follows.

In principle, large-scale gene knockout is a widely accepted strategy for completely eliminating any potential residual proteins to investigate the gene function. In the case of IGF2BP3, exon 1 should be specifically targeted to knockout the RRM domain. However, genomic annotation analysis reveals that *Igf2bp3* can produce multiple isoforms, including isoform X1, which differs in its 3' end compared to the full-length transcript but still encodes the RRM domain. Furthermore, downstream of the second exon of *Igf2bp3* encodes a long non-coding RNA (lncRNA), D030074P21Rik, whose biological function remains unknown yet (Figure 18A in response). Besides, RT-PCR analysis confirmed that these transcripts were indeed expressed in mouse testis (Figure 18B in response). To avoid the dominant-negative effect that could arise from the truncated isoform X1 due to a reading frame shift, as well as to eliminate the potential confounding influence of lncRNA deletion on the phenotype of *Igf2bp3*^{-/-} mice, we did not construct the KO mice by targeting exon 1 or exon 2. Instead, exon 6, which encodes the KH1 domain, was targeted in our study. Luckily, no protein fragment corresponding to RRM1/2 had been observed in the *Igf2bp3*^{-/-} testes. Coincidentally, a similar strategy targeting the exon 6 was also independently employed by Dr. He's laboratory to generate *Igf2bp3* knockout mice during our manuscript submission (Huang *et al*, 2025). Although they did not explicitly explain the rationale for this strategy, their independent approach targeting exon 6 corroborates the appropriateness of generating *Igf2bp3*^{-/-} mice.

Figure 18

Figure 18

A. Schematically depicts the gene structures of *Igf2bp3* (NM_023670.3 and isoform X1) and lncRNA *D030074P21Rik*, showing exons, introns, and RT-PCR primer binding sites.

B. RT-PCR analysis results showing amplified products from primer pairs P1+P2, P3+P2, and P4+P5 alongside a DNA ladder.

Line 25 Results page 7: these results are very nice, and I wonder if you also have information regarding any conserved motif (like the m⁶A motif?)

Answer: We appreciated your positive comments. Except for the m⁶A motif, the CA-rich element, which is essential for IGF2BPs binding in previous works, was also identified in the m⁶A binding motifs of IGF2BP3 in our study (Figure S7E in the previous manuscript). Thus, we believed that the CA-rich element would be a conserved motif of IGF2BP3 in mouse testes.

Section starting at line 11 page 9: could you elaborate briefly on the possibility of the phenotype of *Igf2bp3* mutant mice to depend on m⁶A binding and/or interaction with YBX2, independently?

Answer: Thank you for your question. In our results, overexpression of YBX2 alone had no significant effect on the protein level of CopGFP, which was regulated by the 3'UTR of *Dot1l* (Figures 5G-5I in previous manuscript). In addition, IGF2BP3-YBX2 complex reduced the translation efficiency of CopGFP containing the mutant m⁶A motifs in the 3'UTR of *Dot1l* and *Hdac11* (Figures 6H-6L and S7K-S7N in previous manuscript). Furthermore, the translational repression mediated by IGF2BP3 with a mutant m⁶A-binding domain was markedly reduced on CopGFP regulated by the 3'UTR of *Dot1l*, even in the presence of YBX2 (see also the response to major concern 2 of the Reviewer 2, Figure 12 in response). Collectively, these findings indicated that IGF2BP3 regulated *Dot1l* through a mechanism that relied on both its m⁶A-binding capacity and the translational repression function of YBX2. This further supported the proposed working model of IGF2BP3 in spermatogenesis illustrated in the Figure 6M of the previous manuscript.

Any comments relating to the (sub/in) fertility of female *Igf2bp3* mutant mice? Based on line 15 page 15 I assume they have normal fertility.

Answer: Thanks for your question. The *Igf2bp3*^{-/-} female mouse were fertile, and this information has been added to the Materials and Methods section of the revised manuscript.

References:

Anuar ND, Kurscheid S, Field M, Zhang L, Rebar E, Gregory P, Buchou T, Bowles J, Koopman P, Tremethick DJ *et al* (2019) Gene editing of the multi-copy H2A.B gene and its importance for fertility. *Genome Biol* 20: 23

Biswas J, Patel VL, Bhaskar V, Chao JA, Singer RH, Eliscovich C (2019) The structural basis for RNA selectivity by the IMP family of RNA-binding proteins. *Nat Commun* 10: 4440

Brinster RL, Zimmermann JW (1994) Spermatogenesis following male germ-cell transplantation. *Proceedings of the National Academy of Sciences of the United States of America* 91: 11298-11302

Cassani M, Seydoux G (2024) P-body-like condensates in the germline. *Seminars in cell & developmental biology* 157: 24-32

Conway AE, Van Nostrand EL, Pratt GA, Aigner S, Wilbert ML, Sundararaman B, Freese P, Lambert NJ, Sathe S, Liang TY *et al* (2016) Enhanced CLIP Uncovers IMP Protein-RNA Targets in Human Pluripotent Stem Cells Important for Cell Adhesion and Survival. *Cell reports* 15: 666-679

Fakhar M, Gul M, Li W (2024) Interactive Structural Analysis of KH3-4 Domains of

IGF2BPs with Preferred RNA Motif Having m(6)A Through Dynamics Simulation Studies. *Int J Mol Sci* 25

Goudarzi A, Zhang D, Huang H, Barral S, Kwon OK, Qi S, Tang Z, Buchou T, Vitte AL, He T *et al* (2016) Dynamic Competing Histone H4 K5K8 Acetylation and Butyrylation Are Hallmarks of Highly Active Gene Promoters. *Mol Cell* 62: 169-180

Griffin KN, Walters BW, Li H, Wang H, Biancon G, Tebaldi T, Kaya CB, Kanyo J, Lam TT, Cox AL *et al* (2022) Widespread association of the Argonaute protein AGO2 with meiotic chromatin suggests a distinct nuclear function in mammalian male reproduction. *Genome Res* 32: 1655-1668

Huang F, Wang Y, Zhang X, Gao W, Li J, Yang Y, Mo H, Prince E, Long Y, Hu J *et al* (2025) m(6)A/IGF2BP3-driven serine biosynthesis fuels AML stemness and metabolic vulnerability. *Nat Commun* 16: 4214

Huang H, Weng H, Sun W, Qin X, Shi H, Wu H, Zhao BS, Mesquita A, Liu C, Yuan CL *et al* (2018) Recognition of RNA N(6)-methyladenosine by IGF2BP proteins enhances mRNA stability and translation. *Nature cell biology* 20: 285-295

Huttelmaier S, Zenklusen D, Lederer M, Dichtenberg J, Lorenz M, Meng X, Bassell GJ, Condeelis J, Singer RH (2005) Spatial regulation of beta-actin translation by Src-dependent phosphorylation of ZBP1. *Nature* 438: 512-515

Ketchum CC, Larsen CD, McNeil A, Meyer-Ficca ML, Meyer RG (2018) Early histone H4 acetylation during chromatin remodeling in equine spermatogenesis. *Biol Reprod* 98: 115-129

Lehtiniemi T, Bourgerie M, Ma L, Ahmedani A, Makela M, Asteljoki J, Olotu O,

Laasanen S, Zhang FP, Tan K *et al* (2022) SMG6 localizes to the chromatoid body and shapes the male germ cell transcriptome to drive spermatogenesis. *Nucleic Acids Res* 50: 11470-11491

Li W, Wu J, Kim SY, Zhao M, Hearn SA, Zhang MQ, Meistrich ML, Mills AA (2014) Chd5 orchestrates chromatin remodelling during sperm development. *Nat Commun* 5: 3812

Meikar O, Vagin VV, Chalmel F, Sostar K, Lardenois A, Hammell M, Jin Y, Da Ros M, Wasik KA, Toppari J *et al* (2014) An atlas of chromatoid body components. *RNA (New York, NY)* 20: 483-495

Minhas S, Boeri L, Capogrosso P, Cocci A, Corona G, Dinkelman-Smit M, Falcone M, Jensen CF, Gul M, Kalkanli A *et al* (2025) European Association of Urology Guidelines on Male Sexual and Reproductive Health: 2025 Update on Male Infertility. *Eur Urol* 87: 601-616

Ollinger R, Childs AJ, Burgess HM, Speed RM, Lundegaard PR, Reynolds N, Gray NK, Cooke HJ, Adams IR (2008) Deletion of the pluripotency-associated Tex19.1 gene causes activation of endogenous retroviruses and defective spermatogenesis in mice. *PLoS Genet* 4: e1000199

Qian MX, Pang Y, Liu CH, Haratake K, Du BY, Ji DY, Wang GF, Zhu QQ, Song W, Yu Y *et al* (2013) Acetylation-mediated proteasomal degradation of core histones during DNA repair and spermatogenesis. *Cell* 153: 1012-1024

Ren F, Miao R, Xiao R, Mei J (2021) m(6)A reader Igf2bp3 enables germ plasm assembly by m(6)A-dependent regulation of gene expression in zebrafish. *Sci Bull*

(Beijing) 66: 1119-1128

Shan T, Liu F, Wen M, Chen Z, Li S, Wang Y, Cheng H, Zhou Y (2023) m(6)A modification negatively regulates translation by switching mRNA from polysome to P-body via IGF2BP3. *Molecular cell*

Shiota H, Barral S, Buchou T, Tan M, Coute Y, Charbonnier G, Reynoird N, Boussouar F, Gerard M, Zhu M *et al* (2018) Nut Directs p300-Dependent, Genome-Wide H4 Hyperacetylation in Male Germ Cells. *Cell Rep* 24: 3477-3487 e3476

Tan HH, Wang WX, Zhou CJ, Wang YF, Zhang S, Yang PL, Guo R, Chen W, Zhang JW, Ye L *et al* (2023) Single-cell RNA-seq uncovers dynamic processes orchestrated by RNA-binding protein DDX43 in chromatin remodeling during spermiogenesis. *Nature communications* 14

Van Nostrand EL, Pratt GA, Shishkin AA, Gelboin-Burkhart C, Fang MY, Sundararaman B, Blue SM, Nguyen TB, Surka C, Elkins K *et al* (2016) Robust transcriptome-wide discovery of RNA-binding protein binding sites with enhanced CLIP (eCLIP). *Nat Methods* 13: 508-514

Vong YH, Sivashanmugam L, Leech R, Zaucker A, Jones A, Sampath K (2021) The RNA-binding protein Igf2bp3 is critical for embryonic and germline development in zebrafish. *PLoS Genet* 17: e1009667

World Health Organization. (2010) *WHO laboratory manual for the examination and processing of human semen*. World Health Organization, Geneva

Xu M, Medvedev S, Yang J, Hecht NB (2009) MIWI-independent small RNAs

(MSY-RNAs) bind to the RNA-binding protein, MSY2, in male germ cells. *Proc Natl*

Acad Sci U S A 106: 12371-12376

Yu YE, Zhang Y, Unni E, Shirley CR, Deng JM, Russell LD, Weil MM, Behringer RR,

Meistrich ML (2000) Abnormal spermatogenesis and reduced fertility in transition

nuclear protein 1-deficient mice. *Proc Natl Acad Sci U S A* 97: 4683-4688

Dear Dr. Wang,

Thank you for submitting a revised version of your manuscript. We have now received input from two of the original reviewers, who now support acceptance of the manuscript after a minor textual revisions. There are also a few editorial points that need to be addressed before I can extend official acceptance of the manuscript:

1. Please submit up to five keywords for your manuscript.
2. Please correct the order and headings of the manuscript sections to: Abstract / Keywords / Introduction / Results / Discussion / Methods / Data Availability / Acknowledgments / Disclosure and Competing Interests Statement / References / Figure Legends / Tables / Expanded View Figure Legends.
3. Please add the main figure legends and Expanded View figure legends to the manuscript text after "References" section.
4. Please check that the funding information is correct and identical both in the manuscript and our online system; currently, 32370840 is not listed in the "Acknowledgments" section.
5. CRedit has replaced the traditional author contributions section because it offers a systematic, machine-readable author contributions format that allows for more effective research assessment. Please remove the Authors Contributions from the manuscript and use the free text boxes beneath each contributing author's name in our online submission system to add specific details on the author's contribution. More information is available in our guide to authors.
6. Please rename "Competing interests" section into "Disclosure and competing interests statement" (further info: <https://www.embopress.org/page/journal/14602075/authorguide#conflictsofinterest>).
7. For Datasets EV1 - EV5, please remove the legends from the manuscript text file and add to each dataset file in a separate tab/sheet.
8. Please compile the Appendix in a single file and upload it in PDF format, with the corresponding legends placed underneath each figure. Please also add a table of contents including page numbers to the first page of Appendix.
9. For movies, please remove the legends from the manuscript text file and zipped with each movie file. Further information is available here: <https://www.embopress.org/page/journal/14602075/authorguide#expandedview>
10. All Materials and Methods need to be described in the main text using our 'Structured Methods' format. According to this format, the Methods section includes a Reagents and Tools Table (listing key reagents, experimental models, software and relevant equipment and including their sources and relevant identifiers) followed by a Methods and Protocols section describing the methods, ideally using a step-by-step protocol format. The aim is to facilitate adoption of the methodologies across labs. Please download and fill our Reagents and Tools Table template (.docx), which you can find in our author guidelines: <https://www.embopress.org/page/journal/14602075/authorguide#structuredmethods>
When submitting your revised manuscript, please do not include the Reagents and Tools Table in the Methods section of the manuscript but upload it as a separate file choosing the file type "Reagent Table".
An example of a Method paper with Structured Methods can be found here: <https://www.embopress.org/doi/10.15252/msb.20178071>.
11. Our data editors have flagged the following issues in figure legends that need correcting:
 - Please provide the exact p values in the legends of figures 2B, C, G; 4H, 7F, H, I; EV1 D, E, G, H, EV2 J, K; EV4 C, EV5 H, I, J; S3 F.
 - Please indicate the statistical test used for data analysis in the legends of figures EV3 D, S3 D, E.
 - Please define the box plots in terms of minima, maxima, centre, bounds of box and whiskers, and percentile in the legends of figure S1 C.
 - Please provide information on the number and nature of replicates in the legends of figures S1 C, S3 D E.
 - Please define the dotted borders in the legends of figures 1C, D, E; 3E, F; 5B, 7D, E; EV2 I, EV5 F, G; S1 D.
12. Papers published in The EMBO Journal are accompanied online by a 'Synopsis' to enhance discoverability of the manuscript. It consists of A) a short (1-2 sentences) summary of the findings and their significance, B) 3-4 bullet points highlighting key results (the highlights can be repurposed for this) and C) a synopsis image that is 550x300-600 pixels large (width x height, jpeg or png format). You can either show a model or key data in the synopsis image. Please note that the image size is rather small and that text needs to be readable at the final size.
13. As part of the EMBO Press transparent editorial process, The EMBO Journal will publish online a Peer Review File to accompany accepted manuscripts. This file will be published in conjunction with your paper and will include the anonymous referee reports, your point-by-point response and all pertinent correspondence relating to the manuscript, including decision letters. Please note that the Author Checklist will be published at the end of the Peer Review File.
Please let us know if you want to remove or not any figures or data from the Peer Review File prior to publication. Please note that retaining unpublished data in the Peer Review File means that these count as published and that the Peer Review File would need to be referenced in future publications.

With best wishes,

leva

leva Gailite, PhD
Senior Scientific Editor
The EMBO Journal
Meyerhofstrasse 1
D-69117 Heidelberg
Tel: +4962218891309
i.gailite@embojournal.org

We realize that it is difficult to revise to a specific deadline. In the interest of protecting the conceptual advance provided by the work, we recommend a revision within 3 months (20th Jan 2026). Please discuss the revision progress ahead of this time with the editor if you require more time to complete the revisions.

Referee #1:

The authors have largely addressed my (Reviewer 1) original concerns and improved the manuscript. There are a few remaining points that were incompletely addressed and should be completed before the manuscript moves forward.

- Reviewer Point 1: The presence of substantial IGF2BP3 protein expression in spermatogonia (Fig 1B) is acknowledged in the author response document but not actually addressed in the revised manuscript. Some acknowledgment of this expression should be added to the relevant section of the results (lines 110-119).
- Reviewer point 8: Regarding the 534 genes that are eCLIP-seq targets in both germ and somatic cells, the authors point out that these transcripts appear to be bound with lower affinity compared to the 3,007 germ cell-specific targets, justifying subsequent focus on the germ cell-specific targets. This is a useful point and should be mentioned briefly the relevant section of the results (line 201) or in the discussion.
- Reviewer Point 16: The reference for the YBX2 testis CLIP-seq dataset (PMID:19597149, Xu et al. 2009) was added to Figure EV4A. However, it is still not included in the References section. The reference should be added in the main text near where the dataset is mentioned in the results (line 255-256) and cited appropriately.
- There are quite a few small grammatical mistakes that should be fixed.

Referee #3:

The authors have carried out several experiments to improve or clarify the results and conclusions. All my concerns have been fully answered, so to me, this manuscript is ready for publication.

The authors addressed the remaining editorial changes.

Dear Mei,

Thank you for incorporating the final editorial requests in the manuscript. I am now pleased to inform you that your manuscript has been accepted for publication. Congratulations on a nice study!

Before we forward your manuscript to the publishers, I will look into the synopsis text that you kindly provided and will let you know by the beginning of the next week if any textual edits to the journal style are needed.

If you have any questions, please do not hesitate to contact the Editorial Office or me directly. Thank you for this interesting contribution to The EMBO Journal!

Best wishes,

leva

leva Gailite, PhD
Senior Scientific Editor
The EMBO Journal
Meyerhofstrasse 1
D-69117 Heidelberg
Tel: +4962218891309
i.gailite@embojournal.org

Please note that it is The EMBO Journal policy for the transcript of the editorial process (containing referee reports and your response letters) to be published as an online supplement to each paper. If you should prefer removal of any referee-only figures included in the point-by-point response(s), e.g. because they may still be used for future publication or because they have been reproduced from published work by others, please do let us know immediately via response email.

More information is available here: https://www.embopress.org/transparent-process#Review_Process